# SVD as a Fast Interpretability Method for Transformers

**Min Xue** [1]    **Artur Andrzejak** [1]

## Abstract

Mechanistic interpretability of Transformer models commonly relies on training auxiliary proxy models, such as Sparse Autoencoders or Cross-Layer Transcoders. While effective, these post-hoc approaches introduce approximation bias and incur substantial computational overhead. We propose an alternative, training-free interpretability framework that directly exploits the Singular Value Decomposition (SVD) of weight matrices in Transformer MLP sublayers. By operating natively on model parameters, our method improves scalability while preserving fidelity to the original weights. We show that the projection matrices of MLP sublayers admit a natural decomposition into orthogonal, interpretable rank-1 subspaces, which we term **Detector-Effector Units** (DEUs). Within each unit, a singular vector functions as a detector of input patterns and modulates a coupled effector vector that encodes output semantics. Building on this structure, we introduce **Subspace Contribution Analysis** (SCA), a diagnostic method that quantifies the direct causal contribution of individual native subspaces to model predictions. Experiments across the GPT-2 family demonstrate that our framework, **Native Network Anatomy** (NaNA), identifies dominant functional pathways with orders-of-magnitude efficiency gains over training-based interpretability baselines, while maintaining weight fidelity. Our results suggest that SVD-based analyses provide a scalable and faithful alternative to learned proxy approaches for mechanistic interpretability.

---

[1]Institute of Computer Science, Heidelberg University, Germany. Correspondence to: Min Xue <min.xue@uni-heidelberg.de>.

*Proceedings of the 43rd International Conference on Machine Learning*, Seoul, South Korea. PMLR 306, 2026. Copyright 2026 by the author(s).

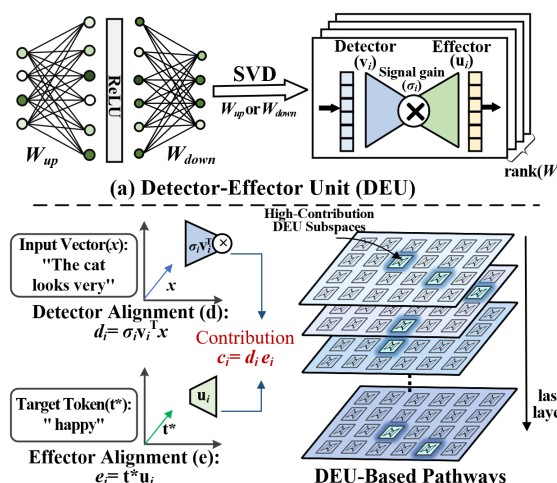

*Figure 1.* **The Native Network Anatomy (NaNA) Framework.** **(a) Detector-Effector Unit (DEU).** The MLP weight matrix is decomposed into interpretable rank-1 subspaces via SVD. Each DEU consists of a detector ($\mathbf{v}_i$), an effector ($\mathbf{u}_i$), and a signal gain ($\sigma_i$). **(b) Subspace Contribution Analysis (SCA).** This method quantifies the contribution of individual DEU subspaces to a target prediction. By projecting the input onto the detector and the target onto the effector, we obtain a contribution map that reveals sparse, dominant pathways across the layers.

## 1. Introduction

Large Language Models (LLMs) (Radford et al., 2019; Touvron et al., 2023) built upon the Transformer architecture (Vaswani et al., 2017) have demonstrated remarkable capabilities across a wide range of tasks. However, their internal decision-making mechanisms remain largely opaque. With substantial progress already achieved in analyzing attention heads (Zheng et al., 2024; Zhao et al., 2025; Elhelo & Geva, 2025), the focus has increasingly shifted towards Multi-Layer Perceptron (MLP) components, aiming to disentangle polysemantic neurons (Scherlis et al., 2022).

Recent approaches utilizing Sparse Autoencoders (SAEs) (Cunningham et al., 2023; Gao et al., 2024; Rajamanoharan et al., 2024a) and TransCoders (Dunefsky et al., 2024) have shown promise in extracting interpretable features from MLP activations (Rudin, 2019). However, these dictionary learning methods face two fundamental limitations. First, they are inherently *post-hoc* and *proxy-based*:

the learned features exist in a reconstructed activation space rather than within the model's native parameters, inevitably introducing approximation errors (Hase & Bansal, 2020; Chanin et al., 2025). Second, training these massive sparse dictionaries is computationally expensive, often incurring costs comparable to pretraining smaller language models (Templeton et al., 2024). Researchers thus face the dual challenges of high training costs for auxiliary models and the decoupling of interpretation from the model's pretrained weights.

This motivates a fundamental research question: *Is there a natively interpretable structure within MLP weights, and can it be extracted without relying on auxiliary modules?*

To answer this question, we introduce **Native Network Anatomy** (NaNA), a training-free framework that analyzes Transformer MLPs from a spectral and functional perspective. Building on the observation of Beren & Black (2022) that singular vectors of MLP weights encode semantic structure, we advance beyond a purely representational view to formalize these spectral components as *computational operators* that directly mediate token prediction. Under the NaNA framework, an MLP projection matrix is expressed as a spectral sum of **Detector-Effector Units** (DEUs), defined as rank-1 subspaces obtained via Singular Value Decomposition (SVD). Each DEU functions as a distinct information channel formed by an outer product: a *detector*, given by the right singular vector, which selectively probes latent attributes in the input representation, and an *effector*, given by the corresponding left singular vector, which orients the resulting update in the MLP output. Crucially, this formulation induces an explicit coupling between detector activation and effector influence, enabling direct quantification of each native subspace's contribution to the residual stream.

We leverage DEUs to propose **Subspace Contribution Analysis** (SCA), a training-free method that identifies sparse, dominant pathways and measures their impact on target token predictions. Moving beyond correlational analysis, we validate the functional necessity of these pathways through targeted intervention and ablation experiments, demonstrating that a small number of top-ranked subspaces determines specific model behaviors. Furthermore, NaNA identifies intrinsic functional pathways, achieving orders-of-magnitude efficiency gains over SAEs and TransCoders. This provides a lightweight yet faithful alternative for mechanistic interpretability.

## 2. Preliminaries

This section reviews Transformer MLP sublayers and summarizes the spectral interpretation of MLP weight matrices introduced by Beren & Black (2022).

### 2.1. Transformer Architecture and MLP Sublayers

A Transformer comprises a vocabulary embedding layer followed by $L$ stacked layers, each consisting of a multi-head self-attention module $\text{Attn}^{(l)}(\cdot)$ and a position-wise MLP sublayer. The embedding matrix is $W_{\text{emb}} \in \mathbb{R}^{V \times d}$, where $V$ denotes the vocabulary size and $d$ the model dimension; $\mathbf{t}_j \in \mathbb{R}^{1 \times d}$ denotes its $j$-th row.

Given an input sequence $X = \{\mathbf{x}_1, \ldots, \mathbf{x}_M\}$, let $\mathbf{h}_m^{(l-1)}$ denote the hidden state at position $m$ entering layer $l$, initialized as $\mathbf{h}_m^{(0)} = \text{emb}(x_m) + \text{pos}(m)$. The attention sublayer produces an intermediate residual state

$$\bar{\mathbf{h}}_m^{(l)} = \mathbf{h}_m^{(l-1)} + \text{Attn}^{(l)}(\gamma(\mathbf{H}^{(l-1)}))_m, \qquad (1)$$

where $\gamma(\cdot)$ denotes layer normalization and $\mathbf{H}^{(l-1)} = (\mathbf{h}_1^{(l-1)}, \ldots, \mathbf{h}_M^{(l-1)})$ is the sequence of hidden states.

The MLP sublayer applies a two-layer feedforward transformation with residual connection, parameterized by an up-projection matrix $W_{\text{up}}^{(l)} \in \mathbb{R}^{n \times d}$ and a down-projection matrix $W_{\text{down}}^{(l)} \in \mathbb{R}^{d \times n}$, where $n$ is the MLP hidden dimension. With pointwise nonlinearity $\phi(\cdot)$, the MLP output is

$$\mathbf{h}_m^{(l)} = \bar{\mathbf{h}}_m^{(l)} + W_{\text{down}}^{(l)} \phi\Big(W_{\text{up}}^{(l)} \gamma(\bar{\mathbf{h}}_m^{(l)})\Big). \qquad (2)$$

### 2.2. Spectral Interpretation

For a matrix $W \in \mathbb{R}^{a \times b}$, its *Compact Singular Value Decomposition* (SVD) is defined as $W = U\Sigma V^\top$. Here, $r = \text{rank}(W)$, $U \in \mathbb{R}^{a \times r}$ and $V \in \mathbb{R}^{b \times r}$ are orthogonal matrices, and $\Sigma \in \mathbb{R}^{r \times r}$ is a diagonal matrix containing non-negative singular values $\sigma_1, \ldots, \sigma_r$. The $i$-th column of $U$, $\mathbf{u}_i \in \mathbb{R}^{a \times 1}$, is a *left* singular vector; analogously, columns $\mathbf{v}_i \in \mathbb{R}^{b \times 1}$ of $V$ are *right* singular vectors. Beren & Black (2022) demonstrated that the singular vectors of MLP weight matrices encode coherent semantic clusters.

**Interpreting Left Singular Vectors of $W_{\text{down}}$.** Following Beren & Black (2022), for a left singular vector $\mathbf{u}_i$ of $W_{\text{down}}$ ($i \in \{1, \ldots, d\}$, assuming $d = r$) we obtain a *score vector* $S_i \in \mathbb{R}^V$ by projecting $\mathbf{u}_i$ onto the vocabulary embedding matrix:

$$S_i = W_{\text{emb}} \mathbf{u}_i. \qquad (3)$$

The $j$-th entry of $S_i$, $s_{ij} = \mathbf{t}_j \mathbf{u}_i$, quantifies the alignment between $\mathbf{u}_i$ and the embedding of the $j$-th token.

**Interpreting Right Singular Vectors of $W_{\text{up}}$.** Analogously, any right singular vector $\mathbf{v}_i$ of $W_{\text{up}}$ gives rise to a score vector $S_i'$ by projecting the vocabulary embeddings onto $\mathbf{v}_i$:

$$S_i' = W_{\text{emb}} \mathbf{v}_i. \qquad (4)$$

Here, the scalar $s_{ij}' = \mathbf{t}_j \mathbf{v}_i$ measures the similarity between the $j$-th token embedding and the singular vector $\mathbf{v}_i$.

## 3. Native Network Anatomy Framework

We introduce the *Native Network Anatomy* (NaNA) framework for mechanistically analyzing MLP layers. We first formalize *Detector–Effector Units* (DEUs) as the elementary functional units of linear projections. We then propose *Subspace Contribution Analysis* (SCA) to quantify the predictive role of individual rank-1 components. Finally, we construct *directed DEU-based pathways* and define spectral interventions for causal analysis.

### 3.1. Detector–Effector Units (DEUs)

Let $W \in \mathbb{R}^{a \times b}$ denote a linear projection (up- or down-projection), with input $x \in \mathbb{R}^{b \times 1}$ and output $y = Wx$. By singular value decomposition, the down-projection admits

$$W_{\text{down}} = \sum_{i=1}^{r} \sigma_i \mathbf{u}_i \mathbf{v}_i^\top, \tag{5}$$

where each rank-1 term $\sigma_i \mathbf{u}_i \mathbf{v}_i^\top$ defines a *functional subspace*, i.e., a rank-1 operator coupling an input direction to an output direction.

**Detector–Effector Units.** Applying a single subspace to $x$ yields

$$(\sigma_i \mathbf{u}_i \mathbf{v}_i^\top)x = \sigma_i(\mathbf{v}_i^\top x)\,\mathbf{u}_i = h_i \mathbf{u}_i, \tag{6}$$

where $h_i = \sigma_i(\mathbf{v}_i^\top x) \in \mathbb{R}$. Thus, each subspace produces an output aligned with $\mathbf{u}_i$, with magnitude controlled by the scalar activation $h_i$.

We interpret each rank-1 operator as a *Detector–Effector Unit* (DEU): $\mathbf{v}_i$ acts as a *detector* measuring input alignment, $\mathbf{u}_i$ as an *effector* determining output orientation, and $\sigma_i$ as a *signal gain*.

**Detector and Effector Probes.** For the $i$-th DEU, we define the detector probe $f_i(x) = \mathbf{v}_i^\top x$ and effector probe $g_i(y) = \mathbf{u}_i^\top y$. These satisfy

$$g_i(y) = \sigma_i f_i(x), \tag{7}$$

indicating that the detector and effector capture a shared latent attribute before and after projection.

Following Section 2.2, we interpret DEUs by examining singular vectors that interface with the residual stream—specifically, the detector of $W_{\text{up}}$ and the effector of $W_{\text{down}}$. Due to SVD sign ambiguity (Kolda & Bader, 2009), we evaluate semantics along both polarities ($\pm\mathbf{u}_i, \pm\mathbf{v}_i$).

**MLPs as Superpositions of DEUs.** For a full projection matrix $W$ we have:

$$Wx = \sum_{i=1}^{r} (\sigma_i \mathbf{u}_i \mathbf{v}_i^\top)x = \sum_{i=1}^{r} h_i \mathbf{u}_i. \tag{8}$$

This formulation reframes the projection $W$ as a linear superposition of orthogonal DEUs, with activations $h_i = \sigma_i(\mathbf{v}_i^\top x)$ modulating each effector direction.

### 3.2. Subspace Contribution Analysis (SCA)

We now quantify the predictive role of individual DEUs via *Subspace Contribution Analysis* (SCA). Given a down-projection $W_{\text{down}} \in \mathbb{R}^{d \times n}$, input $x$, and target token embedding $\mathbf{t}^* \in \mathbb{R}^d$, SCA decomposes each DEU's contribution into detector and effector alignment.

**Detector Alignment.** We define

$$\mathbf{d} = \Sigma V^\top x \in \mathbb{R}^r, \quad d_i = \sigma_i \mathbf{v}_i^\top x, \tag{9}$$

where $d_i$ captures the input activation of the $i$-th subspace, incorporating both geometric alignment and gain. Here, $r = \text{rank}(W_{\text{down}})$ which is assumed to be $r = d$ in practice.

**Effector Alignment.** We define

$$\mathbf{e} = \mathbf{t}^* U \in \mathbb{R}^r, \quad e_i = \mathbf{t}^* \mathbf{u}_i, \tag{10}$$

which measures how strongly each effector aligns with the target embedding.

**Subspace Contribution.** A subspace contributes meaningfully only when it is both activated and aligned with the target. We therefore define

$$\mathbf{c} = \mathbf{d} \odot \mathbf{e}, \quad c_i = (\sigma_i \mathbf{v}_i^\top x)(\mathbf{t}^* \mathbf{u}_i). \tag{11}$$

When $\mathbf{t}^*$ corresponds to the unembedding vector of the target token, $c_i$ equals the signed contribution of the $i$-th DEU to the target logit. Let $\mathcal{I}_{\text{top-}k}$ denote the indices of the top-$k$ subspaces with positive contributions; these are referred to as *high-contribution subspaces*.

### 3.3. Directed DEU-based Pathway

Prior work (Geva et al., 2021; Meng et al., 2022; Paulo et al., 2025) shows that middle-to-late MLP down-projections encode distilled features. We define the *Directed DEU-based Pathway* as the collection of high-contribution subspaces across these layers.

**Selective Subspace Projection.** We conceptualize the model's internal computation as a localized information flow within modular subspaces. For each layer $l$, we decompose

$$W_{\text{down}}^{(l)} = \sum_{i=1}^{r} \sigma_i^{(l)} \mathbf{u}_i^{(l)} \mathbf{v}_i^{(l)\top}. \tag{12}$$

Let $\mathbf{z}_m^{(l)} \in \mathbb{R}^n$ denote the post-activation MLP state,

$$\mathbf{z}_m^{(l)} = \phi\left(W_{\text{up}}^{(l)} \gamma(\bar{\mathbf{h}}_m^{(l)})\right), \tag{13}$$

where $\gamma(\cdot)$ denotes normalization and $\phi(\cdot)$ the pointwise nonlinearity.

We define the *subspace filtering operator*

$$\mathcal{P}^{(l)}(\mathbf{z}_m^{(l)}) = \sum_{i \in \mathcal{I}_{\text{top-}k}^{(l)}} \sigma_i^{(l)} \mathbf{u}_i^{(l)} (\mathbf{v}_i^{(l)\top} \mathbf{z}_m^{(l)}), \tag{14}$$

which retains only the high-contribution DEUs.

**Full-Model Pathway.** To capture the global computational trajectory, we extend the layer-wise operator to a multi-layer pathway formulation. For layers $l \in [l_{\mathrm{st}}, L]$, where $l_{\mathrm{st}}$ denotes the starting layer of the pathway and $L$ is the total number of Transformer layers, the pathway-constrained state update is

$$\mathbf{h}_m^{(l)} = \mathbf{h}_m^{(l-1)} + \mathrm{Attn}^{(l)}(\gamma(\mathbf{H}^{(l-1)}))_m + \mathcal{P}^{(l)}(\mathbf{z}_m^{(l)}). \quad (15)$$

Composing these filtered updates traces a directed functional trajectory aligned with the target $\mathbf{t}^*$.

### 3.4. Subspace Interventions

We assess the role of DEUs using targeted spectral interventions. We consider three complementary intervention types: (i) reconstruction tests sufficiency by isolating a subspace set, (ii) ablation tests necessity by removing it, and (iii) scaling probes directional sensitivity while preserving the remaining spectrum.

**Top-$k$ Reconstruction.** To test sufficiency, we reconstruct the down-projection using only high-contribution subspaces:

$$W_{\mathrm{reconst}}^{(l)} = \sum_{i \in \mathcal{I}_{\mathrm{top}\text{-}k}^{(l)}} \sigma_i^{(l)} \mathbf{u}_i^{(l)} \mathbf{v}_i^{(l)\top}. \quad (16)$$

**Subspace Ablation.** To test necessity, we remove these components:

$$W_{\mathrm{ablated}}^{(l)} = W_{\mathrm{down}}^{(l)} - \sum_{i \in \mathcal{I}_{\mathrm{top}\text{-}k}^{(l)}} \sigma_i^{(l)} \mathbf{u}_i^{(l)} \mathbf{v}_i^{(l)\top}. \quad (17)$$

Performance degradation under ablation quantifies the causal importance of the identified subspaces.

**Scaling Directional Sensitivity.** To modulate an individual DEU, we apply a *relative perturbation coefficient* $\alpha$ to its rank-1 operator:

$$W_{\mathrm{interv}}^{(l)} = W_{\mathrm{down}}^{(l)} + \alpha\,(\sigma_i^{(l)} \mathbf{u}_i^{(l)} \mathbf{v}_i^{(l)\top}), \quad (18)$$

where $\alpha > 0$ amplifies and $\alpha < 0$ suppresses the subspace.

## 4. Experiments

In this section, we validate the NaNA framework[1], extending our analysis from case studies to broader systematic generalizations. Our evaluation addresses two core questions: (i) to what extent do high-contribution subspaces determine the semantic outputs of the model, and (ii) do they constitute a reliable, training-free basis for mechanistic interpretability?

---

[1]The code is available at: https://github.com/minxue29031/NaNA.

### 4.1. Experimental Setup

**Models and Datasets.** We conduct experiments across multiple model scales, including GPT-2, GPT2-Medium, GPT2-Large, and GPT2-XL, using three datasets designed to probe distinct behavioral and functional properties. ***Anchor Cases*** ($\mathcal{D}_{\mathrm{achr}}$): A set of 20 cases spanning syntax, semantics, code understanding, and factual knowledge (details in Appendix Table 8). ***Generalization Dataset*** ($\mathcal{D}_{\mathrm{gene}}$): For each model, it comprises 100 diverse prefixes paired with model-specific target tokens, used to quantify the fidelity of behavioral recovery mediated by the identified pathways. ***Controlled Variation Sets***: To assess the robustness of subspaces under contextual perturbations, we construct two datasets centered on positive-adjective semantics: (i) *Target Variance Dataset* ($\mathcal{D}_{\mathrm{target}}$), fixing the prefix and varying 100 target adjectives; and (ii) *Prefix Variance Dataset* ($\mathcal{D}_{\mathrm{prefix}}$), varying 100 prefixes while holding the target constant. (Details of the dataset construction are provided in Appendix E.1.)

**Baselines.** We evaluate NaNA against the following baselines. *SAEs* ($768 \times 24{,}576$) and *TransCoders* ($768 \times 24{,}576$) serve as representative training-based methods for decomposing polysemantic features. *Raw MLP neurons* provide a baseline to validate the role of spectral decomposition. *Random DEU subspaces* act as a stochastic control for verifying causal significance. Finally, *Full MLP* and *Full TransCoder* define the upper bound for predictive performance.

**Evaluation Metrics.** Our evaluation is based on the following four metrics. ***Fidelity***: Measures the sufficiency of subspaces or features for recovering predictive performance. We quantify the probability and rank of the original target token by reconstructing the MLP using only the selected components. ***Faithfulness***: Evaluates the necessity of subspaces or features by quantifying the resulting degradation in predictive performance. We measure the decline in the target token's probability and rank after ablating these components from the MLP. ***Interpretability***: Assesses semantic consistency. For a given concept across various prefixes and targets, we examine whether the high-contribution subspaces consistently align with the associated semantic properties. ***Efficiency***: Reports the computational overhead in terms of total runtime and peak GPU memory consumption.

**Intervention Mechanism.** We perform causal interventions on MLP projections using PyTorch hooks (Nanda & Bloom, 2022). At each target layer, we intercept the final-token activation and substitute the original output with a projection computed using a modified weight matrix, thereby propagating the intervened signal through the network.

### 4.2. Case Study on Positive Semantics

In this section, we present a case study on positive semantics ("*the cat looks very*" → "*happy*") to dissect how DEU-

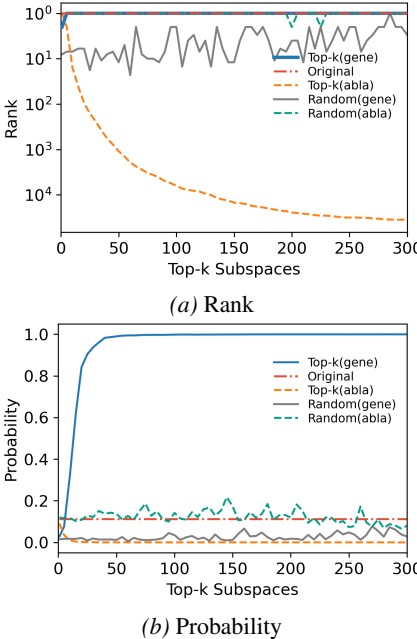

*(a)* Rank

*(b)* Probability

*Figure 2.* **Pathway Analysis via Top-$k$ Subspace Interventions in GPT2-Medium.** We evaluate the impact of high-contribution subspaces on the target token's probability (↑) and rank (↓) in the positive semantic case (" *the cat looks very*" → " *happy*").

*(a)* Standard Experiment.

*(b)* Ablation Experiment.

*Figure 3.* **Reconstruction of Middle-to-Late MLP Layers via Pathways within Top-$k$ High-Contribution Subspaces: Standard vs. Ablation.** Here, we visualize the top-15 predicted tokens of GPT2-Medium under standard and ablation interventions for the case "*The cat looks very*" → " *happy*".

based pathways encode positive semantic information across layers. (Other cases are provided in Appendix H.) Unless otherwise stated, top-$k$ subspaces are selected per sample based on SCA scores and then aggregated across samples for frequency analysis. For comparative evaluations we use subspaces that appear most frequently among per-sample top-$k$ sets.

**Impact of Pathways on Predictive Performance.** We evaluate the identified DEU-based pathways by measuring their impact on target token prediction. To assess fidelity, we reconstruct $W_{\text{down}}$ using the top-$k$ high-contribution subspaces identified by the SCA method (Top-$k$ (gene)) and randomly selected DEU subspaces (Random (gene)), with the original weights serving as a baseline. To evaluate faithfulness, we ablate the target pathways by reconstructing $W_{\text{down}}$ with the top-$k$ (Top-$k$ (abla)) or random (Random (abla)) DEU subspaces removed. As shown in Figure 2, we intervene on the MLP down-projection matrices across the middle-to-late layers (7–23) of GPT2-Medium, monitoring the resulting fluctuations in target token rank (↓) and probability (↑). Here, rank denotes the target token's position in the model's output distribution.

As illustrated in Figure 2, we observe a distinct phase transition in the model's behavior. The Top-$k$ (Gene) trajectory reveals that the model's predictive capability can be almost entirely recovered using only a small fraction of subspaces (e.g., $k < 20$), with the target token's rank quickly improving and the probability saturating near 1.0. This result suggests that these semantic pathways are intrinsically

sparse. Conversely, the Top-$k$ (Abla) curves exhibit a sharp performance collapse, where ablating the top-$k$ subspaces triggers an exponential decay in rank. The flat trajectories of the random baselines further confirm that the dominant subspaces identified by SCA are causally necessary for the target prediction. In addition, we extend this analysis to the GPT-2 family, demonstrating that this sparsity pattern remains consistent across parameter scales (see Appendix I.1).

**Causal Evidence of Pathway-Driven Predictions.** To provide direct causal evidence that SCA-identified pathways drive model predictions, we visualize the evolution of the model's output distribution across varying pathway widths, defined by the top-$k$ subspaces. Figure 3 illustrates the top-15 predicted tokens for the prompt "*The cat looks very*" as we reconstruct the middle-to-late MLPs of GPT2-Medium using an increasing number of high-contribution subspaces.

*Semantic Recovery (Fidelity).* As shown in Figure 3a, reconstructing with only the top-5 subspaces is sufficient to restore the target token "*happy*". As more subspaces are included, the top-15 predictions increasingly focus on positive adjectives such as "*pleased*", "*comfortable*", and "*good*". This rapid recovery demonstrates that a small number of high-contribution DEU subspaces play a dominant role in driving the model's behavior.

*Semantic Collapse via Ablation (Faithfulness).* Conversely, Figure 3b reveals a dramatic semantic collapse upon targeted ablation. The removal of merely the top 10 subspaces extinguishes the target token "*happy*" from the top-tier pre-

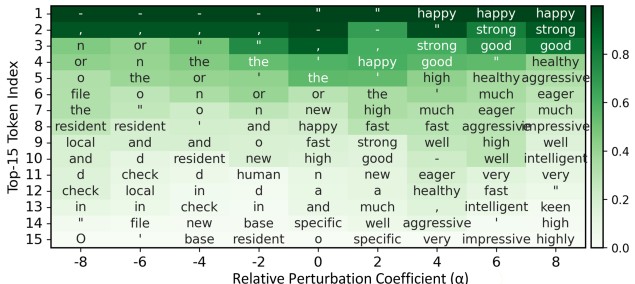

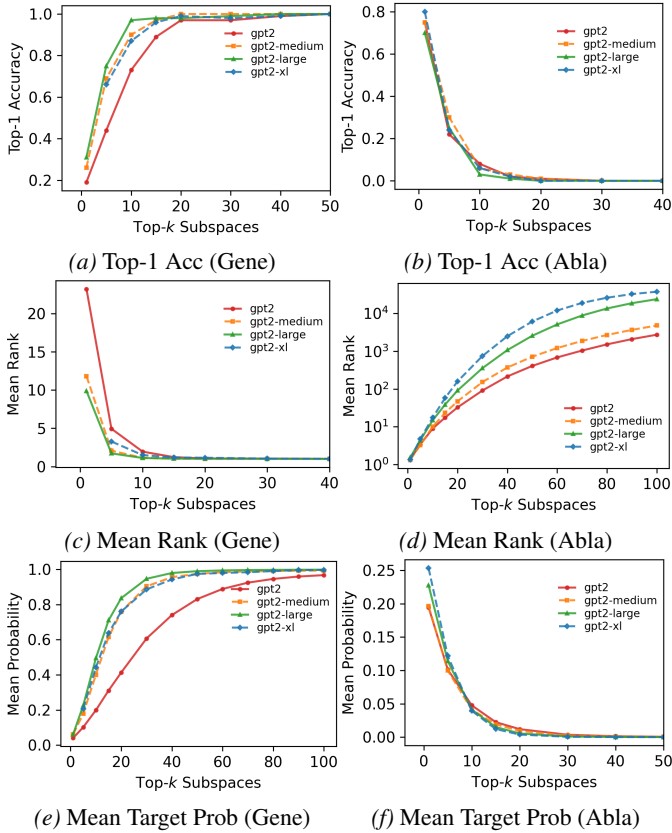

*Figure 4.* **Model Lexical Preferences under Subspace Intervention.** Using the Logit Lens, we project modified layer outputs onto the vocabulary. We show the evolution of the top-15 tokens as the relative perturbation coefficient $\alpha$ varies from suppression ($\alpha < 0$) to amplification ($\alpha > 0$).

dictions. As more high-contribution subspaces are systematically ablated, the model loses the ability to generate positive adjectives and instead produces generic terms such as "*much*", "*similar*", and "*different*". This sharp divergence confirms that high-contribution DEUs constitute the primary functional modules for the target prediction. (Detailed results for 20 diverse cases across models are provided in Appendix H.)

**Subspace Interventions.** To investigate the functional role of individual subspaces, we employ an intervention mechanism modulated by the relative perturbation coefficient $\alpha$. In this setup, $\alpha = 0$ implies no intervention, while $\alpha > 0$ and $\alpha < 0$ correspond to semantic amplification and suppression, respectively. We track the resulting shifts in the predictive distribution by projecting the intervened states onto the vocabulary via the Logit Lens (Belrose et al., 2023). In Figure 4, we visualize the effects of manipulating Subspace 7 of Layer 16 in GPT2-Medium for the prompt "*The cat looks very*", where Subspace 7 encodes the semantics of "*positive qualities*".

*Amplifying Target Semantic Contribution ($\alpha > 0$).* Increasing the intervention coefficient from 0 to 8 yields a progressive dominance of Subspace 7. Compared to the baseline ($\alpha = 0$), amplifying Subspace 7 forces the model to prioritize tokens such as "*happy*", "*strong*", and "*healthy*". Notably, at high amplification levels, the layer's output semantics converge almost exclusively toward positive attributes. These results confirm that manipulating Subspace 7 can steer the model's lexical preferences toward specific semantic concepts.

*Suppressing Target Semantic Contribution ($\alpha < 0$).* As $\alpha$ drops below zero, the top tokens degenerate into irrelevant nouns (e.g., "*file*", "*resident*") or generic function words (e.g., "*the*", "*or*"). This functional collapse suggests that Subspace 7 is a primary carrier of positive quality attributes. Its suppression effectively divests the latent representation of its semantic directionality, driving the output away from positive descriptors.

*(a)* Top-1 Acc (Gene)

*(b)* Top-1 Acc (Abla)

*(c)* Mean Rank (Gene)

*(d)* Mean Rank (Abla)

*(e)* Mean Target Prob (Gene)

*(f)* Mean Target Prob (Abla)

*Figure 5.* **Evaluating the Fidelity and Faithfulness of Pathways. Left:** Semantic recovery via top-$k$ subspace reconstruction. **Right:** Semantic degradation via top-$k$ subspace ablation. Metrics include mean probability ($\uparrow$), mean rank ($\downarrow$), and top-1 accuracy ($\uparrow$) of target tokens across all samples.

### 4.3. Generalization Evaluation

**Pathway Generalization Evaluation.** To assess the fidelity and faithfulness of the identified pathways, we conduct a generalization evaluation across the GPT-2 variants using $\mathcal{D}_{\text{gene}}$, including both standard and ablation experiments. We quantify behavioral recovery by computing the mean probability, rank, and top-1 accuracy of target tokens across all samples. Here, top-1 accuracy reflects the model's ability to correctly predict the ground-truth token.

*Standard Experiment (Fidelity):* As illustrated in Figures 5a, 5c, and 5e, reconstructing the middle-to-late layers using pathways composed of only the top-20 subspaces yields near-optimal performance. Across all model scales, the mean rank converges toward the optimum, and top-1 accuracy reaches a remarkable 97-99%. Furthermore, the mean probability increases monotonically as additional subspaces are incorporated, confirming that SCA identifies the key subspaces driving the target-directed predictions.

*Ablation Experiment (Faithfulness):* The ablation results presented in Figures 5b, 5d, and 5f establish the causal necessity of the identified pathways. Specifically, removing

*Table 1.* Layer-wise Interpretation of Subspaces in GPT2-Medium (Layers 14–17). The "Positive" and "Negative" columns denote semantic concepts corresponding to the opposing directions of the singular vector ($\pm \mathbf{u}_i$), accounting for the inherent sign ambiguity in SVD (Kolda & Bader, 2009). Here, parentheses denote subspace interpretability levels (High, Med, or Low).

| ID | Positive Direction | Negative Direction | Y/N |
|---|---|---|---|
| L14_SP13 | *null* (null) | vibrant leisure (High) | Y |
| L14_SP2 | *null* (null) | Punctuation and Symbols (High) | N |
| L14_SP74 | Government Challenges (Med) | *null* (null) | N |
| L14_SP3 | electricity concepts (Med) | mental states (Med) | Y |
| L14_SP10 | Positive Excellence (High) | Measurement and Time (Med) | Y |
| L15_SP14 | updates and terminations (High) | emotional experiences (High) | Y |
| L16_SP7 | Government Deaths (High) | positive qualities (High) | Y |
| L16_SP12 | Respect and Worthiness (High) | Numbers and Quantities (High) | Y |
| L16_SP2 | Punctuation Marks (High) | *null* (null) | N |
| L16_SP3 | Observation and Critique (Med) | electric devices (High) | Y |
| L17_SP14 | Cognitive Values (High) | Security incidents (High) | Y |

*Figure 6.* **Top-10 Active Subspaces in GPT2-Medium (Layers 14–17).** This visualizes the activation frequency across target-variance and prefix-variance datasets. Here, color intensity reflects activation frequency, and SP$i$ denotes subspace $i$.

the high-contribution DEU subspaces leads to a catastrophic drop in mean top-1 accuracy and in probability, and an exponential increase in mean rank. This degradation confirms that a marginal fraction of DEU subspaces dominates the model's predictive behavior, and their removal effectively severs the underlying semantic pathway.

**Highly Active Subspaces and Their Semantic Roles.** To investigate whether certain subspaces are consistently highly active and serve as interpretable mechanisms for processing specific contextual semantics, we utilize the *Controlled Variation Sets*. These sets consist of a target-variance dataset $\mathcal{D}_{\text{target}}$ and a prefix-variance dataset $\mathcal{D}_{\text{prefix}}$, both focused on positive-adjective semantics. As subspaces derived from $\mathcal{D}_{\text{target}}$ capture intrinsic semantic structures more robustly than the noise-prone $\mathcal{D}_{\text{prefix}}$ setting, we use the former for primary analysis and the latter for validation.

Figure 6 illustrates the top-10 active subspaces in Layers 14-17 of GPT2-Medium, identified by their recurrence frequency among the top-10 contributors across all positive semantic samples. Here, color intensity represents activation frequency, and SP$i$ denotes subspace $i$. We find that only a small subset of subspaces are consistently active within each layer. This suggests that a limited number of subspaces act as principal carriers of semantic information, playing a pivotal role in mediating specific semantic features.

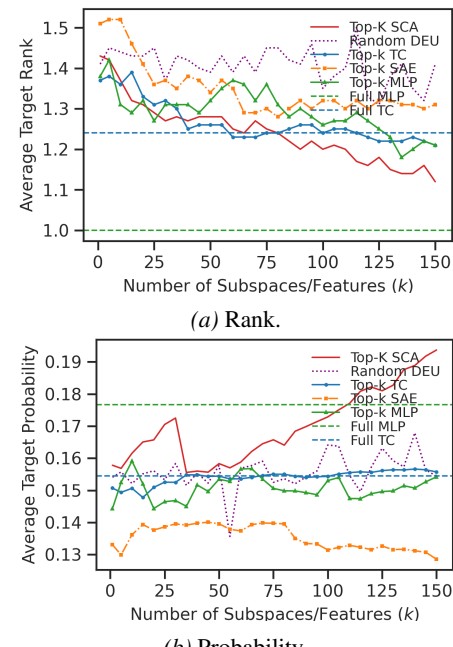

*(a)* Rank.

*(b)* Probability.

*Figure 7.* **Fidelity Assessment of Active Components in GPT-2 (Layer 10).** Baselines include original MLP neurons (Top-$k$ MLP), TransCoder features (Top-$k$ TC), SAE features (Top-$k$ SAE), and Random DEU. Full MLP and Full TC denote upper bounds.

We further examine whether these highly active subspaces exhibit direct and interpretable associations with positive semantic attributes. Table 1 details the semantic interpretations of these subspaces under $\mathcal{D}_{\text{target}}$, where L$m$_SP$i$ denotes subspace $i$ in layer $m$. Our analysis shows that these high-frequency subspaces are typically related to positive semantics. For example, in Layer 14, Subspace 13 encodes the semantic of "*Vibrant Leisure*" and Subspace 10 relates to "*Positive Excellence*". Similarly, in Layer 16, Subspace 7 encodes "*Positive Qualities*", Subspace 12 represents "*Respect and Worthiness*", and Subspace 3 captures "*Observation and Critique*". These findings suggest that the model's predictions are mediated by specific, functional modules rather than diffuse representations. Collectively, these subspaces drive the robust generation of positive semantics. (Appendix F.1 provides further layer-wise statistics and interpretations across various models.)

### 4.4. Comparative Evaluation

**Comparison with Neuronal and Dictionary-Based Methods.** We utilize the generalization dataset ($\mathcal{D}_{\text{gene}}$) to evaluate the reconstruction fidelity of the top-$k$ active components in recovering the target token's prediction within GPT-2 (Layer 10). We compare SCA against four representative baselines: original MLP neurons, TransCoder features, SAE features, and random DEU subspaces, where active features are selected based on activation variance. The *Full MLP* and *Full TC* settings serve as performance upper bounds. For

*Table 2.* **Computational Efficiency of Full-Subspace and Feature Interpretation on a Single Projection Matrix in GPT-2 (NVIDIA A100, 80GB).** Here, the runtime of SAE and TransCoder includes external module training.

|  | NaNA | SAE | TransCoder |
|---|---|---|---|
| **Runtime** | < 2 min | ≈ 30 min | ≈ 30 min |
| **Training & Tuning** | Not needed | Needed | Needed |
| **Training Data** | Not needed | 60M tokens | 60M tokens |
| **GPU Memory** | < 1 GB | ≈ 16 GB | ≈ 16 GB |

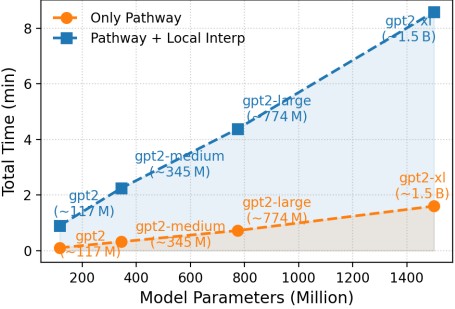

*Figure 8.* **Linear Scaling of the NaNA Framework Across Model Sizes.** Here, runtimes are measured on a single A100 (80GB) GPU for pathway extraction and local interpretation using the top-20 subspaces.

SCA, the active subspace is derived from high-frequency components shared across samples.

Figure 7 evaluates reconstruction performance based on Average Target Rank (↓) and Average Target Probability (↑), demonstrating that SCA consistently outperforms baseline methods on both metrics. Specifically, as illustrated in Figure 7a, while SAE and TransCoder exhibit performance saturation beyond $k > 60$, SCA continues to achieve performance gains.

We attribute this sustained gain to the explicit filtering for *causal bridges*, defined as subspaces that satisfy both input activation (detector alignment) and output relevance (effector alignment). This directed extraction enables the capture of implicit supporters, characterized as subspaces with low activation magnitude yet significant directional influence, which are typically neglected by unsupervised dictionary learning methods.

In terms of stability, SCA exhibits robust and monotonic rank reduction, avoiding the high volatility observed in MLP and Random DEU baselines caused by target-irrelevant semantic interference. Notably, Figure 7b shows a transient probability dip around $k = 30$. We ascribe this to Softmax competition arising from semantic broadening (Holtzman et al., 2020): subspaces at this stage amplify a cluster of related synonyms, temporarily diluting the specific target's probability mass (as shown in Figure 3).

**Computational Efficiency and Resource Overhead.** In this section, we evaluate the computational efficiency and scalability of the NaNA framework in comparison to SAEs and Transcoders.

*Runtime and GPU Usage on Single-Layer.* Table 2 contrasts the computational resource requirements of the NaNA framework against training-based methods for analyzing a single projection matrix in GPT-2. Notably, NaNA is training-free and highly efficient, completing the analysis in under 2 minutes with negligible memory overhead (<1GB) on an NVIDIA A100 (80GB). In sharp contrast, SAEs and TransCoders necessitate training auxiliary modules on approximately 60 million tokens, consuming around 30 minutes and 16GB of memory (Dunefsky et al., 2024). Furthermore, these training-based methods typically require extensive hyperparameter tuning, which substantially amplifies the total computational cost.

*On-Demand Pathway Analysis.* In training-based paradigms like SAEs and TransCoders, pathway extraction necessitates the computationally intensive pre-computation of global auxiliary modules. Conversely, the NaNA framework facilitates fast, training-free pathway extraction and on-demand analysis by selectively decoding only the top-ranked subspaces. Empirically, pre-computing all modules for GPT2-XL with SAEs or TransCoders takes over 24 hours on a single A100 (80GB) GPU (Dunefsky et al., 2024). By comparison, as shown in Figure 8, our framework completes pathway extraction and local interpretation in only 8.59 minutes using the top-20 subspaces (which, as shown in Figure 5, are sufficient to preserve target predictions). Importantly, NaNA exhibits linear scalability with respect to model size. The runtime for both pathway extraction and local interpretation grows linearly rather than exponentially, ensuring the framework remains computationally efficient as model size increases. This property allows NaNA to overcome the scalability and computational bottlenecks inherent in training-based interpretability methods.

## 5. Conclusion

In this work, we introduce the Native Network Anatomy (NaNA) framework, which bridges the gap between the linear algebraic structure of weights and the semantic behavior of MLPs. By integrating Detector-Effector Units (DEUs) with Subspace Contribution Analysis (SCA), we demonstrate that model predictions are driven by sparse DEU-based pathways. Crucially, NaNA achieves orders-of-magnitude improvements in efficiency without compromising fidelity, circumventing the approximation bias inherent in proxy-based interpretation methods.

NaNA's efficacy hinges on SCA's explicit targeting of "causal bridges"—subspaces that satisfy the dual constraints of input activation (detector alignment) and output relevance (effector alignment). This focus on functional hubs allows the framework to capture "implicit supporters": low-magnitude components that exert significant directional influence but are typically discarded by unsupervised dictio-

nary learning (Trenton et al., 2023; Dunefsky et al., 2024). Ultimately, NaNA establishes a scalable, high-fidelity foundation for interpreting the internal logic of large-scale models through their native structural geometry.

## Impact Statement

This paper introduces a training-free framework for mechanistic interpretability, dissecting MLPs through their native computational structures. We position this work as a foundational step; its broader impact is expected to manifest through downstream applications of pathway analysis, ultimately contributing to the improved understanding, controllability, and reliability of Large Language Models (LLMs).

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

# APPENDIX

## Appendix Contents

## A. Related Work

**Sparse Dictionary Learning.** To address the challenge of polysemanticity in neural representations (Elhage et al., 2022), sparse dictionary learning (Tang et al., 2026; Kreutz-Delgado et al., 2003) has emerged as a dominant paradigm for disentangling superposed features (Nanda et al., 2023; Templeton et al., 2024; Elhage et al., 2021). Pioneering work by Cunningham et al. (2023) demonstrated that training sparse autoencoders (SAEs) on model activations can effectively decompose superposition into semantically interpretable feature directions (Arora et al., 2018). Building on this foundation, Trenton et al. (2023) established the scalability of SAEs for extracting interpretable features from large-scale models, enabling systematic circuit-level analyses. Their study further provided a detailed characterization of SAE scaling behavior, along with a taxonomy of learned features across model sizes. Subsequent research has introduced architectural refinements to mitigate specific limitations of standard SAEs. For instance, Rajamanoharan et al. (2024a) addressed feature shrinkage by decoupling direction learning from magnitude estimation, applying $L_1$ regularization exclusively to a gating mechanism to reduce undesirable suppression effects. In parallel, Gao et al. (2024) proposed $k$-sparse autoencoders, which impose a hard $k$-sparsity constraint to explicitly control the sparsity budget (Makhzani & Frey, 2014). Beyond reconstruction-based approaches, Dunefsky et al. (2024) introduced the TransCoder, which learns a direct mapping from MLP inputs to outputs to simulate layer-level functionality (Gao et al., 2024; Paulo et al., 2025). Despite these advancements, existing approaches largely rely on *post-hoc proxies*. These auxiliary models require substantial computational resources for training and

inevitably introduce reconstruction errors, potentially leading to deviations from the original network's true behavior.

**Linear Representations and Mechanistic Interpretability of MLPs.** Our work is grounded in the Linear Representation Hypothesis (Park et al., 2024), which posits that semantic concepts are encoded as approximately linear directions in activation space. This linearity assumption provides a principled basis for applying spectral methods to decompose model weights. Within Transformer Feed-Forward Networks (FFNs), Geva et al. (2021) conceptualized each MLP layer as a Key–Value Memory, where the first linear transformation acts as a "key" to detect input patterns, and the second as a "value" to project features into the output space. Building on this view, Geva et al. (2022) further demonstrated that FFNs operate by selectively promoting specific concepts toward the output space. To empirically validate such hypotheses, the field has extensively relied on activation patching and causal tracing to localize and attribute model behaviors (Syed et al., 2023; Wang et al., 2022; Dumas et al., 2025). Notably, Meng et al. (2022) leveraged causal tracing to pinpoint factual knowledge storage, introducing Rank-One Model Editing (ROME) to manipulate down-projection weights (Meng et al., 2023; Li et al., 2024). Similarly, Conmy et al. (2023) proposed automated circuit discovery via recursive patching techniques (Heimersheim & Nanda, 2024). In a separate line of work, Beren & Black (2022) observed that the singular vectors of Transformer weight matrices admit meaningful semantic interpretations (Praggastis et al., 2022). Nevertheless, such spectral analyses have largely remained qualitative or static, lacking a rigorous causal framework for systematically linking individual spectral components to concrete model behaviors. Along this line of work, Ahmad et al. (2026) proposed learning sparse continuous masks over singular directions to identify the key singular vectors that supported a model's performance on specific tasks. However, this method relies on dataset-specific training optimization and remains limited in enabling targeted, fine-grained attribution and interpretability at the single-sample (prompt–target) level.

While Sparse Autoencoders (Cunningham et al., 2023) and TransCoders (Dunefsky et al., 2024) effectively disentangle polysemantic representations, they act as computationally expensive proxies that approximate, rather than faithfully reflect, the underlying model. Conversely, classical linear analyses (Beren & Black, 2022; Praggastis et al., 2022) avoid proxy training but offer only static interpretations, lacking the causal machinery required to identify functionally relevant pathways. Our work bridges this methodological gap by introducing the **Native Network Anatomy** framework (NaNA). Unlike prior spectral approaches, NaNA extends static weight inspection to dynamic causal attribution. Crucially, unlike learned dictionary methods, it identifies sparse, dominant computational pathways directly from the native spectral decomposition of pre-trained weights, achieving exact functional fidelity without additional training.

## B. Discussion and Limitations

While our proposed training-free framework, NaNA, demonstrates significant advantages in both efficiency and fidelity, we acknowledge certain limitations that delineate avenues for future research.

**Architectural Scope and Applicability.** Our empirical validation is currently focused on the GPT-2 family. The generalizability of NaNA to contemporary LLM architectures with distinct structural paradigms, such as Mixture-of-Experts (MoE) models (Fedus et al., 2022; Jiang et al., 2024), remains an open question. Furthermore, our analysis is strictly confined to MLP layers. Although MLPs constitute the majority of model parameters, attention heads play a pivotal role in information routing and context integration. Extending the spectral decomposition methodology to encompass attention mechanisms represents a critical direction for future work.

**Prediction Paradigm and Task Scope.** Consistent with prior work on TransCoders and SAEs, our current analysis is primarily grounded in the next-token prediction paradigm. Although this objective underlies the fundamental training of LLMs, interpretations based solely on single-step predictions may not fully capture the models' macroscopic behavior in complex tasks, such as multi-step reasoning or long-text generation. A key open question is how the local computational pathways identified by NaNA compose to support higher-level capabilities, including Chain-of-Thought reasoning (Wei et al., 2022).

## C. Threats to Validity

We acknowledge that the structural heterogeneity between DEU subspaces and baseline feature representations (i.e., SAEs and TransCoders) presents a potential challenge to comparative validity. Baseline methods typically rely on a high-dimensional, overcomplete dictionary, whereas the NaNA framework derives orthogonal DEU subspaces directly from pretrained model weights. This structural disparity implies that neither the intrinsic nature nor the information density per unit is inherently equivalent. Following Dunefsky et al. (2024), we mitigate this concern by imposing a strict sparsity

constraint on the number of active components. This ensures a fair comparison of reconstruction efficiency by focusing on the number of distinct computational units required to recover the target.

A second limitation stems from our reliance on an external LLM for automated subspace interpretation. We recognize that the validity of such assessments is inherently contingent upon the semantic reasoning capabilities and potential biases of the evaluator model. To ensure reproducibility and mitigate variability, we provide the full model configurations and exact prompt templates used in our experiments, facilitating transparent replication and verification of our results.

## D. Why Rank-1 SVD Components Define Functional Subspaces

### D.1. Theoretical properties distinguishing an SVD-induced basis from an arbitrary orthogonal basis

Following the optimal low-rank approximation (Eckart–Young theorem) (Eckart & Young, 1936), among all rank-k decompositions, the top-k SVD terms minimise the approximation error to the full weight matrix under the Frobenius norm (or any unitarily invariant norm). This means that each additional rank-1 SVD term captures the maximum possible remaining influence on the matrix's action. For example, a rank-1 component $u_i \sigma_i v_i^\top$ represents the direction in the input space ($v_i$) that undergoes the maximum possible amplification ($\sigma_i$) into the output space ($u_i$). A random orthogonal basis enjoys no such ordering: its components do not correspond to any notion of functional contribution, and there is no principled way to rank or select among them. In more informal terms, because the SVD isolates the directions of maximum variance, these components represent the most "dominant" pathways through the layer. This is one of the aspects making this basis special.

Further justification comes from the diagonalization property of the SVD. By construction, $W = U\Sigma V^\top$ maps each input direction $v_i$ exclusively onto the output direction $u_i$, scaled by $\sigma_i$, with zero coupling to any other output direction $u_j, j \neq i$. This is not true of an arbitrary orthogonal decomposition, where cross-terms between different basis directions generally remain. As a consequence, each rank-1 SVD term represents a linearly decoupled input–output channel: activating detector $v_i$ produces a response only in effector $u_i$, not in any other effector. It is in this precise sense, and not merely because the basis vectors happen to be orthogonal as is true of any orthonormal basis, that SVD-induced Detector-Effector units operate with minimal interference between one another, representing the linearly decoupled functional streams the model has learned (Nayak et al., 2026; Gargiulo et al., 2025).

Another indication (yet not a rigid proof) that the SVD basis is privileged comes from studies on the learning dynamics of gradient descent (e.g., Saxe et al. (2013)). This work demonstrates that for deep linear neural networks, gradient descent tends to learn independent input-output mappings sequentially, exactly along the singular vectors of the task's structural covariance. The network does not learn in an arbitrary orthogonal basis; it explicitly builds its weights by incrementally adding capacity along the principal directions of the data manifold. This provides an existence proof that SVD-aligned structure can emerge from gradient-based learning, rather than being an arbitrary post-hoc decomposition.

### D.2. Empirical validation

We provide a series of evidence showing that high-contribution SVD components behave as functional subspaces, rather than merely convenient basis vectors:

*Dominance in token prediction.* If SVD components were merely a basis artefact, one would expect that predicting a specific token would require a diffuse mixture of many components, and that the identity of the dominant components would vary arbitrarily across prompts. Instead, Sections 4.2 and 4.3 show that a small, consistent set of high-contribution rank-1 terms suffices to dominate target-token prediction, while ablating them causes a marked drop. This selectivity, with specific components for specific predictions, is the empirical signature of functional separability.

*Cross-prompt robustness.* Section 4.3 (Experiment 2) and Appendix F use Controlled Variation Sets to show that the same subspaces recur as dominant contributors across prompts sharing a common semantic target (e.g., positive-adjective semantics), even when surface form and prefix vary. An arbitrary orthogonal basis would not exhibit this consistency; the fact that SVD components do suggests they track something functionally real in the network's computation.

# E. Experimental Setup: Datasets, Models, and Prompts

## E.1. Datasets.

This section details the experimental datasets constructed to evaluate model behavior and pathway generalization. We categorize our data into three primary subsets: **Anchor Cases** for broad linguistic coverage, **Controlled Variation Sets** for isolating semantic variables, and a **Generalization Dataset** for systematic assessment. All datasets are provided in the *Supplementary Material*.

- **Anchor Cases ($\mathcal{D}_{\mathbf{achr}}$).** To facilitate in-depth probing, we selected 20 representative instances spanning diverse functional capabilities, including syntactic dependency, semantic completion, code generation, and factual retrieval. These cases are designed to isolate and observe distinct mechanisms within the model. The complete list is detailed in Table 8.

- **Generalization Dataset ($\mathcal{D}_{\mathbf{gene}}$).** We constructed $\mathcal{D}_{\mathrm{gene}}$ to examine the behavioral generalization of the pathway across diverse contexts in positive-adjective semantics. For each model $m$ in the GPT-2 family, we generated 100 diverse prefixes $c_i$. To ensure alignment with the model's native distribution, the target token $t_i^{(m)}$ for each instance is defined as the model's original top-1 prediction. This yields a model-specific dataset $\mathcal{D}_{\mathrm{gene}}^{(m)} = \{(c_i, t_i^{(m)})\}$ tailored to the predictive behavior of each variant.

- **Controlled Variation Sets.** To assess the robustness of subspaces under contextual perturbations, we construct two complementary datasets focused on positive-adjective semantics:

  - *Target Variance Dataset ($\mathcal{D}_{target}$).* The target token is varied while the prefix is held fixed. Specifically, we use a fixed prefix $c_{\mathrm{fix}} =$ "*The cat looks very*" paired with a set of 100 distinct positive adjectives $T = \{t_1, t_2, \ldots, t_{100}\}$ (e.g., " *happy*", " *sunny*", " *glad*", " *calm*"). Formally, the dataset is defined as $\mathcal{D}_{\mathrm{target}} = \{(c_{\mathrm{fix}}, t_i) \mid t_i \in T\}$.
  - *Prefix Variance Dataset ($\mathcal{D}_{prefix}$).* The prefix is varied while the target token is held constant. We generate 100 diverse prefixes $C = \{c_1, c_2, \ldots, c_{100}\}$ (e.g., *"The dancer appears very", "The baby seems truly"*) paired with a fixed target token $t_{\mathrm{fix}} =$ " *happy*". Formally, the dataset is defined as $\mathcal{D}_{\mathrm{prefix}} = \{(c_i, t_{\mathrm{fix}}) \mid c_i \in C\}$.

## E.2. Models and Prompt.

This section describes the models and prompts used in our experiments. A complete list of evaluated models and their variants is provided in Table 3. Furthermore, the prompt and model employed for the subspace interpretability analysis are illustrated in Figure 9. All experiments were executed on a single **NVIDIA A100 GPU (80GB)**.

*Table 3.* **Summary of the GPT-2 Family, TransCoders, and SAEs Used in Experiments.**

| Model | Hidden Size | MLP Dim | Layers | Access Link | Reference |
|---|---|---|---|---|---|
| GPT-2 | 768 | 3072 | 12 | openai-community/gpt2 | Radford et al. (2019) |
| GPT2-Medium | 1024 | 4096 | 24 | openai-community/gpt2-medium | Radford et al. (2019) |
| GPT2-Large | 1280 | 5120 | 36 | openai-community/gpt2-large | Radford et al. (2019) |
| GPT2-XL | 1600 | 6400 | 48 | openai-community/gpt2-xl | Radford et al. (2019) |
| GPT-Neo | 2560 | 10240 | 32 | EleutherAI/gpt-neo-2.7B | Gao et al. (2020) |
| GPT-J | 4096 | 16384 | 28 | EleutherAI/gpt-j-6b | Wang & Komatsuzaki (2021) |
| TransCoder | 768 | 24576 | - | pchlenski/gpt2-transcoders | Dunefsky et al. (2024) |
| SAE | 768 | 24576 | - | tommmcgrath/gpt2-small-mlp-out-saes | Cunningham et al. (2023); Bloom et al. (2024) |
| JumpReLU SAE | 768 | 24576 | - | Codebase | Rajamanoharan et al. (2024b) |
| Top-$k$ SAE | 768 | 24576 | - | Codebase | Gao et al. (2025) |
| BatchTopK SAE | 768 | 24576 | - | Codebase | Bussmann et al. (2024) |

---

[2]https://platform.openai.com/docs/models/compare?model=gpt-4o-mini

---

**SYSTEM PROMPT TEMPLATE**

**Role:** You are an expert in neural network interpretability. Reply strictly in JSON format.

---

**User Instruction:** Analyze the semantic consistency of the following direction.

**Input Data:**
- **Direction ID:** `<ID>(<Polarity>)`
- **Top Tokens:** `[<Token List>]`

**Analysis Criteria:** My goal is to identify directions that represent a specific semantic meanings or grammatical functions (concepts, topics, or distinct grammatical roles).

1. **Concept Label:**
   - Summarize the common meaning using 2-4 English words.
   - Do NOT use labels like "Subwords" or "Prefixes" – these are structural, not semantic. If no semantic theme exists, label it *null*.
2. **Consistency Score:** Assign strictly based on MEANING, not spelling.
   - **High:** Tokens clearly share a specific concept.
   - **Medium:** Tokens are thematically related but broad.
   - **Low:** Tokens have totally unrelated meanings. If tokens share structure (e.g., all start with '#') but have unrelated meanings, score as *low*.

**Output Format:**
   {{ "label": "Your concept label" or null, "score": "High" / "Medium" / "Low" or null }}

---

| Layer | Direction | Top Activating Tokens | Label | Score |
|-------|-----------|----------------------|-------|-------|
| **L26_SP15** | **Positive** | ['compliant', 'qualified', 'certified', 'suitable', 'suited', 'wisely', 'capable', 'preferred', 'licensed', 'ideal', 'recommended', 'approved', 'proficient', 'Certified', 'worthy', 'specialize', 'authorized', '#Works', 'competent', 'compatible'] | **qualified approval** | **High** |
| | **Negative** | ['disruptions', 'disturbances', 'outbreaks', 'earthquakes', 'storms', 'glitches', 'crashes', 'collisions', 'riots', 'explosions', 'casualties', 'occurrences', 'spills', 'turbulence', 'floods', 'surges', '#effects', 'scandals', 'leakage', 'wildfires'] | **natural disasters** | **High** |

*Figure 9.* **Prompt Design for Automated Interpretation.** For example, we illustrate the interpretation of Subspace 15 in Layer 26 of $W_{\text{down}}$ (GPT2-XL). The "Positive" and "Negative" directions correspond to the semantic concepts aligned with the opposing directions of the singular vector ($\pm\mathbf{u}_i$), reflecting the inherent sign ambiguity in SVD (Kolda & Bader, 2009). Here, *gpt-4o-mini*[2] is used to provide semantic interpretations for the subspaces.

## F. Layer-wise Analysis of Highly Active DEU Subspaces

In this section, we examine the layer-wise highly active subspaces in the GPT-2 family. This analysis assesses whether specific subspaces contribute consistently and robustly to target semantic features, and further examines whether these highly active subspaces exhibit direct and interpretable associations with particular semantic scenarios. Because subspaces identified under the target variance setting ($\mathcal{D}_{\text{target}}$) more directly reflect intrinsic semantic structure, whereas the prefix variance setting ($\mathcal{D}_{\text{prefix}}$) is comparatively more susceptible to contextual noise, we use the former for primary analysis and the latter for validation.

### F.1. Consistently Activated Subspaces

Figures 10-13 illustrate the ten most active subspaces, defined as those that most frequently appear among the top-10 contributors across both target token variants ($\mathcal{D}_{\text{target}}$) and prefix variants ($\mathcal{D}_{\text{prefix}}$). We observe that only a small subset of subspaces appears with high frequency within each layer, indicating that semantic contributions are highly concentrated rather than uniformly distributed, and that these subspaces function as dominant semantic carriers. This suggests that a limited number of subspaces play a disproportionately central and stable role in mediating specific semantic features.

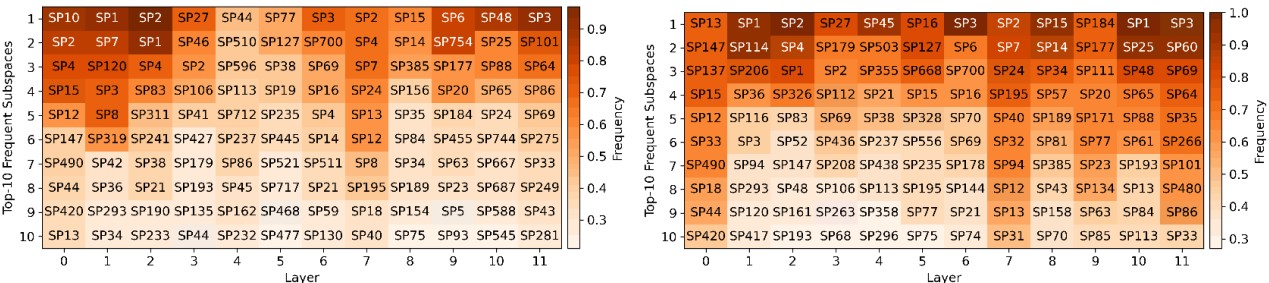

*(a)* Top-10 contributors across 100 target token variants ($\mathcal{D}_{\text{target}}$).      *(b)* Top-10 contributors across 100 prefix variants ($\mathcal{D}_{\text{prefix}}$).

*Figure 10.* **Layer-wise Consistency of Top-10 Subspace Activations Across Target Token Variants ($\mathcal{D}_{\text{target}}$) and Prefix Variants ($\mathcal{D}_{\text{prefix}}$) in GPT-2.** The heatmaps illustrate, for each layer, the frequency with which each subspace appears among the top-10 contributors across all variants, thereby identifying subspaces that are consistently activated across diverse target tokens. Here, SP$i$ denotes subspace $i$.

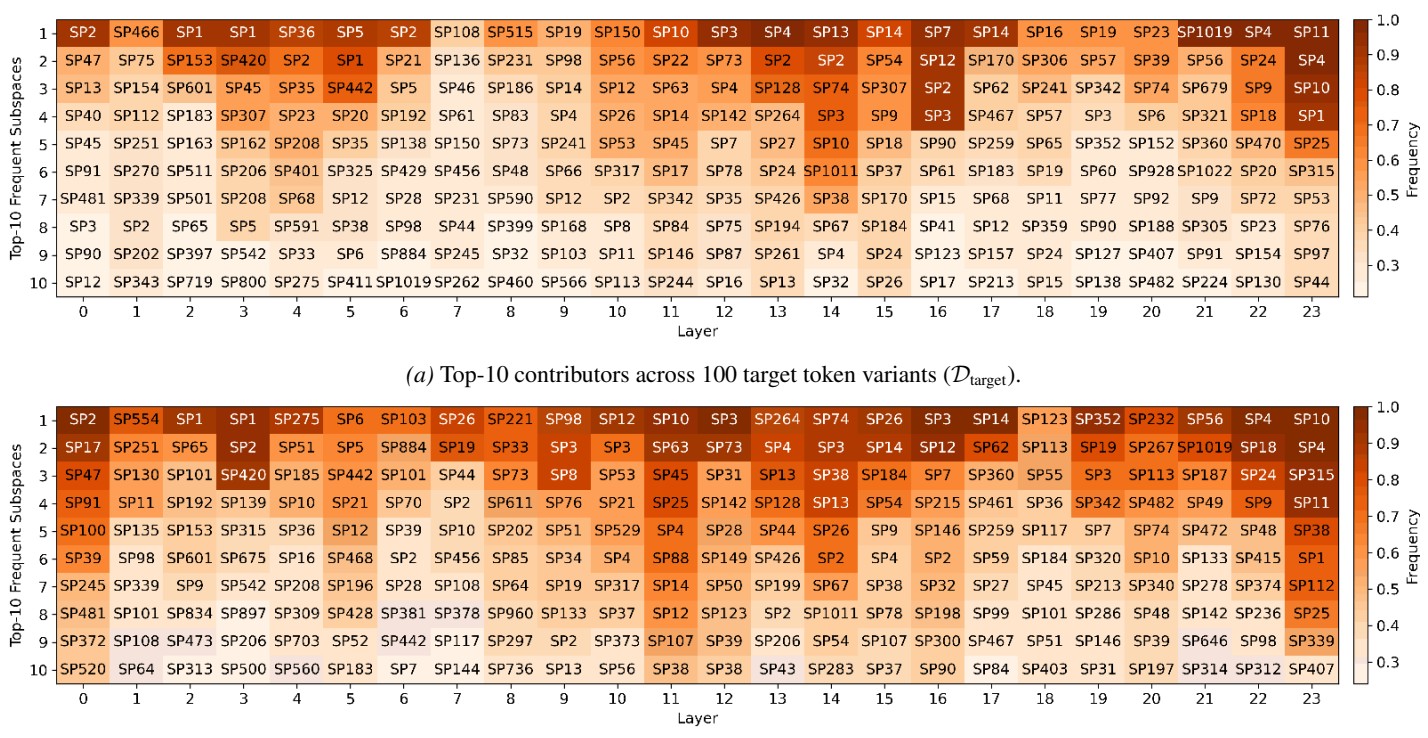

*(a)* Top-10 contributors across 100 target token variants ($\mathcal{D}_{\text{target}}$).

*(b)* Top-10 contributors across 100 prefix variants ($\mathcal{D}_{\text{prefix}}$).

*Figure 11.* **Layer-wise Consistency of Top-10 Subspace Activations Across Target Token Variants ($\mathcal{D}_{\text{target}}$) and Prefix Variants ($\mathcal{D}_{\text{prefix}}$) in GPT2-Medium.** The heatmaps illustrate, for each layer, the frequency with which each subspace appears among the top-10 contributors across all variants, thereby identifying subspaces that are consistently activated across diverse target tokens. Here, SP$i$ denotes subspace $i$.

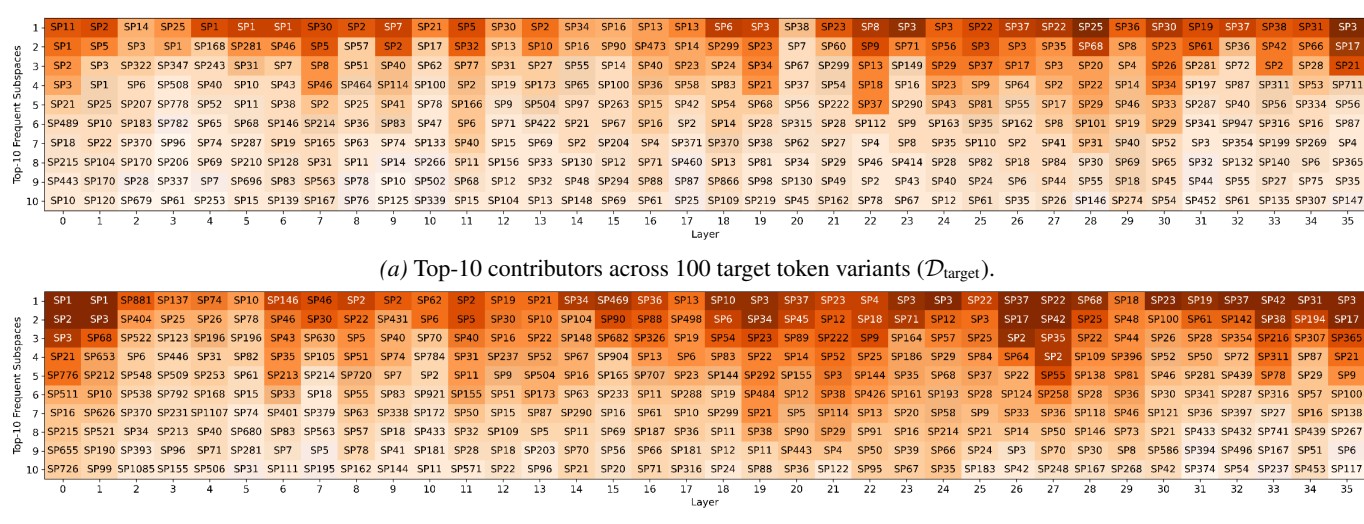

*(a)* Top-10 contributors across 100 target token variants ($\mathcal{D}_{\text{target}}$).

*(b)* Top-10 contributors across 100 prompt variants ($\mathcal{D}_{\text{prefix}}$).

*Figure 12.* **Layer-wise Consistency of Top-10 Subspace Activations Across Target Token Variants ($\mathcal{D}_{\text{target}}$) and Prefix Variants ($\mathcal{D}_{\text{prefix}}$) in GPT2-Large.** The heatmaps illustrate, for each layer, the frequency with which each subspace appears among the top-10 contributors across all variants, thereby identifying subspaces that are consistently activated across diverse target tokens. Here, SP$i$ denotes subspace $i$.

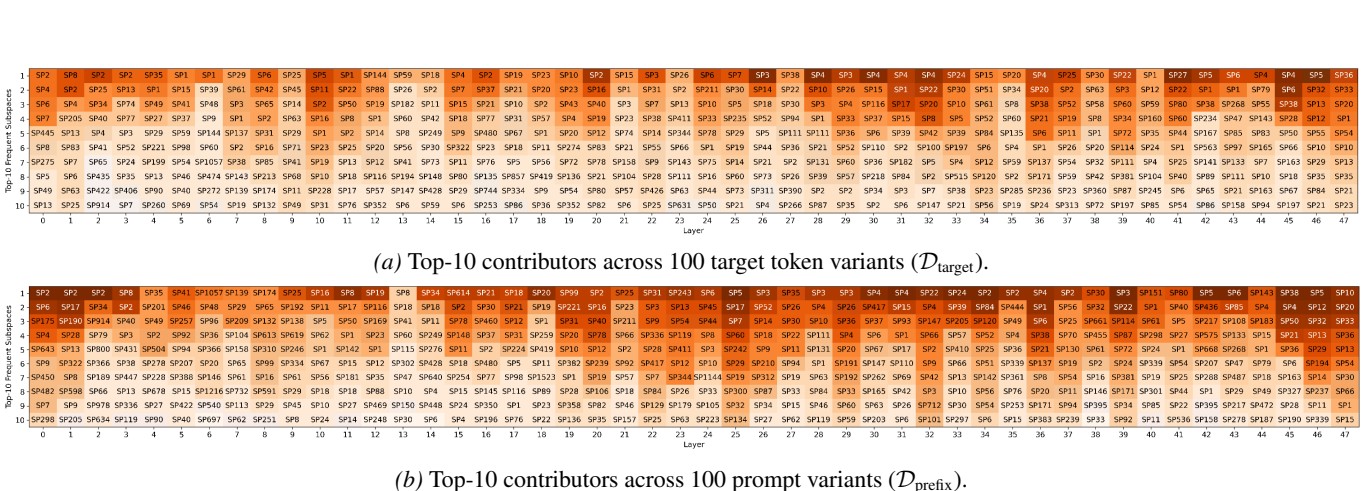

*(a)* Top-10 contributors across 100 target token variants ($\mathcal{D}_{\text{target}}$).

*(b)* Top-10 contributors across 100 prompt variants ($\mathcal{D}_{\text{prefix}}$).

*Figure 13.* **Layer-wise Consistency of Top-10 Subspace Activations Across Target Token Variants ($\mathcal{D}_{\text{target}}$) and Prefix Variants ($\mathcal{D}_{\text{prefix}}$) in GPT2-XL.** The heatmaps illustrate, for each layer, the frequency with which each subspace appears among the top-10 contributors across all variants, thereby identifying subspaces that are consistently activated across diverse target tokens. Here, SP$i$ denotes subspace $i$.

## F.2. Interpretation of Consistently Activated Subspaces

In this section, we examine whether these highly active subspaces exhibit direct and interpretable associations with specific semantic contexts. Tables 4-7 report the interpretability analysis of these subspaces under $\mathcal{D}_{\text{target}}$, where L$m$_SP$i$ denotes subspace $i$ in layer $m$. The result indicates that these high-frequency subspaces are typically related to positive semantics. It suggests that the model's predictions are mediated by specific, functional subspaces rather than by diffuse representations. Multiple subspaces, while semantically distinct, exhibit strong correlations and jointly contribute to robust predictions of positive semantics.

*Table 4.* **Layer-wise Interpretation of Highly Active Subspaces in GPT-2.** Here, parentheses denote subspace interpretability levels (High, Med, or Low). **Green** highlights denote subspaces associated with positive semantics.

| ID | Positive Direction | Negative Direction | ID | Positive Direction | Negative Direction |
|---|---|---|---|---|---|
| **L0_SP10** | Retail and Online Shopping (Med) | Prep. & Conj. (High) | **L7_SP2** | *null* (null) | *null* (null) |
| **L0_SP2** | punctuation marks (High) | Retail and Nature (Med) | **L7_SP4** | Legal and Social Issues (Med) | Photography Terms (Med) |
| **L0_SP4** | punctuation marks (High) | *null* (null) | **L7_SP7** | Neglect and Awareness (High) | game activities (High) |
| **L0_SP15** | *null* (null) | Content Management (Med) | **L7_SP24** | Prison-related terms (Med) | mindfulness and diligence (High) |
| **L0_SP12** | *null* (null) | Awakening Determination (Med) | **L7_SP13** | *null* (null) | recent events (High) |
| **L1_SP1** | function words (High) | *null* (null) | **L7_SP12** | death and decline (High) | Review and Insight (High) |
| **L1_SP7** | function words (High) | Retail and Delivery (Med) | **L8_SP15** | advantageous roles (Med) | time periods (Med) |
| **L1_SP120** | real estate topics (Med) | *null* (null) | **L8_SP14** | value and importance (High) | Deletion actions (High) |
| **L1_SP3** | Wrestling concepts (High) | Magnitudes and Rarity (Med) | **L8_SP385** | Flavoring Concepts (Med) | *null* (null) |
| **L1_SP8** | punctuation&conjunctions(Med) | Dragon Magazine (Med) | **L9_SP6** | Location references (High) | *null* (null) |
| **L1_SP319** | Menopause Related Terms (Med) | *null* (null) | **L9_SP754** | *null* (null) | *null* (null) |
| **L2_SP2** | *null* (null) | Utility Organizations (High) | **L9_SP177** | *null* (null) | political themes (Med) |
| **L2_SP1** | common conjunctions (High) | *null* (null) | **L9_SP20** | *null* (null) | *null* (null) |
| **L2_SP4** | *null* (null) | Travel and Logistics (Med) | **L10_SP48** | day references (Med) | oratory terms (Med) |
| **L2_SP83** | *null* (null) | Media and Learning (Med) | **L10_SP25** | *null* (null) | *null* (null) |
| **L3_SP27** | *null* (null) | social dynamics (Med) | **L10_SP88** | communication expressions (Med) | Skiing Activities (Med) |
| **L3_SP46** | Criminality and Exclusion (High) | advertising concepts (High) | **L11_SP3** | Retail and Confirmation (Med) | punctuation usage (High) |
| **L3_SP2** | Chance and Opportunity (Med) | *null* (null) | **L11_SP101** | system concepts (Med) | Nationalities and Locations (High) |
| **L3_SP106** | *null* (null) | Food items (Med) | **L11_SP64** | ability concepts (High) | Locations and Places (Med) |
| **L6_SP3** | User Interface Elements (Med) | *null* (null) | **L11_SP86** | advice and guidance (Med) | Emotional expressions (Med) |
| **L6_SP700** | *null* (null) | Public Figures (Med) | | | |

*Table 5.* **Layer-wise Interpretation of Highly Active Subspaces in GPT2-Medium.** Here, parentheses denote subspace interpretability levels (High, Med, or Low). **Green** highlights denote subspaces associated with positive semantics.

| ID | Positive Direction | Negative Direction |
|---|---|---|
| L0_SP2 | *null* (null) | punctuation usage (High) |
| L1_SP466 | Rewards and Characters (Med) | Detention and Control (Med) |
| L2_SP1 | Common conjunctions (High) | *null* (null) |
| L2_SP153 | Cultural References (Med) | *null* (null) |
| L3_SP1 | *null* (null) | punctuation usage (Med) |
| L3_SP420 | Government and Authority (Med) | *null* (null) |
| L3_SP45 | *null* (null) | Information Retrieval (Med) |
| L3_SP307 | Management Concepts (Med) | Political Terms (Med) |
| L4_SP36 | virtual balance (Med) | Social Events (Med) |
| L4_SP2 | *null* (null) | punctuation&conjunctions (Med) |
| L4_SP35 | Nebraska and related terms (Med) | Critique and Analysis (High) |
| L5_SP5 | *null* (null) | Energy Industries (High) |
| L5_SP1 | articles and conjunctions (High) | *null* (null) |
| L5_SP442 | Media and Communication (Med) | *null* (null) |
| L6_SP2 | *null* (null) | Political Issues (Med) |
| L8_SP515 | Addiction and Faces (Med) | Conflict and Grievance (High) |
| L10_SP150 | *null* (null) | *null* (null) |
| L10_SP56 | *null* (null) | Business Management (Med) |
| L10_SP12 | Scientific terminology (Med) | *null* (null) |
| L11_SP10 | Names and Places (Med) | Quantity and Rarity (Med) |
| L11_SP22 | trustworthiness attributes (High) | *null* (null) |
| L11_SP63 | Food Quality (Med) | Financial Terms (Med) |
| L11_SP14 | Education and Travel (Med) | negatives and errors (Med) |
| L12_SP3 | *null* (null) | *null* (null) |
| L12_SP73 | *null* (null) | Positive Traits (High) |
| L12_SP4 | *null* (null) | people and organizations (High) |
| L13_SP4 | game items (Med) | warning concepts (Med) |
| L13_SP2 | *null* (null) | Punctuation and Symbols (Med) |
| L13_SP128 | education assessment (Med) | *null* (null) |

| ID | Positive Direction | Negative Direction |
|---|---|---|
| L14_SP13 | *null* (null) | vibrant leisure (High) |
| L14_SP2 | *null* (null) | Punctuation and Symbols (High) |
| L14_SP74 | Government Challenges (Med) | *null* (null) |
| L14_SP3 | electricity concepts (Med) | mental states (Med) |
| L14_SP10 | Positive Excellence (High) | Measurement and Time (Med) |
| L14_SP101 | struggle and challenges (Med) | *null* (null) |
| L14_SP38 | termination concepts (Med) | indifference and desire (Med) |
| L15_SP14 | updates and terminations (High) | emotional experiences (High) |
| L15_SP54 | Geographical Terms (Med) | Social Media Roles (Med) |
| L15_SP307 | Weddings and Events (Med) | *null* (null) |
| L15_SP9 | similarity concepts (High) | travel actions (High) |
| L16_SP7 | Government Deaths (High) | positive qualities (High) |
| L16_SP12 | Respect and Worthiness (High) | Numbers and Quantities (High) |
| L16_SP2 | Punctuation Marks (High) | *null* (null) |
| L16_SP3 | Observation and Critique (Med) | electric devices (High) |
| L17_SP14 | Cognitive Values (High) | Security incidents (High) |
| L18_SP16 | one and variations (High) | second person pronouns (High) |
| L18_SP306 | Data Management (Med) | *null* (null) |
| L19_SP19 | *null* (null) | *null* (null) |
| L19_SP57 | Face Tracking (Med) | Advertising Concepts (Med) |
| L20_SP23 | Outsider references (Med) | action or activity (High) |
| L20_SP39 | *null* (null) | Creation and Time (Med) |
| L20_SP74 | work and charity (Med) | growth and vitality (Med) |
| L21_SP1019 | *null* (null) | Marketing Concepts (Med) |
| L22_SP4 | *null* (null) | articles and conjunctions (High) |
| L22_SP24 | *null* (null) | people and workers (High) |
| L22_SP9 | Nationalities and Places (Med) | Prep. & Conj. (High) |
| L22_SP18 | comparative adjectives (High) | action verbs (Med) |
| L23_SP11 | the definite article (High) | Gaming References (Med) |
| L23_SP4 | *null* (null) | punctuation&conjunctions (Med) |
| L23_SP10 | *null* (null) | of and related (High) |
| L23_SP1 | *null* (null) | punctuation&conjunctions (Med) |
| L23_SP25 | Creation and Discussion (Med) | Veterinary Medicine (High) |

*Table 6.* **Layer-wise Interpretation of Highly Active Subspaces in GPT2-Large.** Here, parentheses denote subspace interpretability levels (High, Med, or Low). **Green** highlights denote subspaces associated with positive semantics.

| ID | Positive Direction | Negative Direction | ID | Positive Direction | Negative Direction |
|---|---|---|---|---|---|
| L0_SP11 | *null* (null) | Voting and Associations (Med) | L23_SP3 | Fictional Characters (High) | challenges and limitations (High) |
| L0_SP1 | Media and Gaming (Med) | *null* (null) | L23_SP71 | emotional decline (High) | impact assessment (Med) |
| L0_SP2 | Financial Analysis (High) | *null* (null) | L24_SP3 | Names of People (High) | action and consequences (High) |
| L1_SP2 | Government Support (Med) | material properties (High) | L24_SP56 | Celebration and Excellence (High) | Statements and Predictions (High) |
| L1_SP5 | *null* (null) | Music and Culture (Med) | L24_SP29 | management concepts (High) | vivid sensations (High) |
| L3_SP25 | Celebration Events (Med) | Developmental Concepts (Med) | L25_SP22 | investigation actions (High) | positive attributes (High) |
| L4_SP1 | *null* (null) | social issues (Med) | L25_SP3 | Names of People (High) | discussion engagement (High) |
| L5_SP1 | *null* (null) | people and actions (Med) | L25_SP37 | modification terms (Med) | bravery and acclaim (High) |
| L5_SP281 | Courage and Visibility (Med) | *null* (null) | L26_SP37 | negative traits (High) | historical interpretations (High) |
| L6_SP1 | people and actions (Med) | *null* (null) | L26_SP3 | Names and Titles (High) | *null* (null) |
| L6_SP46 | Health and Technology (Med) | Tools and Equipment (Med) | L26_SP17 | *null* (null) | Positive Emotions (High) |
| L7_SP30 | Cells and Structures (Med) | *null* (null) | L27_SP22 | uncertainty concepts (High) | *null* (null) |
| L7_SP5 | overcoming challenges (High) | Screen elements (Med) | L27_SP35 | protection and safety (High) | Adventure Themes (Med) |
| L7_SP8 | visual representation (High) | Raids and Settlements (High) | L27_SP3 | *null* (null) | thematic references (Med) |
| L7_SP46 | official documentation (Med) | *null* (null) | L28_SP25 | Positive Attributes (High) | nature elements (Med) |
| L8_SP2 | *null* (null) | general expressions (Med) | L28_SP68 | war and conflict (High) | carelessness and rights (Med) |
| L9_SP7 | *null* (null) | Booking and Recording (Med) | L29_SP36 | rejection themes (Med) | Between Points (Med) |
| L9_SP2 | conjunctions and discourse (Med) | *null* (null) | L30_SP30 | extreme quality (High) | *null* (null) |
| L10_SP21 | sports competition (Med) | *null* (null) | L30_SP23 | *null* (null) | data representation (Med) |
| L11_SP5 | *null* (null) | real estate activities (High) | L30_SP26 | stability and safety (Med) | *null* (null) |
| L11_SP32 | competition winners (High) | development abilities (Med) | L30_SP34 | Quality Descriptors (High) | *null* (null) |
| L12_SP30 | natural disasters (High) | Fashion and Style (Med) | L31_SP19 | Punctuation marks (High) | Punctuation and Symbols (High) |
| L13_SP2 | Unleashed Outbreak (Med) | *null* (null) | L31_SP61 | *null* (null) | *null* (null) |
| L13_SP10 | surrounding contexts (High) | *null* (null) | L32_SP37 | Right Direction (Med) | Gaming Entities (High) |
| L17_SP13 | Apocalyptic Events (High) | collaboration roles (High) | L33_SP38 | *null* (null) | Financial Transactions (High) |
| L18_SP6 | uniqueness and perfection (Med) | International Organizations (High) | L33_SP42 | Danger and States (Med) | public technology (Med) |
| L18_SP299 | Licensing&Engagements (Med) | *null* (null) | L33_SP2 | *null* (null) | Prep. & Conj. (High) |
| L19_SP3 | Measurement Terms (Med) | Famous Individuals (High) | L34_SP31 | Time references (High) | Ordinal Numbers (High) |
| L19_SP23 | Financial Disqualification (High) | creativity and inspiration (High) | L34_SP66 | power and energy (High) | Nautical Terms (Med) |
| L19_SP34 | monitoring techniques (Med) | Preferred Choices (High) | L35_SP3 | causal conjunctions (Med) | *null* (null) |
| L19_SP21 | Positive Feedback (High) | underground structures (High) | L35_SP17 | *null* (null) | Years Mentioned (High) |
| L21_SP23 | chaos management (Med) | Friendship and Kindness (High) | L35_SP21 | degree comparison (High) | Geographical Entities (High) |
| L22_SP8 | abandonment themes (Med) | inspiration and insight (High) | | | |
| L22_SP9 | Event References (High) | human behaviors (High) | | | |
| L22_SP13 | Advocacy and Roles (High) | quantitative measures (High) | | | |
| L22_SP18 | struggles and recovery (High) | Geometric Shapes (High) | | | |
| L22_SP37 | change and history (High) | emotional distress (High) | | | |

*Table 7.* **Layer-wise Interpretation of Highly Active Subspaces in GPT2-XL.** Here, parentheses denote subspace interpretability levels (High, Med, or Low). **Green** highlights denote subspaces associated with positive semantics.

| ID | Positive Direction | Negative Direction | ID | Positive Direction | Negative Direction |
|---|---|---|---|---|---|
| L0_SP2 | Behavioral Goals (Med) | *null* (null) | L30_SP4 | cautionary observations (High) | *null* (null) |
| L0_SP4 | Financial Instruments (High) | *null* (null) | L30_SP15 | social interactions (High) | Administrative Terms (Med) |
| L0_SP6 | Financial Terms (Med) | Friendship Concepts (Med) | L30_SP116 | Media and Culture (Med) | *null* (null) |
| L0_SP7 | Travel and Hospitality (Med) | environmental concepts (Med) | L30_SP37 | resource allocation (Med) | daily activities (High) |
| L1_SP8 | *null* (null) | Organizations&Agencies (High) | L31_SP4 | *null* (null) | emotional responses (High) |
| L1_SP2 | Media and Promotion (Med) | *null* (null) | L31_SP1 | *null* (null) | Punctuation Marks (High) |
| L1_SP4 | Automotive Advertising (Med) | Liquid Enhancement (High) | L31_SP17 | time duration (High) | positive attributes (High) |
| L2_SP2 | *null* (null) | *null* (null) | L31_SP15 | Economic Hardship (High) | Reports and References (Med) |
| L2_SP25 | *null* (null) | Communication Actions (Med) | L32_SP4 | *null* (null) | briefing documents (High) |
| L2_SP34 | optimism themes (Med) | *null* (null) | L32_SP22 | failure and success (Med) | Visual Media (High) |
| L3_SP2 | *null* (null) | negative experiences (High) | L32_SP20 | excellence and power (High) | negation terms (High) |
| L3_SP13 | technology usage (Med) | Publishing Concepts (Med) | L32_SP8 | Locations and Events (Med) | Negative Evaluation (High) |
| L4_SP35 | Plants and Growth (High) | *null* (null) | L33_SP24 | *null* (null) | personal growth (High) |
| L4_SP1 | articles and conjunctions (High) | *null* (null) | L33_SP30 | rejection behaviors (High) | Validation Process (High) |
| L4_SP49 | Social Activities (Med) | food substances (Med) | L33_SP10 | *null* (null) | extreme qualities (High) |
| L5_SP1 | function words (High) | *null* (null) | L33_SP5 | *null* (null) | greatness and limits (Med) |
| L5_SP15 | *null* (null) | romantic themes (High) | L34_SP15 | documents and media (Med) | improvement actions (High) |
| L6_SP1 | function words (High) | *null* (null) | L35_SP20 | *null* (null) | numerical references (High) |
| L8_SP6 | transportation services (High) | Action and Conditions (Med) | L36_SP4 | *null* (null) | *null* (null) |
| L8_SP42 | Computing Processes (High) | Relief and Assurance (Med) | L36_SP20 | action and evaluation (Med) | high quality (Med) |
| L8_SP65 | Energy Resources (Med) | Technology Terms (Med) | L36_SP38 | negation terms (High) | Health Risks (High) |
| L10_SP5 | *null* (null) | Rental Properties (High) | L36_SP21 | Personal Pronouns (High) | possessive pronouns (High) |
| L10_SP11 | events and activities (Med) | color attributes (High) | L36_SP6 | Possessive Pronouns (High) | Fashion and Activities (Med) |
| L10_SP2 | *null* (null) | Prep. & Conj. (High) | L37_SP25 | *null* (null) | Travel and Aviation (High) |
| L10_SP16 | *null* (null) | Environmental Issues (High) | L37_SP2 | business topics (Med) | *null* (null) |
| L10_SP1 | large scale (High) | *null* (null) | L37_SP52 | Informative Communication (Med) | Measurement Terms (Med) |
| L11_SP1 | *null* (null) | large scale (High) | L37_SP19 | *null* (null) | Numerical References (High) |
| L11_SP22 | Buyers and Sellers (High) | disruption and change (High) | L37_SP11 | *null* (null) | *null* (null) |
| L11_SP50 | artistic expression (Med) | accumulation and tracking (Med) | L38_SP30 | Emotional expressions (Med) | returning actions (Med) |
| L12_SP144 | Awakening Concepts (Med) | Urban Areas (High) | L39_SP22 | *null* (null) | comparative qualities (Med) |
| L12_SP88 | news reporting (High) | Trade and Supply (High) | L39_SP3 | *null* (null) | Political Power (High) |
| L15_SP4 | justification and meaning (Med) | Urban Institutions (High) | L39_SP60 | *null* (null) | Financial Transactions (High) |
| L15_SP7 | *null* (null) | conjunctions&functions (High) | L39_SP34 | office locations (Med) | Emotional expressions (Med) |
| L15_SP15 | Creative expression (Med) | financial regulations (High) | L39_SP72 | Length Measurement (High) | *null* (null) |
| L16_SP2 | *null* (null) | *null* (null) | L40_SP1 | large scale (High) | *null* (null) |
| L16_SP37 | chaos and greatness (Med) | Professional Roles (Med) | L41_SP27 | *null* (null) | *null* (null) |
| L17_SP19 | gathering activities (High) | Legal and Safety Terms (High) | L41_SP22 | Corporate Terms (Med) | positive attributes (High) |
| L18_SP23 | success and persistence (Med) | emotional concepts (High) | L41_SP80 | *null* (null) | Command Functions (Med) |
| L18_SP20 | individuality concepts (Med) | Media&Communication (Med) | L41_SP60 | Direction and Action (High) | Motherhood Concepts (Med) |
| L19_SP10 | indifference and rigidity (Med) | Financial Institutions (High) | L42_SP5 | Space Exploration (Med) | *null* (null) |
| L19_SP23 | requirements and options (High) | adventure themes (Med) | L42_SP1 | large scale concepts (High) | *null* (null) |
| L19_SP43 | navigation concepts (Med) | Deception Concepts (High) | L42_SP38 | have and had forms (High) | Financial Terms (Med) |
| L20_SP2 | Conj. & Punct. (Med) | *null* (null) | L43_SP6 | *null* (null) | finance and work (Med) |
| L20_SP16 | love and inspiration (High) | dependency concepts (High) | L43_SP1 | large scale concepts (High) | *null* (null) |
| L20_SP40 | Health and Technology (Med) | pleasant and weather (High) | L44_SP4 | *null* (null) | Advertising Terms (Med) |
| L20_SP19 | large quantities (High) | professionalism themes (Med) | L45_SP4 | Mathematical Symbols (Med) | *null* (null) |
| L21_SP15 | Legal & Finance (High) | Aesthetic Descriptions (High) | L45_SP6 | *null* (null) | Prep. & Conj. (High) |
| L22_SP3 | advertising concepts (Med) | *null* (null) | L45_SP38 | Cultural References (Med) | Logical Operators (High) |
| L24_SP6 | governmental authority (High) | *null* (null) | L45_SP28 | Punctuation and Symbols (High) | Quotations and Speech (High) |
| L24_SP211 | scheduling concepts (Med) | warning and action (Med) | L46_SP5 | *null* (null) | *null* (null) |
| L24_SP10 | collaboration and growth (High) | errors and failures (High) | L46_SP32 | time references (High) | openness and closure (High) |
| L24_SP33 | Behavioral Analysis (High) | financial support (High) | L46_SP13 | *null* (null) | Numerical References (High) |
| L25_SP7 | innovation and impact (High) | indoor activities (Med) | L46_SP12 | Influencing actions (High) | possessive pronouns (High) |
| L26_SP3 | Names of People (High) | emotional struggle (High) | L47_SP36 | *null* (null) | Dates and Months (High) |
| L26_SP14 | emotional experiences (High) | representation concepts (High) | L47_SP33 | games and players (High) | web documents (High) |
| L27_SP38 | education contribution (High) | hidden truths (High) | L47_SP20 | question words (High) | Reports and Publications (High) |
| L28_SP4 | confusion themes (High) | *null* (null) | L47_SP1 | Fantasy Characters (High) | *null* (null) |
| L28_SP10 | Completion status (High) | Social Groups (High) | L47_SP54 | legal rules (High) | positive emotions (High) |
| L28_SP3 | *null* (null) | Corporate Entities (Med) | | | |
| L29_SP3 | negative emotions (High) | *null* (null) | | | |
| L29_SP26 | Unpredictable roles (Med) | disappearance&removal (High) | | | |
| L29_SP4 | misunderstanding (High) | Names and Places (Med) | | | |
| L29_SP33 | Reports and Analysis (High) | Trust and Relationships (High) | | | |

# G. Computational Efficiency and Scalability

In this section, we describe the operational flexibility of our framework, which supports two distinct modes:

- **Global Pre-Computation:** This mode constructs a comprehensive interpretability datastore for every DEU subspace across all layers. Consequently, during pathway extraction, the interpretations of high-contribution subspaces can be directly queried, enabling efficient and reusable access.

- **On-Demand Interpretation:** This model first extracts the pathway and subsequently generates interpretations solely for the required top-$k$ high-contribution subspaces. This procedure minimizes upfront computational overhead and enables efficient targeted analysis.

## G.1. Computational Cost of Building the Interpretability Datastore

To assess the computational feasibility of constructing the interpretability datastore, we evaluated resource consumption across four GPT-2 variants. The analysis quantifies the computational overhead per layer as a function of matrix size, defined as $hidden\_size \times mlp\_dim$. To ensure consistent performance metrics, all benchmarks were executed on a single **NVIDIA A100 GPU (80GB)**.

As illustrated in Figure 14, the system demonstrates efficient scaling behavior regarding both runtime and memory usage. The total runtime, which aggregates the $W_{\text{up}}$ and $W_{\text{down}}$ interpretation stages, exhibits a strictly linear correlation with matrix size. Notably, even for the largest variant, GPT2-XL (characterized by 20.5M parameters per layer), the process remains highly efficient and requires approximately **7 minutes per layer**. Similarly, peak GPU memory consumption scales linearly, ranging from less than 1 GB for GPT-2 to approximately 6.5 GB for GPT2-XL, a range that falls well within the capacity of standard consumer and enterprise hardware.

The linear scaling trends and low absolute resource requirements confirm that building the interpretability datastore is computationally feasible. The process is highly scalable, allowing for the efficient interpretation of larger language model layers without requiring prohibitive hardware resources.

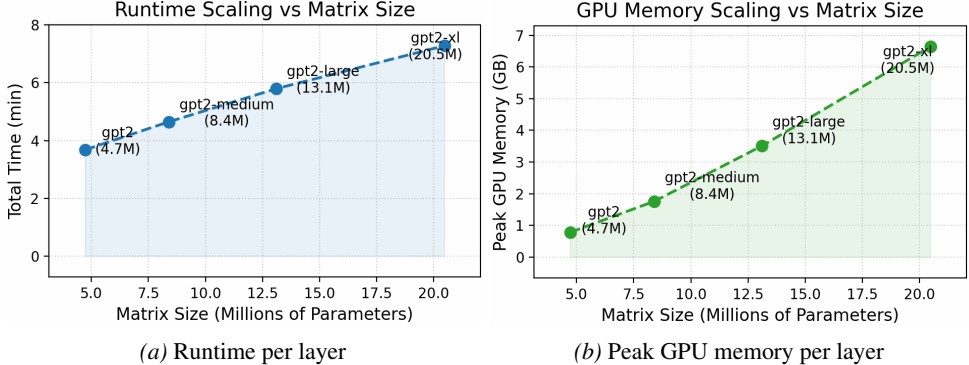

*(a)* Runtime per layer        *(b)* Peak GPU memory per layer

*Figure 14.* **Computational Cost of Constructing an Interpretability Datastore per Layer for Models of Varying Scale on a Single NVIDIA A100 GPU (80GB).** Total runtime (in minutes) is reported as the sum of the $W_{\text{up}}$ and $W_{\text{down}}$ interpretation stages. Matrix size is measured in millions of parameters, defined as $hidden\_size \times mlp\_dim$.

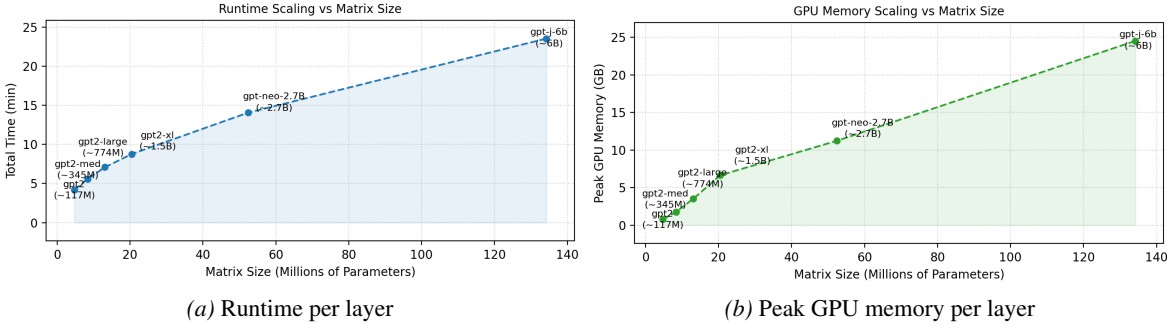

*(a)* Runtime per layer

*(b)* Peak GPU memory per layer

*Figure 15.* **Computational Cost of Constructing an Interpretability Datastore per Layer for Models of Varying Scale on a Single NVIDIA A100 GPU (40GB).** Total runtime (in minutes) is reported as the sum of the $W_{\mathrm{up}}$ and $W_{\mathrm{down}}$ interpretation stages. Matrix size is measured in millions of parameters, defined as $hidden\_size \times mlp\_dim$.

## G.2. Runtime of Pathway Extraction

We further evaluate the computational cost of pathway extraction under varying sparsity levels, considering top-$k$ high-contribution subspaces with $k \in [1, 100]$ across different model variants.

Figure 16 shows that the total runtime remains effectively constant whether pathway extraction is restricted to a single subspace or extended to 100 subspaces. This indicates that increasing the granularity of subspace analysis incurs no additional computational overhead. Moreover, pathway extraction is consistently efficient across model scales, requiring approximately 0.32 minutes for GPT2-Medium and 0.75 minutes for GPT2-Large. Even for GPT2-XL, the maximum recorded runtime is merely 1.64 minutes.

Overall, pathway extraction is completed within two minutes across all configurations, underscoring the method's efficiency and applicability in real-world settings.

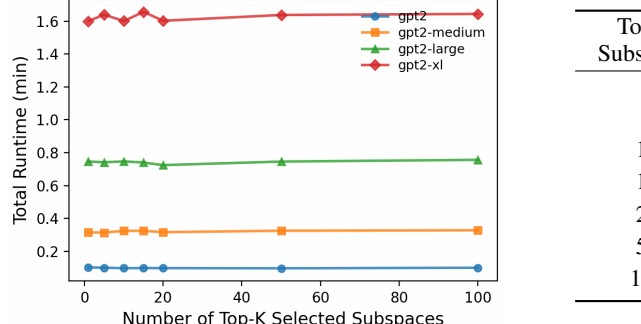

| Top-K Subspaces | GPT-2 | GPT2-Medium | GPT2-Large | GPT2-XL |
|---|---|---|---|---|
| 1 | 0.10 | 0.31 | 0.75 | 1.60 |
| 5 | 0.10 | 0.31 | 0.74 | 1.64 |
| 10 | 0.10 | 0.32 | 0.75 | 1.60 |
| 15 | 0.10 | 0.32 | 0.74 | 1.66 |
| 20 | 0.10 | 0.32 | 0.72 | 1.60 |
| 50 | 0.10 | 0.32 | 0.75 | 1.64 |
| 100 | 0.10 | 0.33 | 0.76 | 1.64 |

*Figure 16.* Runtime for Pathway Extraction Using Top-$K$ Subspaces Across the GPT-2 Family on a Single NVIDIA A100 (80GB).

## G.3. Runtime of Pathway Extraction and On-Demand Interpretation

Unlike SAEs and TransCoders that generally rely on extensive precomputation of global features, our framework enables flexible, on-demand analysis. This method enables direct interpretation of the top-ranked subspaces within the identified pathways, while avoiding the computational overhead associated with processing irrelevant model components.

Figure 17 reports the runtime of the complete analysis pipeline, which encompasses both pathway extraction and local interpretation across all layers. The results exhibit an approximately linear relationship between runtime and the number of top-$k$ subspaces, with the overall pipeline remaining highly efficient. For the largest model, GPT2-XL, the pipeline requires 6.62 minutes for a single subspace ($k = 1$) and 17.56 minutes for a comprehensive analysis of 100 subspaces ($k = 100$). For smaller models such as GPT-2, the entire pipeline is completed in approximately 3 minutes even at $k = 100$. These findings indicate that the framework can efficiently extract mechanistic pathways and yield interpretable insights within a short timeframe, highlighting its suitability for practical deployment in large-scale model analyses.

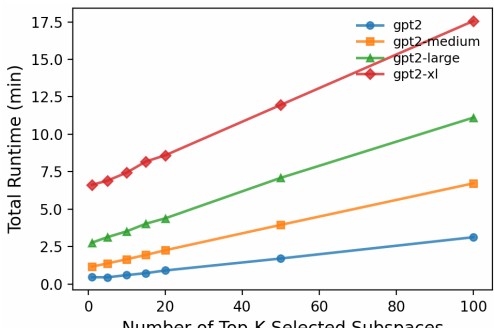

| Top-K Subspaces | GPT-2 | GPT2-Medium | GPT2-Large | GPT2-XL |
|---|---|---|---|---|
| 1 | 0.45 | 1.16 | 2.76 | 6.62 |
| 5 | 0.45 | 1.37 | 3.13 | 6.90 |
| 10 | 0.59 | 1.65 | 3.51 | 7.43 |
| 15 | 0.73 | 1.94 | 4.02 | 8.18 |
| 20 | 0.90 | 2.25 | 4.38 | 8.59 |
| 50 | 1.70 | 3.95 | 7.09 | 11.96 |
| 100 | 3.12 | 6.72 | 11.11 | 17.56 |

*Figure 17.* Runtime for Pathway Extraction with Local Interpretation Using Top-$K$ Subspaces Across the GPT-2 Family on a Single NVIDIA A100 (80GB).

## H. Twenty Representative Cases with Pathway Extraction Analysis

In this section, we present twenty representative cases covering a broad spectrum of linguistic phenomena. Each case illustrates distinct patterns in model behavior, enabling a detailed evaluation of pathway extraction and interpretability across syntactic, semantic, and pragmatic contexts.

### H.1. 20 Cases of Diverse Linguistic Phenomena

Within the NaNA framework, SCA serves as a directed pathway extraction method, distinguished by its explicit filtering of *causal bridges*. We define such bridges as subspaces that simultaneously exhibit input activation (detector alignment) and target alignment (effector alignment). To validate the extracted pathways, we evaluate whether they can recover or steer generation toward the specified targets. Notably, for the 20 anchor cases, certain target tokens do not correspond to the model's original top-1 predictions. Since our method is explicitly designed for directed pathway extraction toward arbitrary targets, this setting is entirely valid.

*Table 8.* Twenty Representative Cases Covering Diverse Linguistic Phenomena.

| # | Input Example | Target Token | Pathway Type | Linguistic Phenomenon |
|---|---|---|---|---|
| 1 | The cat looks very | happy | **Syntax–Semantics** | Adjective completion |
| 2 | Code snippet: for i in range(5): print | ( | **Code Syntax** | Parenthesis completion |
| 3 | The square root of 64 is | 8 | **Factual / Math** | Arithmetic fact retrieval |
| 4 | The dogs in the park | are | **Syntax** | Subject–verb agreement |
| 5 | The capital of Germany is | Berlin | **Factual Knowledge** | Geographic knowledge retrieval |
| 6 | Water boils at 100 degrees | Celsius | **Factual Knowledge** | Scientific fact retrieval |
| 7 | Code snippet: for char in 'hello': print | char | **Code Syntax** | Loop variable completion |
| 8 | John and Mary went to the | church | **Semantics** | Plausible location inference |
| 9 | She didn't see the sign, so she | didn | **Morphology** | Negation contraction |
| 10 | She quickly ran to the | door | **Semantics** | Goal argument completion |
| 11 | The sun rises in the | east | **Factual Knowledge** | Common-sense astronomy |
| 12 | The food was not bad, it was | good | **Semantics / Pragmatics** | Scalar implicature |
| 13 | The boy who cried wolf finally | got | **Semantics** | Pragmatic inference |
| 14 | The committee decided to postpone the | vote | **Semantics** | Event nominal completion |
| 15 | During the thunderstorm, the lights | of | **Discourse** | Causal continuation |
| 16 | The movie was thrilling but also | sad | **Semantics** | Contrastive adjective completion |
| 18 | The recipe calls for two cups of | sugar | **Factual / Procedural** | Ingredient selection |
| 16 | To solve the equation, one must first | simplify | **Procedural Knowledge** | Step ordering |
| 19 | Alice gave Bob the book because she | wanted | **Discourse / Coreference** | Implicit intention inference |
| 20 | If I had studied harder, I | would | **Syntax / Semantics** | Counterfactual construction |

## H.2. Evaluation of Pathway Extraction Across Twenty Linguistic Cases: Standard vs. Ablation

### Case 1: Prompt "The cat looks very" with Target Token " happy".

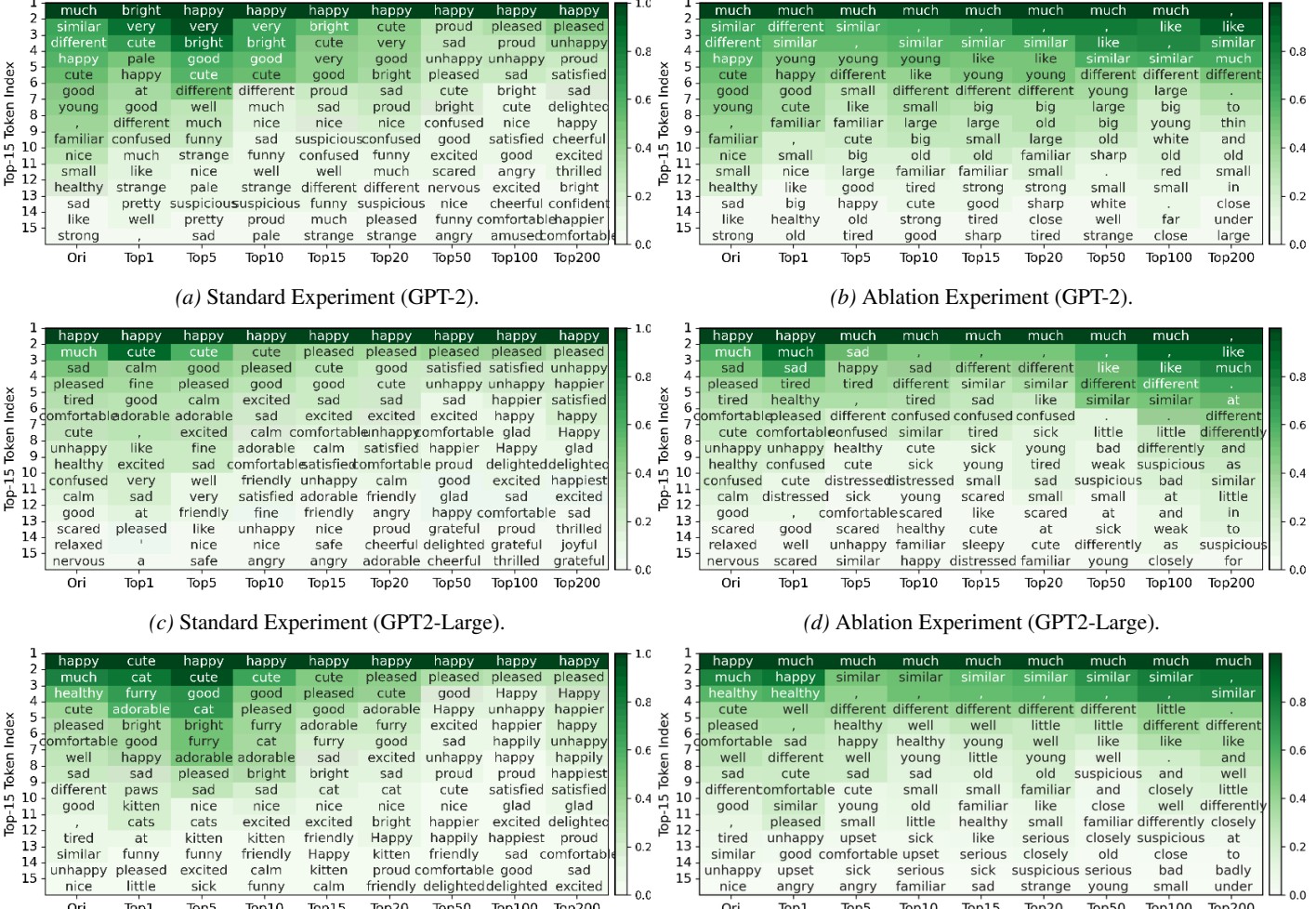

*(a)* Standard Experiment (GPT-2).  *(b)* Ablation Experiment (GPT-2).

*(c)* Standard Experiment (GPT2-Large).  *(d)* Ablation Experiment (GPT2-Large).

*(e)* Standard Experiment (GPT2-XL).  *(f)* Ablation Experiment (GPT2-XL).

*Figure 18.* **Reconstruction of Middle-to-Late MLP Layers via Pathways within Top-$k$ High-Contribution Subspaces: Standard vs. Ablation Studies.** This figure illustrates a case study using the prompt "*The cat looks very*" with the target token " *happy*". We visualize the top-15 predicted tokens across GPT-2 family under standard and ablation settings.

**Case 2: Prompt "Code snippet: for i in range(5): print" with Target Token " (".**

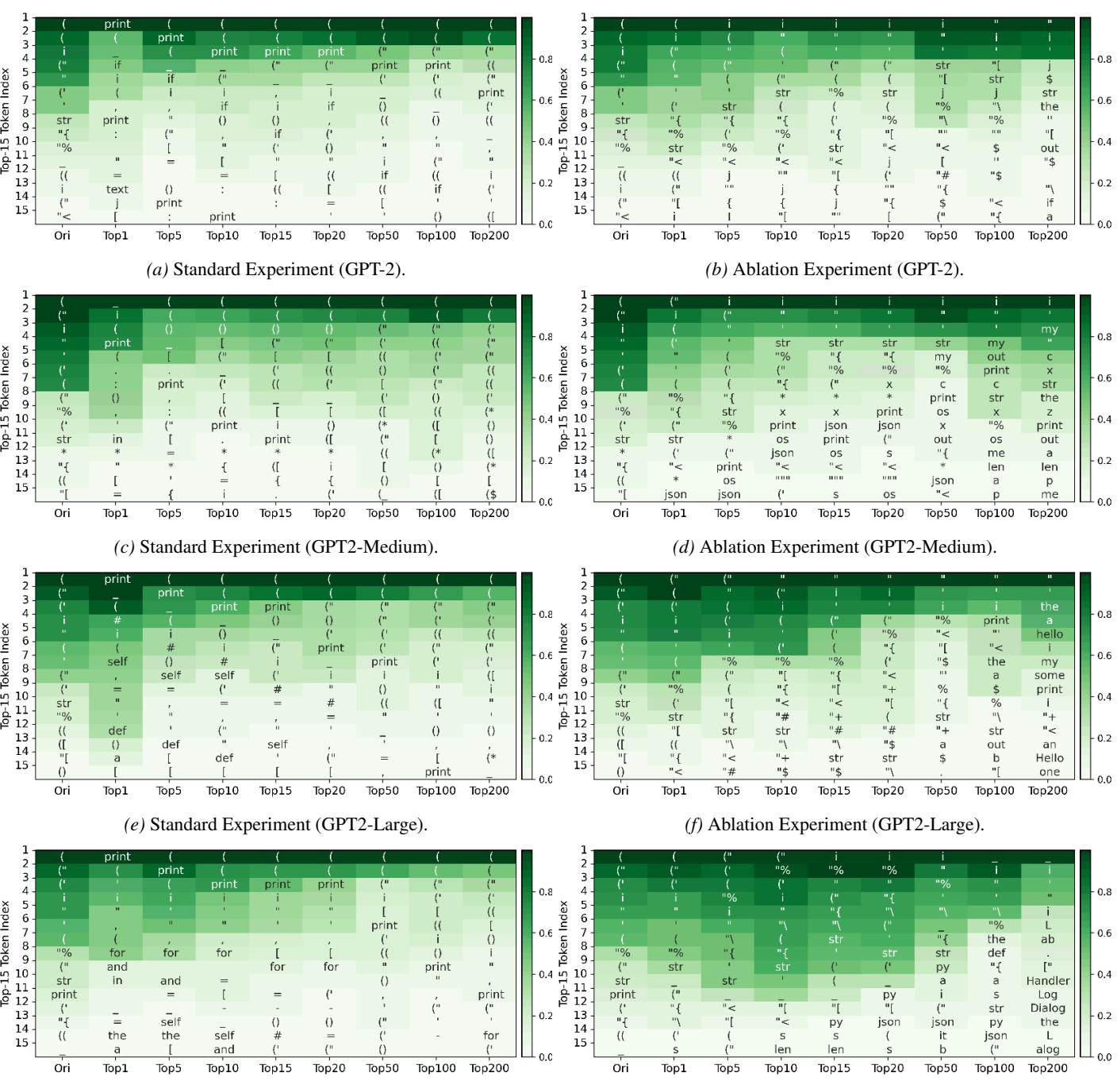

*(a)* Standard Experiment (GPT-2).

*(b)* Ablation Experiment (GPT-2).

*(c)* Standard Experiment (GPT2-Medium).

*(d)* Ablation Experiment (GPT2-Medium).

*(e)* Standard Experiment (GPT2-Large).

*(f)* Ablation Experiment (GPT2-Large).

*(g)* Standard Experiment (GPT2-XL).

*(h)* Ablation Experiment (GPT2-XL).

*Figure 19.* **Reconstruction of Middle-to-Late MLP Layers via Pathways within Top-$k$ High-Contribution Subspaces: Standard vs. Ablation Studies.** This figure illustrates a case study using the prompt "*Code snippet: for i in range(5): print*" with the target token " (". We visualize the top-15 predicted tokens across GPT-2 variants under standard and ablation settings.

**Case 3: Prompt "The square root of 64 is" with Target Token " 8".**

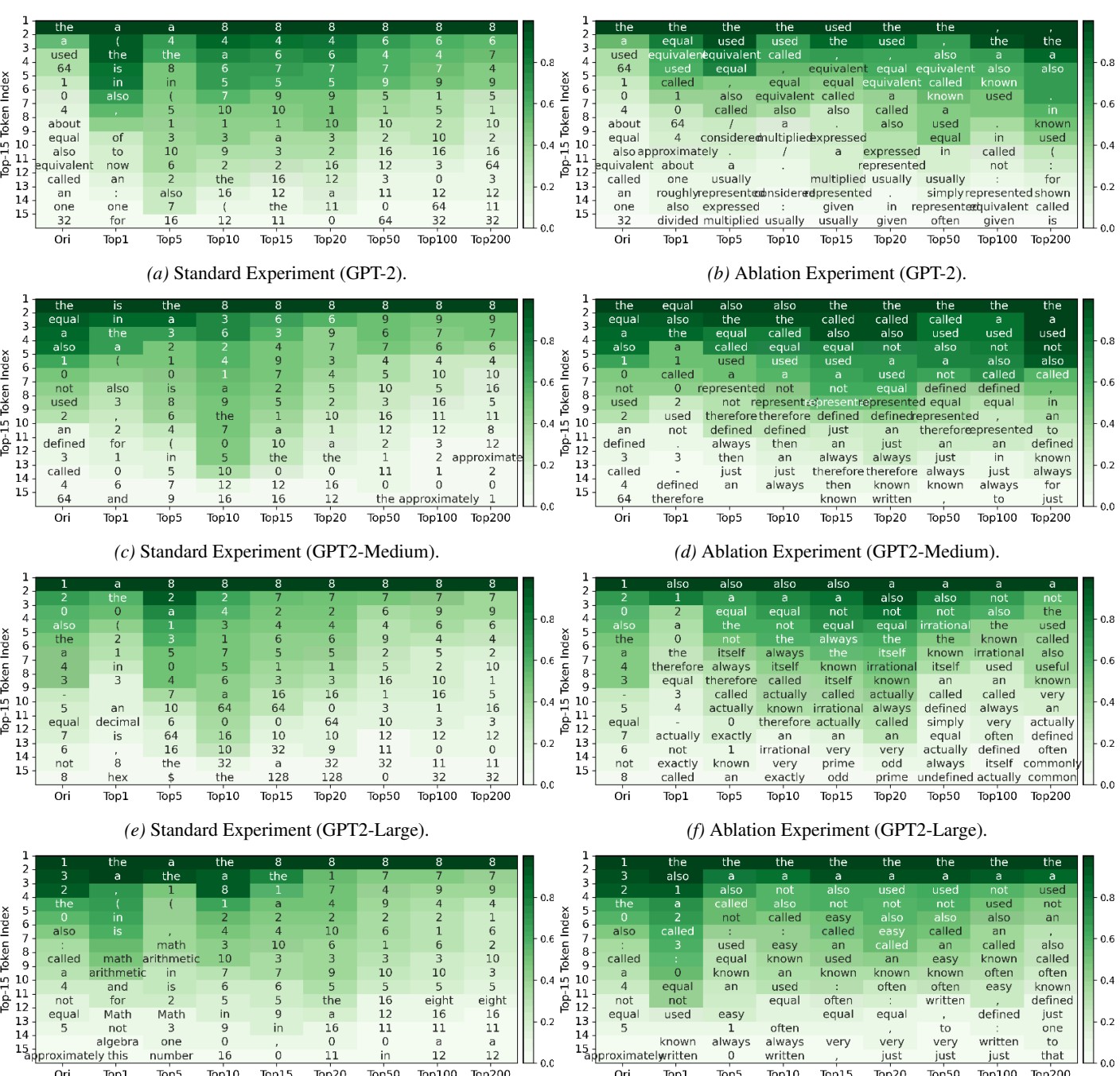

*(a)* Standard Experiment (GPT-2).

*(b)* Ablation Experiment (GPT-2).

*(c)* Standard Experiment (GPT2-Medium).

*(d)* Ablation Experiment (GPT2-Medium).

*(e)* Standard Experiment (GPT2-Large).

*(f)* Ablation Experiment (GPT2-Large).

*(g)* Standard Experiment (GPT2-XL).

*(h)* Ablation Experiment (GPT2-XL).

*Figure 20.* **Reconstruction of Middle-to-Late MLP Layers via Pathways within Top-$k$ High-Contribution Subspaces: Standard vs. Ablation Studies.** This figure illustrates a case study using the prompt "*The square root of 64 is*" with the target token " *8*". We visualize the top-15 predicted tokens across GPT-2 variants under standard and ablation settings.

**Case 4: Prompt "The dogs in the park" with Target Token " are".**

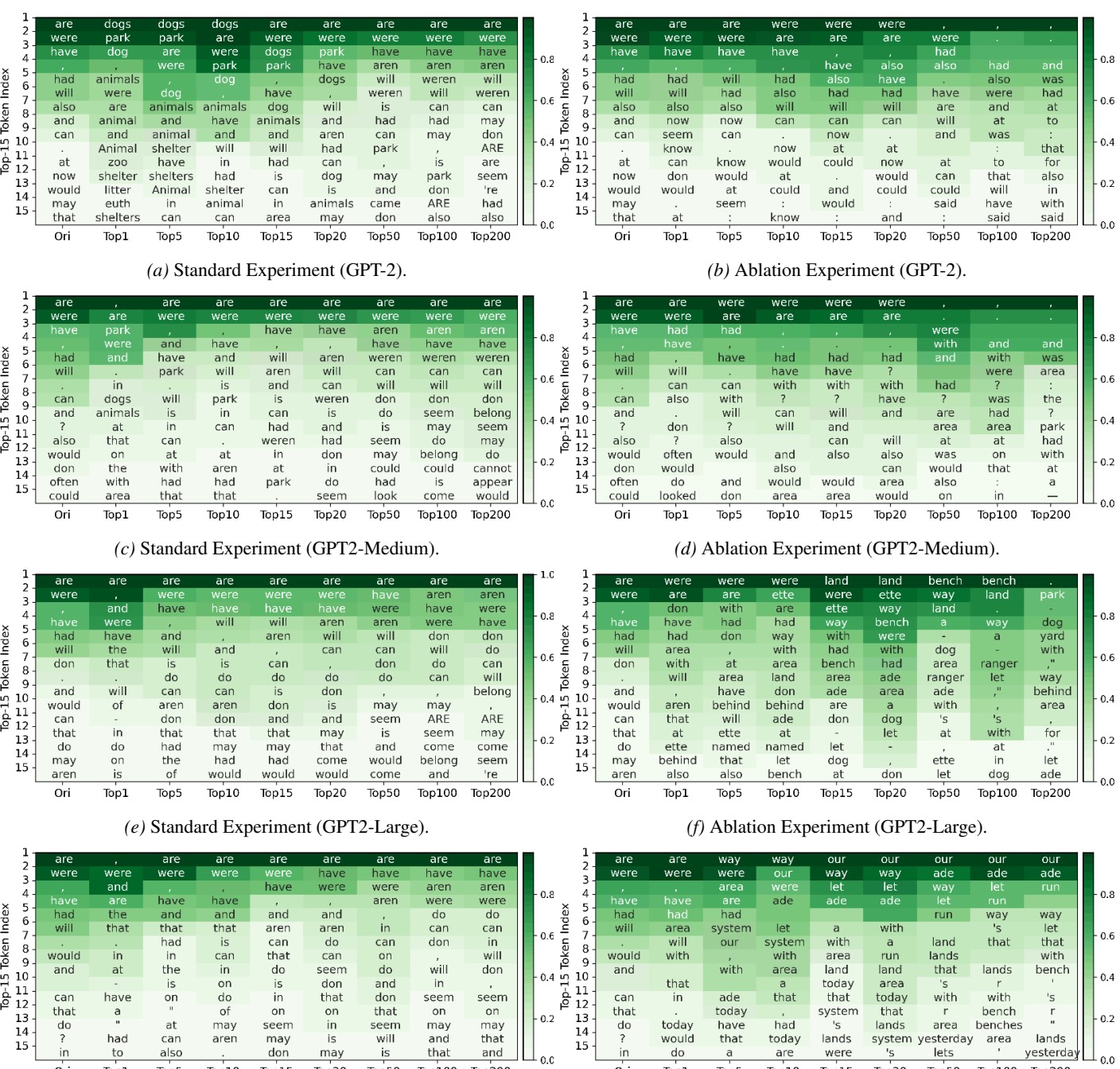

*(a)* Standard Experiment (GPT-2).

*(b)* Ablation Experiment (GPT-2).

*(c)* Standard Experiment (GPT2-Medium).

*(d)* Ablation Experiment (GPT2-Medium).

*(e)* Standard Experiment (GPT2-Large).

*(f)* Ablation Experiment (GPT2-Large).

*(g)* Standard Experiment (GPT2-XL).

*(h)* Ablation Experiment (GPT2-XL).

*Figure 21.* **Reconstruction of Middle-to-Late MLP Layers via Pathways within Top-$k$ High-Contribution Subspaces: Standard vs. Ablation Studies.** This figure illustrates a case study using the prompt "*The dogs in the park*" with the target token " *are*". We visualize the top-15 predicted tokens across GPT-2 variants under standard and ablation settings.

**Case 5: Prompt "The capital of Germany is" with Target Token " Berlin".**

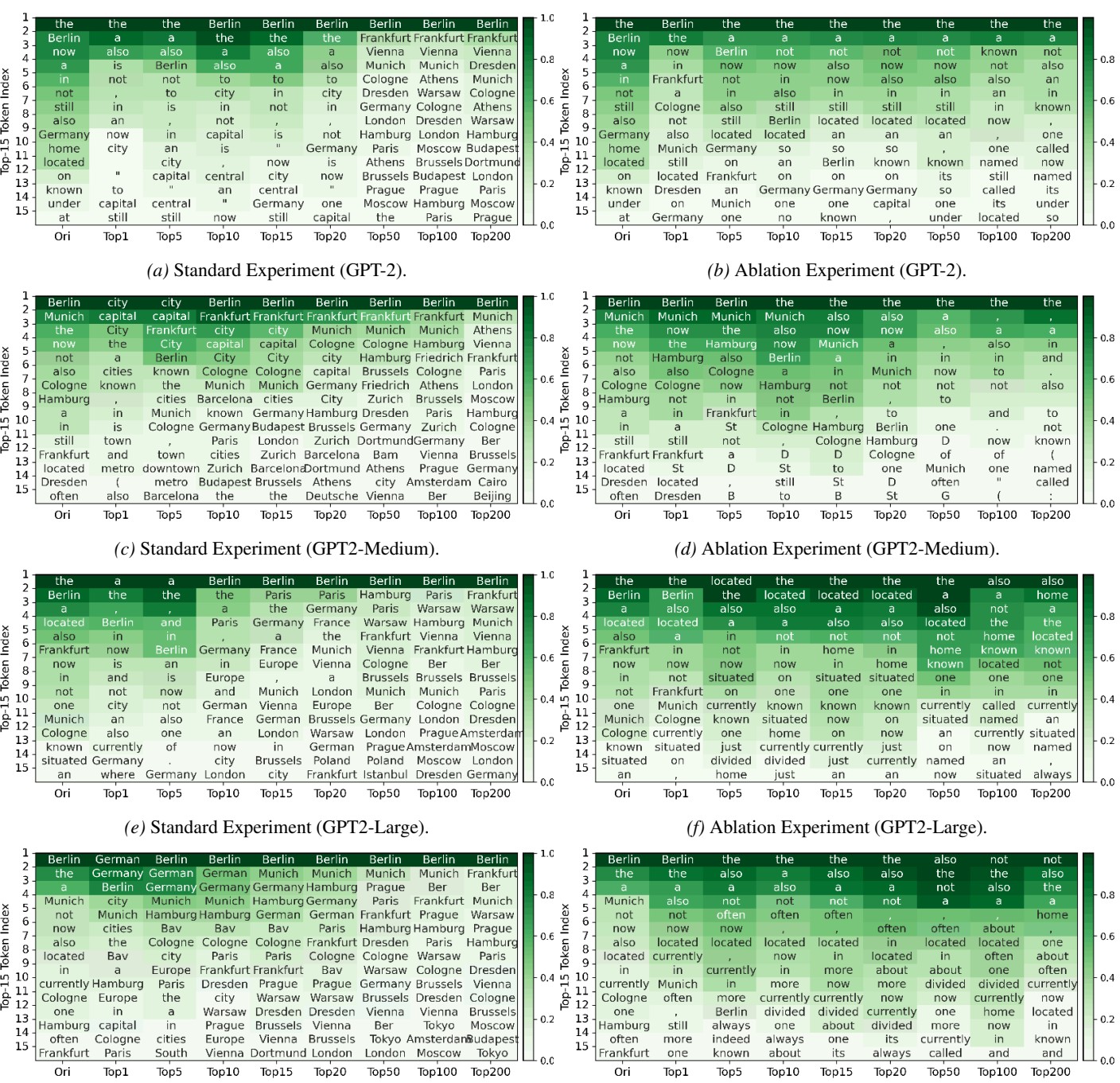

*(a)* Standard Experiment (GPT-2).

*(b)* Ablation Experiment (GPT-2).

*(c)* Standard Experiment (GPT2-Medium).

*(d)* Ablation Experiment (GPT2-Medium).

*(e)* Standard Experiment (GPT2-Large).

*(f)* Ablation Experiment (GPT2-Large).

*(g)* Standard Experiment (GPT2-XL).

*(h)* Ablation Experiment (GPT2-XL).

*Figure 22.* **Reconstruction of Middle-to-Late MLP Layers via Pathways within Top-$k$ High-Contribution Subspaces: Standard vs. Ablation Studies.** This figure illustrates a case study using the prompt "*The capital of Germany is*" with the target token " *Berlin*". We visualize the top-15 predicted tokens across GPT-2 variants under standard and ablation settings.

**Case 6: Prompt "Water boils at 100 degrees" with Target Token " Celsius".**

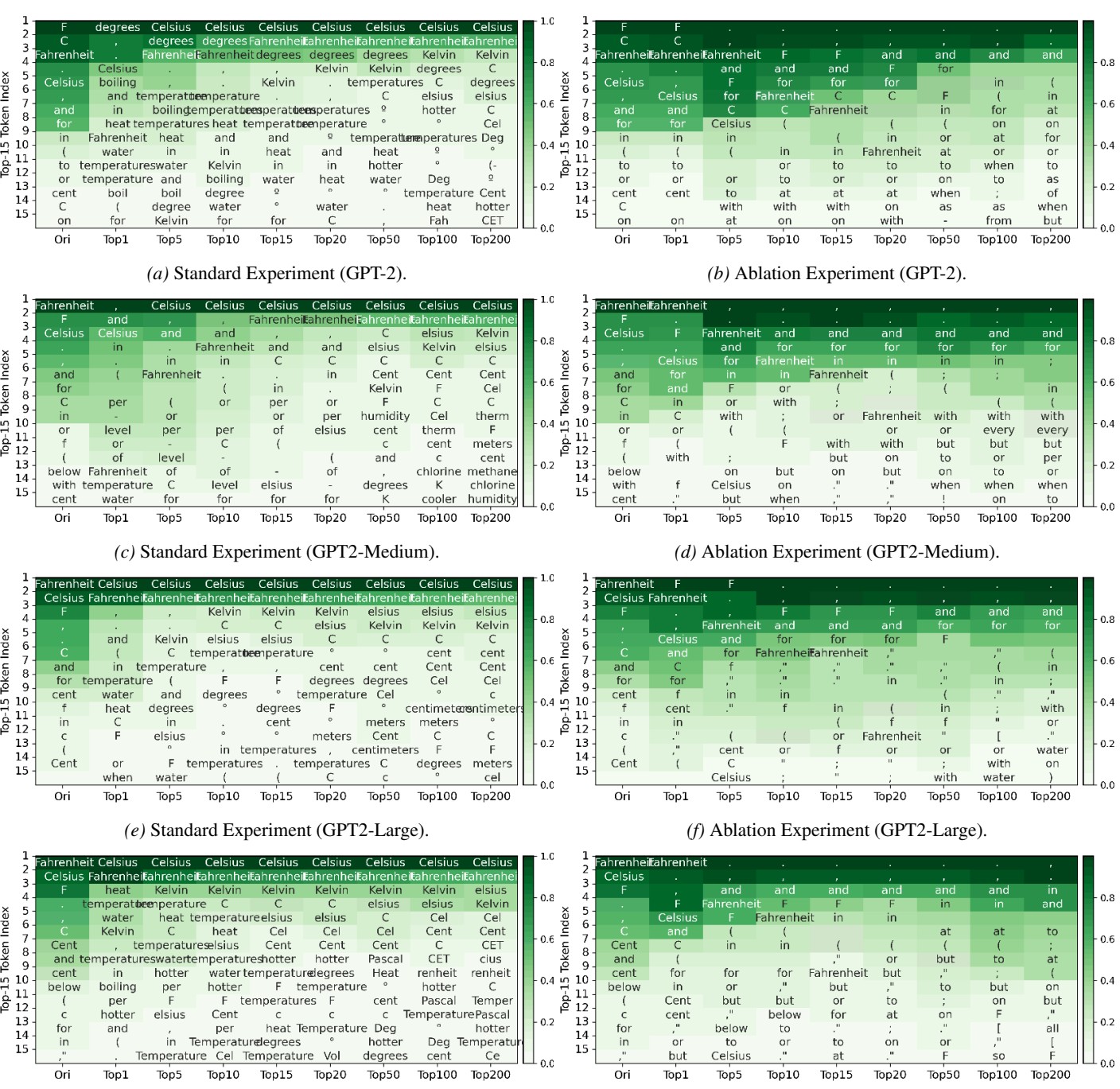

*(a)* Standard Experiment (GPT-2).

*(b)* Ablation Experiment (GPT-2).

*(c)* Standard Experiment (GPT2-Medium).

*(d)* Ablation Experiment (GPT2-Medium).

*(e)* Standard Experiment (GPT2-Large).

*(f)* Ablation Experiment (GPT2-Large).

*(g)* Standard Experiment (GPT2-XL).

*(h)* Ablation Experiment (GPT2-XL).

*Figure 23.* **Reconstruction of Middle-to-Late MLP Layers via Pathways within Top-$k$ High-Contribution Subspaces: Standard vs. Ablation Studies.** This figure illustrates a case study using the prompt "*Water boils at 100 degrees*" with the target token " *Celsius*". We visualize the top-15 predicted tokens across GPT-2 variants under standard and ablation settings.

**Case 7: Prompt "Code snippet: for char in 'hello': print" with Target Token " char".**

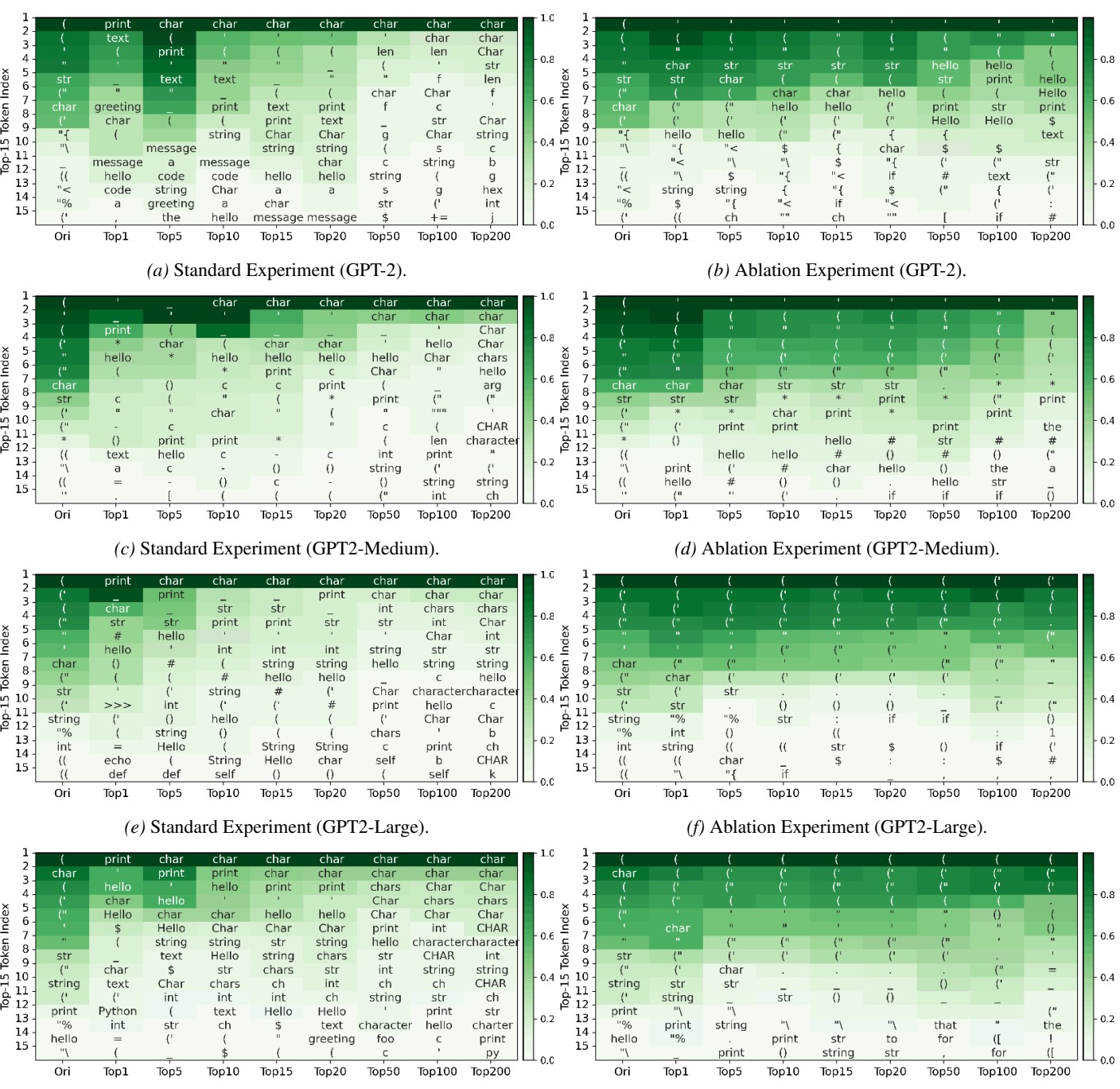

*Figure 24.* **Reconstruction of Middle-to-Late MLP Layers via Pathways within Top-$k$ High-Contribution Subspaces: Standard vs. Ablation Studies.** This figure illustrates a case study using the prompt "*Code snippet: for char in 'hello': print*" with the target token " char". We visualize the top-15 predicted tokens across GPT-2 variants under standard and ablation settings.

**Case 8: Prompt "John and Mary went to the" with Target Token " church".**

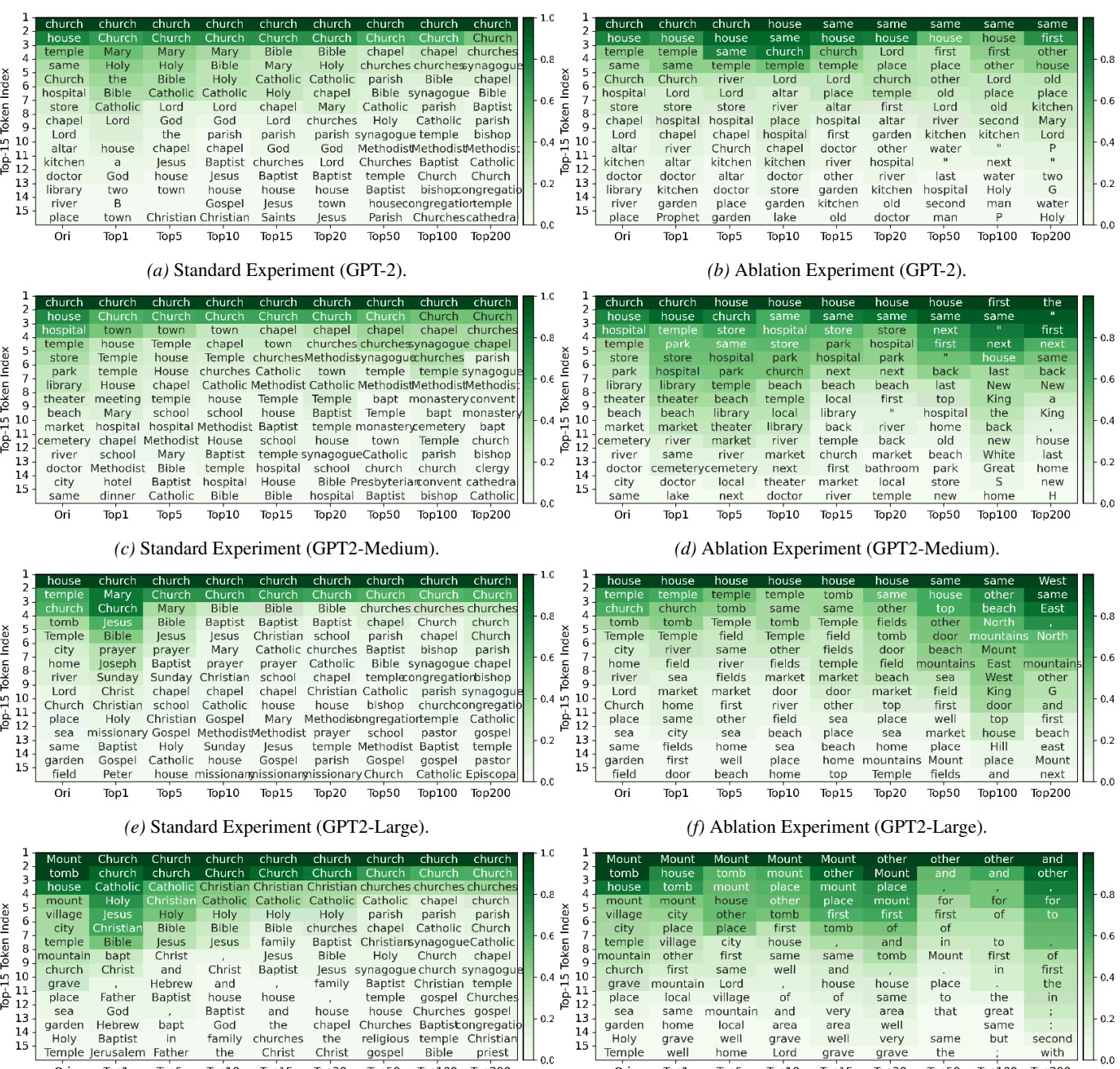

*(a)* Standard Experiment (GPT-2).

*(b)* Ablation Experiment (GPT-2).

*(c)* Standard Experiment (GPT2-Medium).

*(d)* Ablation Experiment (GPT2-Medium).

*(e)* Standard Experiment (GPT2-Large).

*(f)* Ablation Experiment (GPT2-Large).

*(g)* Standard Experiment (GPT2-XL).

*(h)* Ablation Experiment (GPT2-XL).

*Figure 25.* **Reconstruction of Middle-to-Late MLP Layers via Pathways within Top-$k$ High-Contribution Subspaces: Standard vs. Ablation Studies.** This figure illustrates a case study using the prompt "*John and Mary went to the*" with the target token " *church*". We visualize the top-15 predicted tokens across GPT-2 variants under standard and ablation settings.

**Case 9: Prompt "She didn't see the sign, so she" with Target Token " didn".**

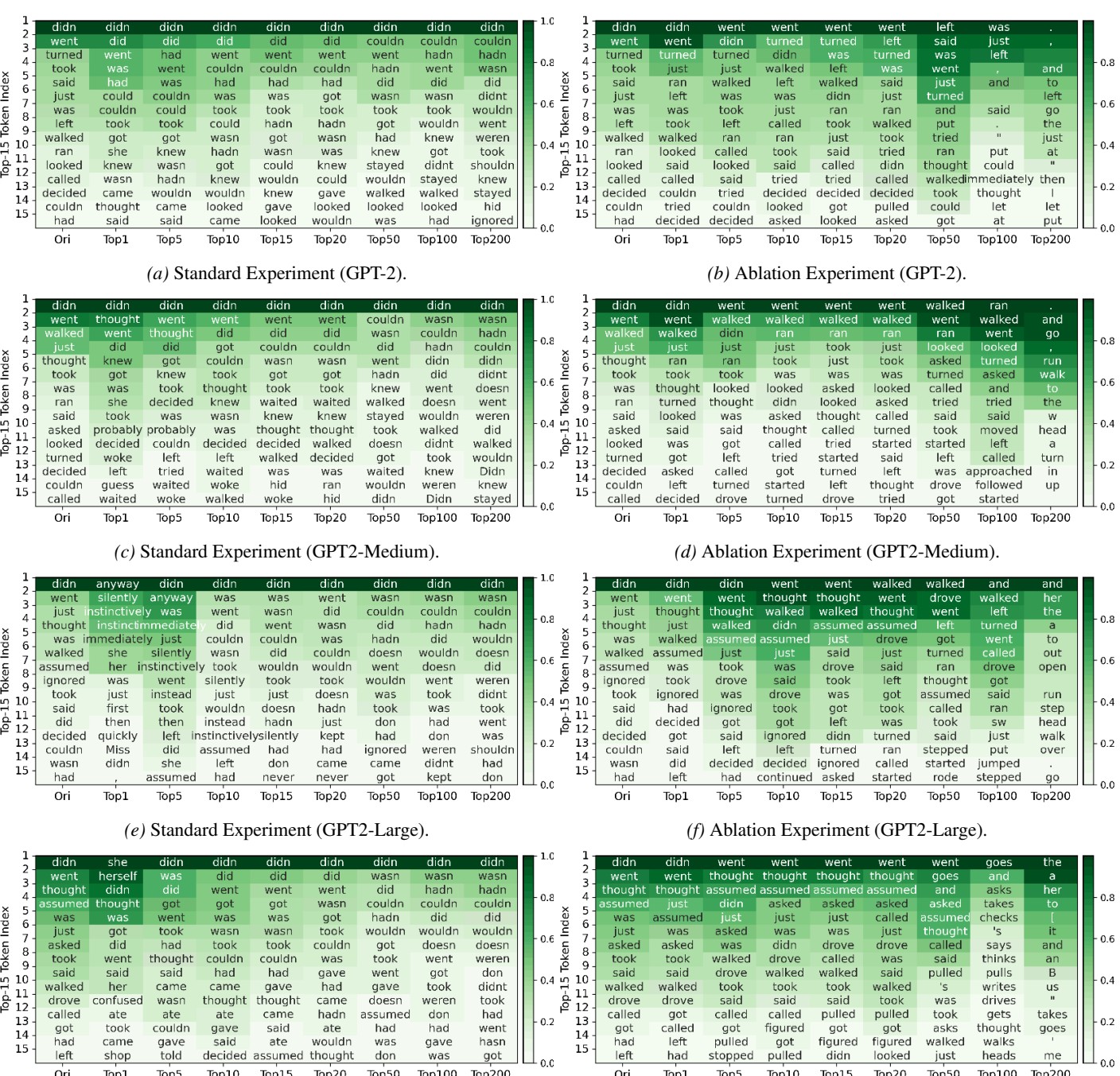

*(a)* Standard Experiment (GPT-2).

*(b)* Ablation Experiment (GPT-2).

*(c)* Standard Experiment (GPT2-Medium).

*(d)* Ablation Experiment (GPT2-Medium).

*(e)* Standard Experiment (GPT2-Large).

*(f)* Ablation Experiment (GPT2-Large).

*(g)* Standard Experiment (GPT2-XL).

*(h)* Ablation Experiment (GPT2-XL).

*Figure 26.* **Reconstruction of Middle-to-Late MLP Layers via Pathways within Top-*k* High-Contribution Subspaces: Standard vs. Ablation Studies.** This figure illustrates a case study using the prompt "*She didn't see the sign, so she*" with the target token " *didn*". We visualize the top-15 predicted tokens across GPT-2 variants under standard and ablation settings.

**Case 10: Prompt "She quickly ran to the" with Target Token " door".**

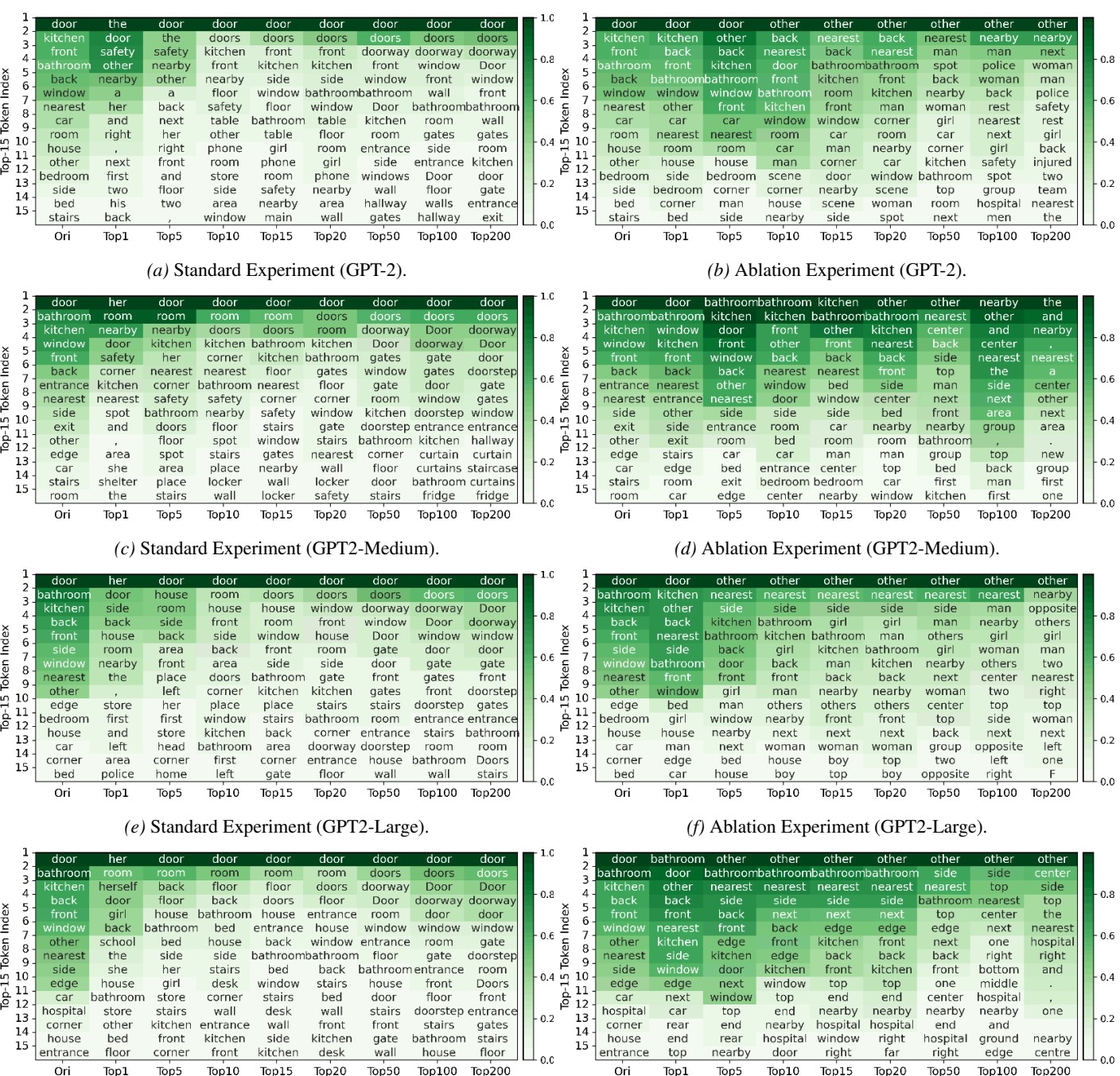

*(a)* Standard Experiment (GPT-2).

*(b)* Ablation Experiment (GPT-2).

*(c)* Standard Experiment (GPT2-Medium).

*(d)* Ablation Experiment (GPT2-Medium).

*(e)* Standard Experiment (GPT2-Large).

*(f)* Ablation Experiment (GPT2-Large).

*(g)* Standard Experiment (GPT2-XL).

*(h)* Ablation Experiment (GPT2-XL).

*Figure 27.* **Reconstruction of Middle-to-Late MLP Layers via Pathways within Top-$k$ High-Contribution Subspaces: Standard vs. Ablation Studies.** This figure illustrates a case study using the prompt "*She quickly ran to the*" with the target token " *door*". We visualize the top-15 predicted tokens across GPT-2 variants under standard and ablation settings.

**Case 11: Prompt "The sun rises in the" with Target Token " east".**

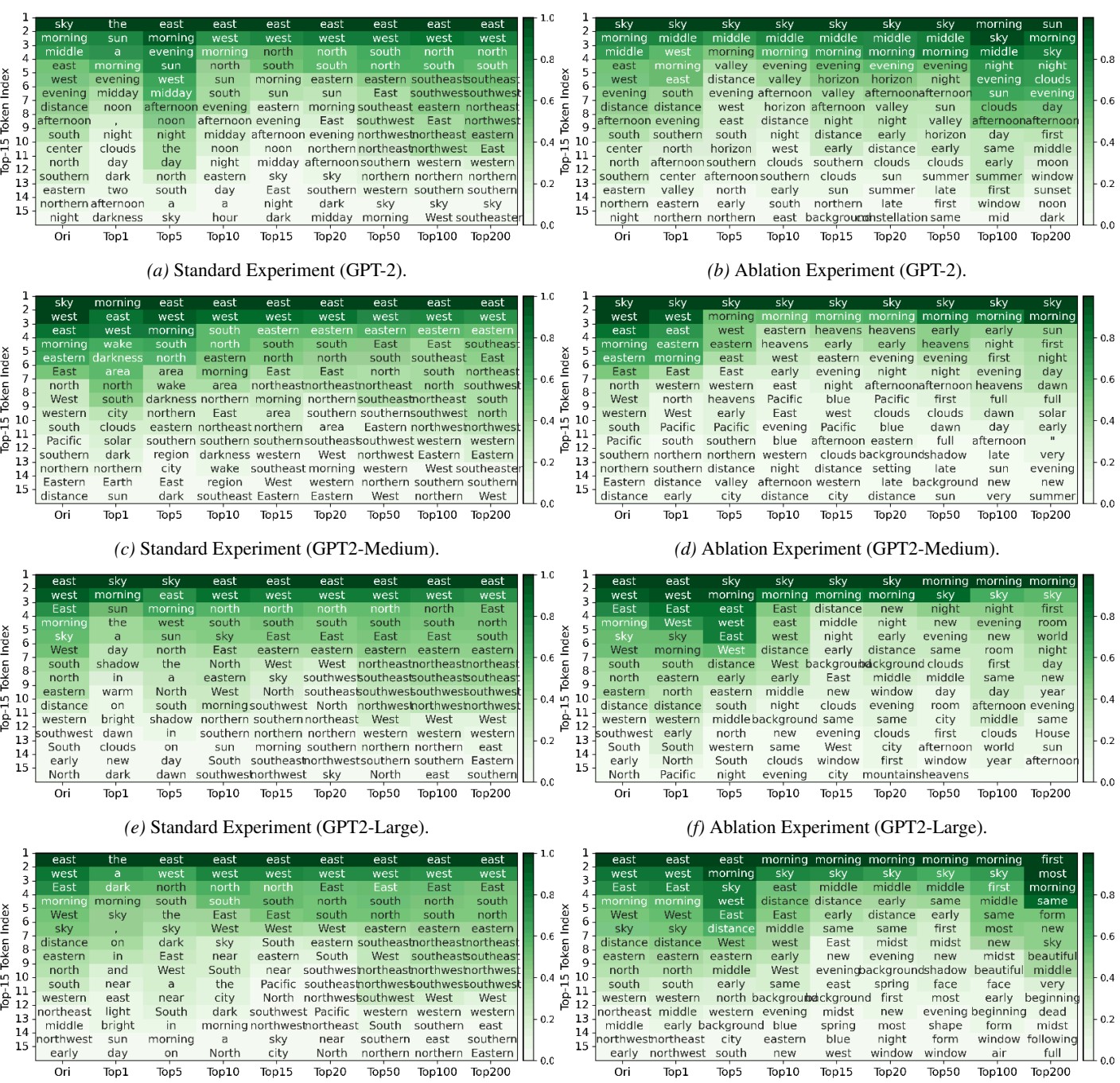

*(a)* Standard Experiment (GPT-2).

*(b)* Ablation Experiment (GPT-2).

*(c)* Standard Experiment (GPT2-Medium).

*(d)* Ablation Experiment (GPT2-Medium).

*(e)* Standard Experiment (GPT2-Large).

*(f)* Ablation Experiment (GPT2-Large).

*(g)* Standard Experiment (GPT2-XL).

*(h)* Ablation Experiment (GPT2-XL).

*Figure 28.* **Reconstruction of Middle-to-Late MLP Layers via Pathways within Top-$k$ High-Contribution Subspaces: Standard vs. Ablation Studies.** This figure illustrates a case study using the prompt "*The sun rises in the*" with the target token " *east*". We visualize the top-15 predicted tokens across GPT-2 variants under standard and ablation settings.

**Case 12: Prompt "The food was not bad, it was" with Target Token " good".**

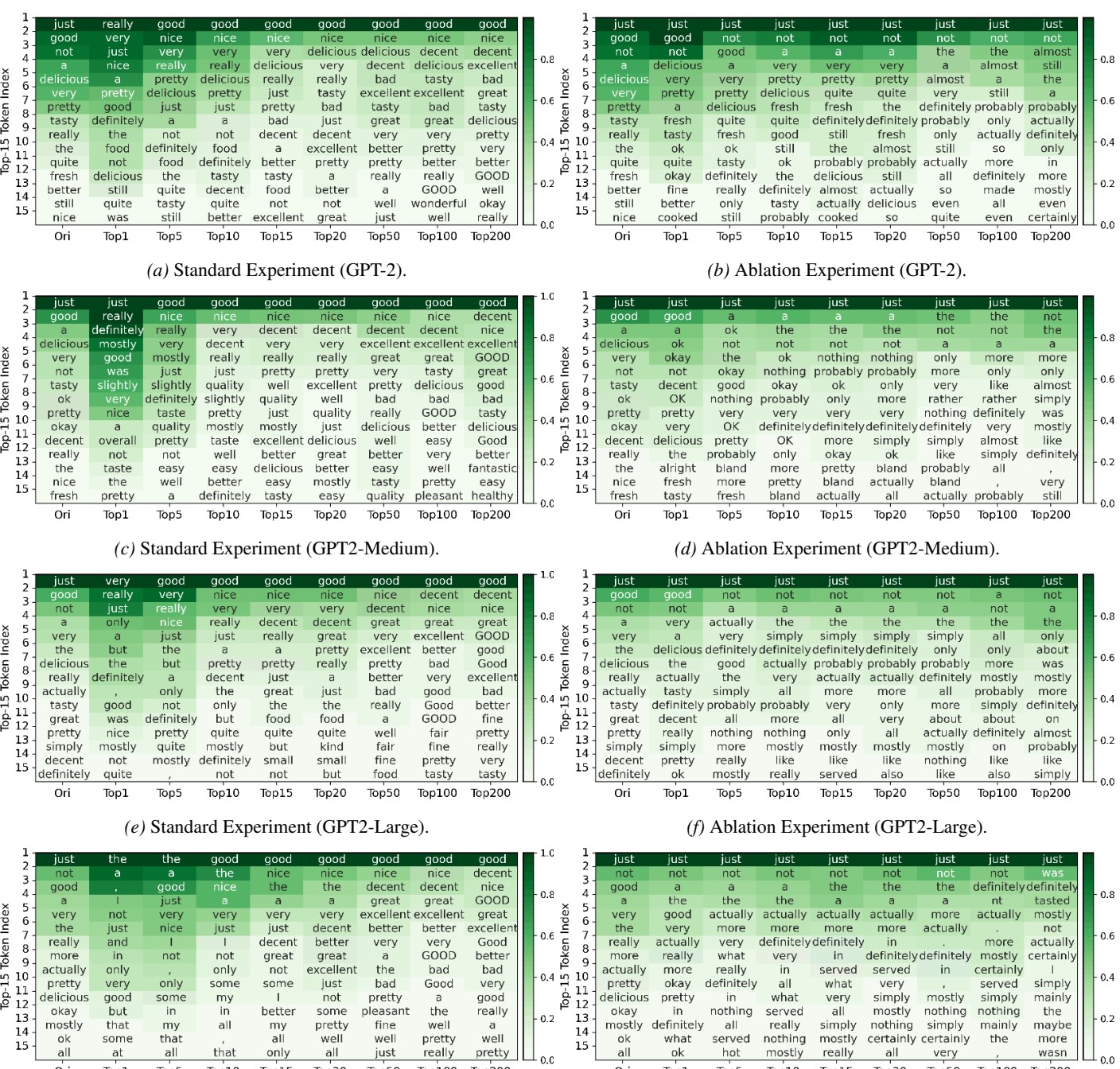

*(a)* Standard Experiment (GPT-2).        *(b)* Ablation Experiment (GPT-2).

*(c)* Standard Experiment (GPT2-Medium).      *(d)* Ablation Experiment (GPT2-Medium).

*(e)* Standard Experiment (GPT2-Large).      *(f)* Ablation Experiment (GPT2-Large).

*(g)* Standard Experiment (GPT2-XL).       *(h)* Ablation Experiment (GPT2-XL).

*Figure 29.* **Reconstruction of Middle-to-Late MLP Layers via Pathways within Top-$k$ High-Contribution Subspaces: Standard vs. Ablation Studies.** This figure illustrates a case study using the prompt "*The food was not bad, it was*" with the target token " *good*". We visualize the top-15 predicted tokens across GPT-2 variants under standard and ablation settings.

**Case 13: Prompt "The boy who cried wolf finally" with Target Token " got".**

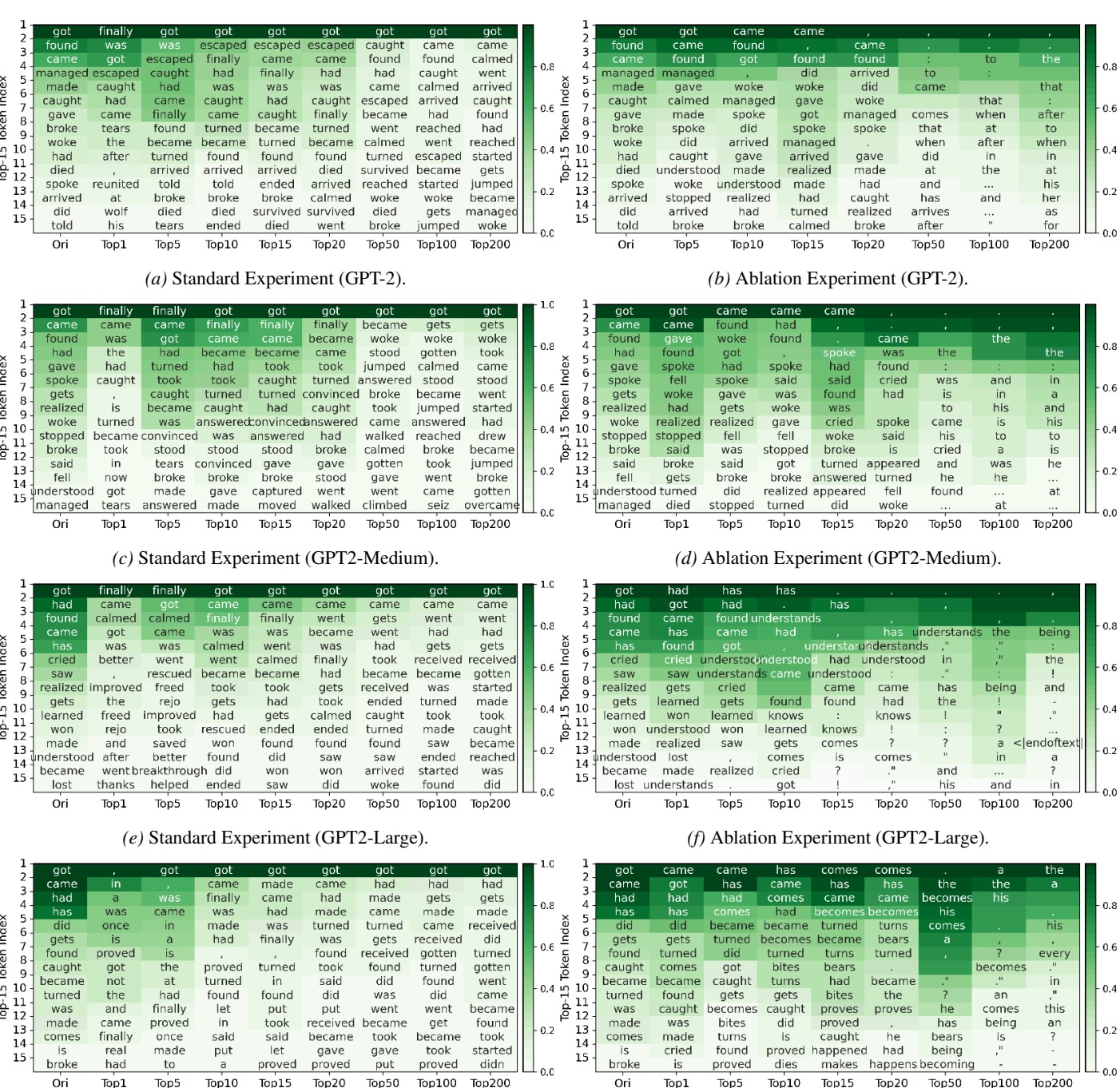

*(a)* Standard Experiment (GPT-2).   *(b)* Ablation Experiment (GPT-2).

*(c)* Standard Experiment (GPT2-Medium).   *(d)* Ablation Experiment (GPT2-Medium).

*(e)* Standard Experiment (GPT2-Large).   *(f)* Ablation Experiment (GPT2-Large).

*(g)* Standard Experiment (GPT2-XL).   *(h)* Ablation Experiment (GPT2-XL).

*Figure 30.* **Reconstruction of Middle-to-Late MLP Layers via Pathways within Top-$k$ High-Contribution Subspaces: Standard vs. Ablation Studies.** This figure illustrates a case study using the prompt "*The boy who cried wolf finally*" with the target token " *got*". We visualize the top-15 predicted tokens across GPT-2 variants under standard and ablation settings.

**Case 14: Prompt "The committee decided to postpone the" with Target Token " vote".**

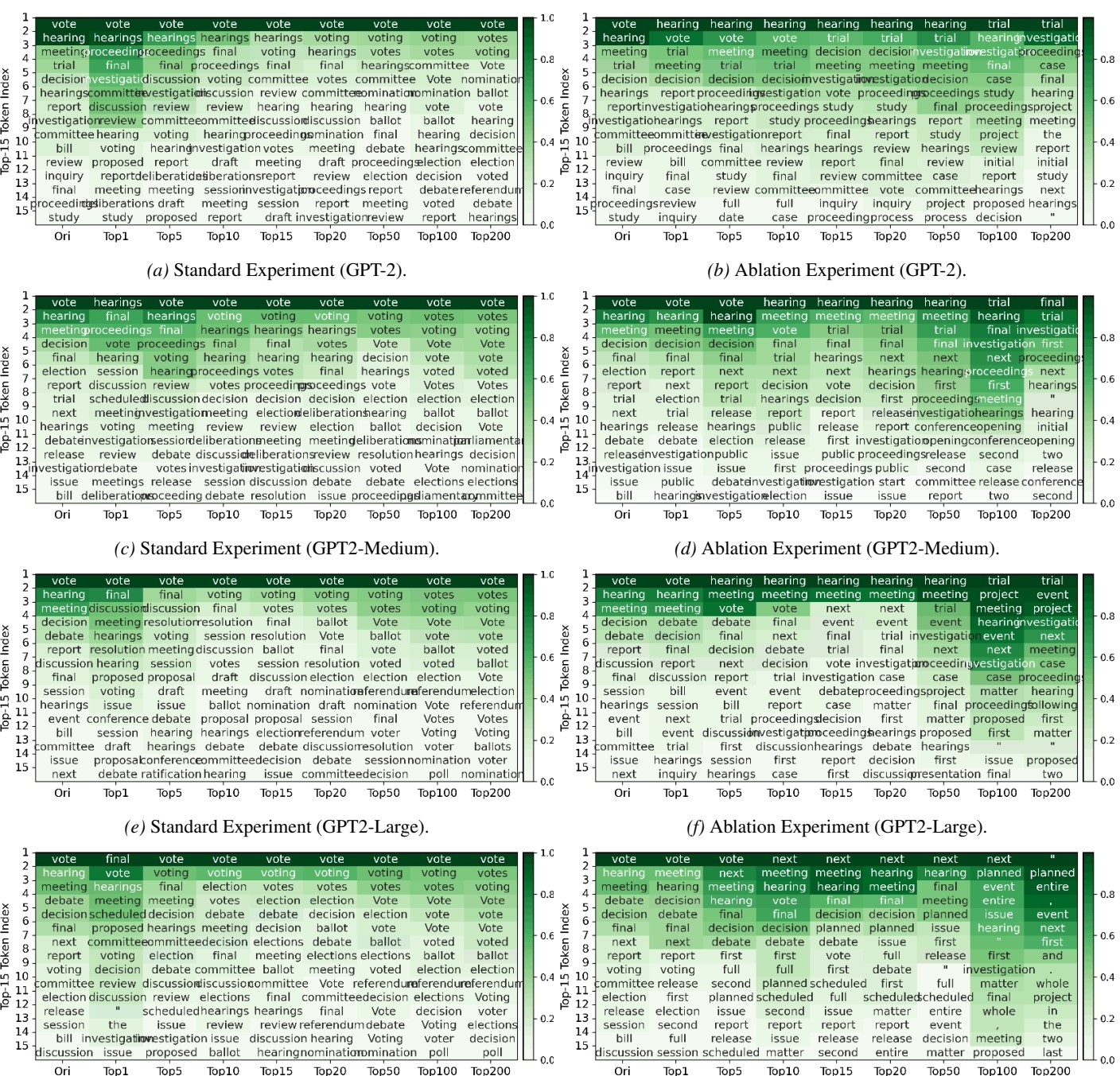

*(a)* Standard Experiment (GPT-2).

*(b)* Ablation Experiment (GPT-2).

*(c)* Standard Experiment (GPT2-Medium).

*(d)* Ablation Experiment (GPT2-Medium).

*(e)* Standard Experiment (GPT2-Large).

*(f)* Ablation Experiment (GPT2-Large).

*(g)* Standard Experiment (GPT2-XL).

*(h)* Ablation Experiment (GPT2-XL).

*Figure 31.* **Reconstruction of Middle-to-Late MLP Layers via Pathways within Top-$k$ High-Contribution Subspaces: Standard vs. Ablation Studies.** This figure illustrates a case study using the prompt "*The committee decided to postpone the*" with the target token " *vote*". We visualize the top-15 predicted tokens across GPT-2 variants under standard and ablation settings.

**Case 15: Prompt "During the thunderstorm, the lights" with Target Token " of".**

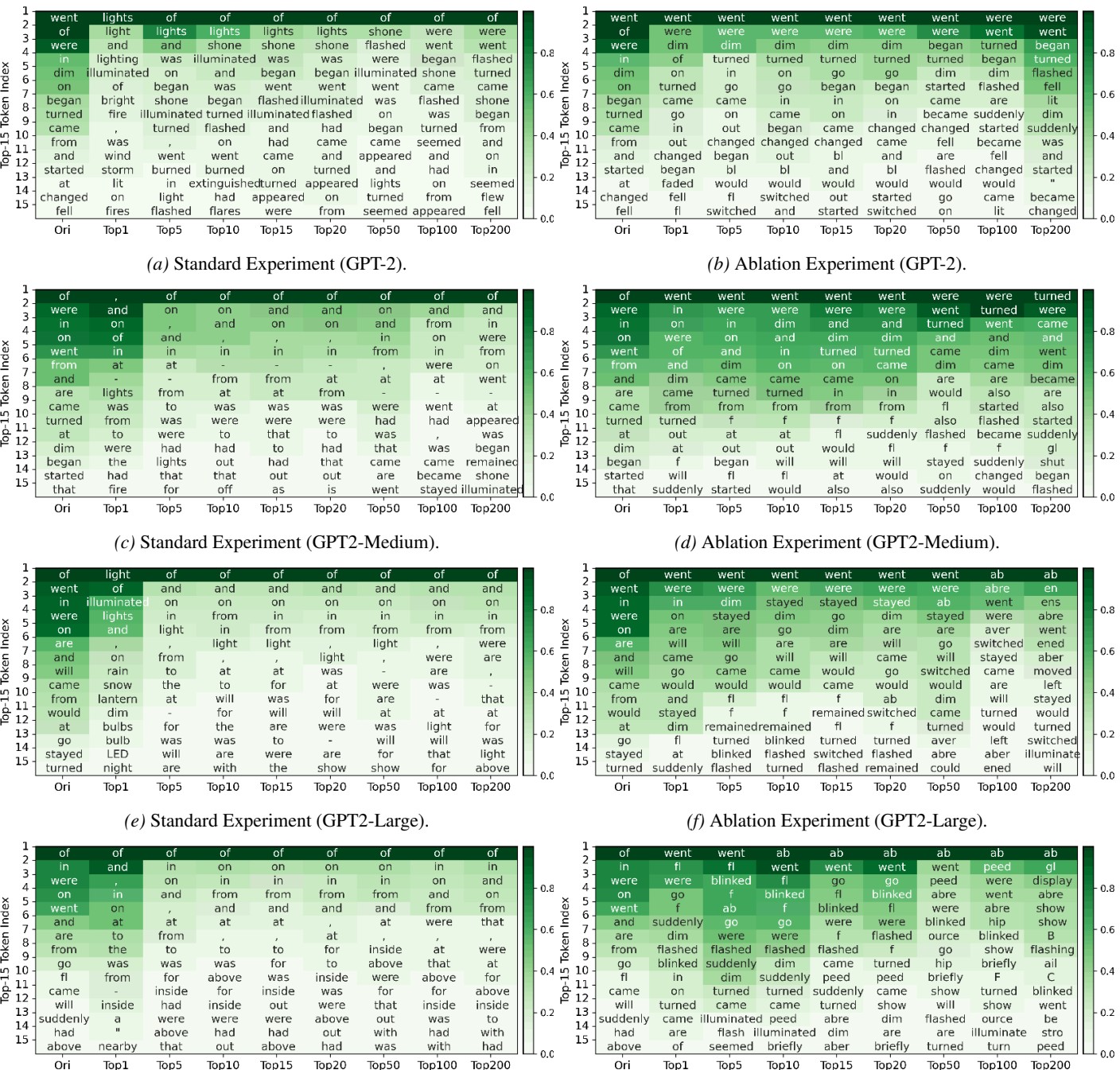

*(a)* Standard Experiment (GPT-2).

*(b)* Ablation Experiment (GPT-2).

*(c)* Standard Experiment (GPT2-Medium).

*(d)* Ablation Experiment (GPT2-Medium).

*(e)* Standard Experiment (GPT2-Large).

*(f)* Ablation Experiment (GPT2-Large).

*(g)* Standard Experiment (GPT2-XL).

*(h)* Ablation Experiment (GPT2-XL).

*Figure 32.* **Reconstruction of Middle-to-Late MLP Layers via Pathways within Top-$k$ High-Contribution Subspaces: Standard vs. Ablation Studies.** This figure illustrates a case study using the prompt "*During the thunderstorm, the lights*" with the target token " *of*". We visualize the top-15 predicted tokens across GPT-2 variants under standard and ablation settings.

**Case 16: Prompt "The movie was thrilling but also" with Target Token " sad".**

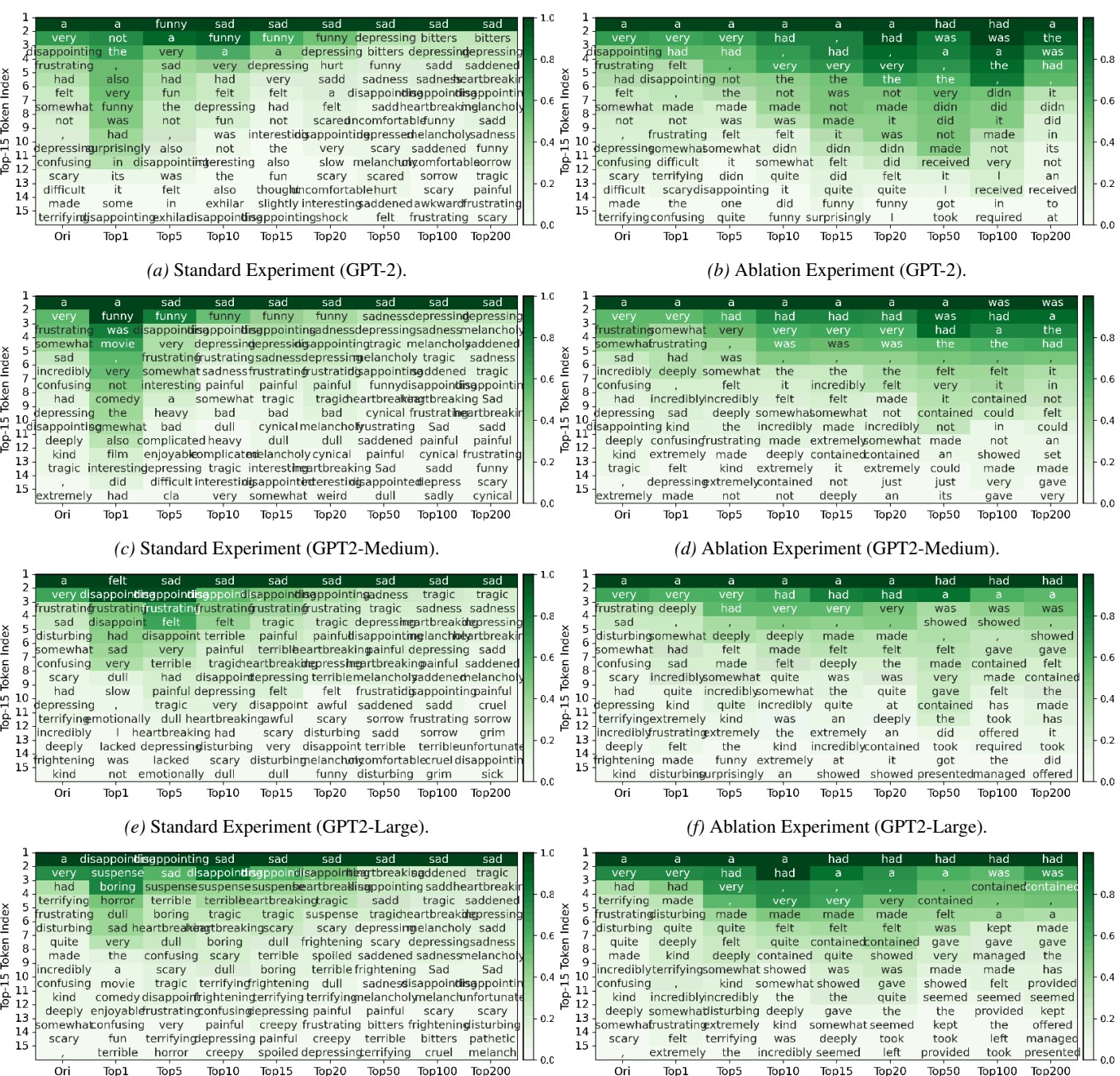

*(a)* Standard Experiment (GPT-2).      *(b)* Ablation Experiment (GPT-2).

*(c)* Standard Experiment (GPT2-Medium).      *(d)* Ablation Experiment (GPT2-Medium).

*(e)* Standard Experiment (GPT2-Large).      *(f)* Ablation Experiment (GPT2-Large).

*(g)* Standard Experiment (GPT2-XL).      *(h)* Ablation Experiment (GPT2-XL).

*Figure 33.* **Reconstruction of Middle-to-Late MLP Layers via Pathways within Top-*k* High-Contribution Subspaces: Standard vs. Ablation Studies.** This figure illustrates a case study using the prompt "*The movie was thrilling but also*" with the target token " *sad*". We visualize the top-15 predicted tokens across GPT-2 variants under standard and ablation settings.

**Case 17: Prompt "The recipe calls for two cups of" with Target Token " sugar".**

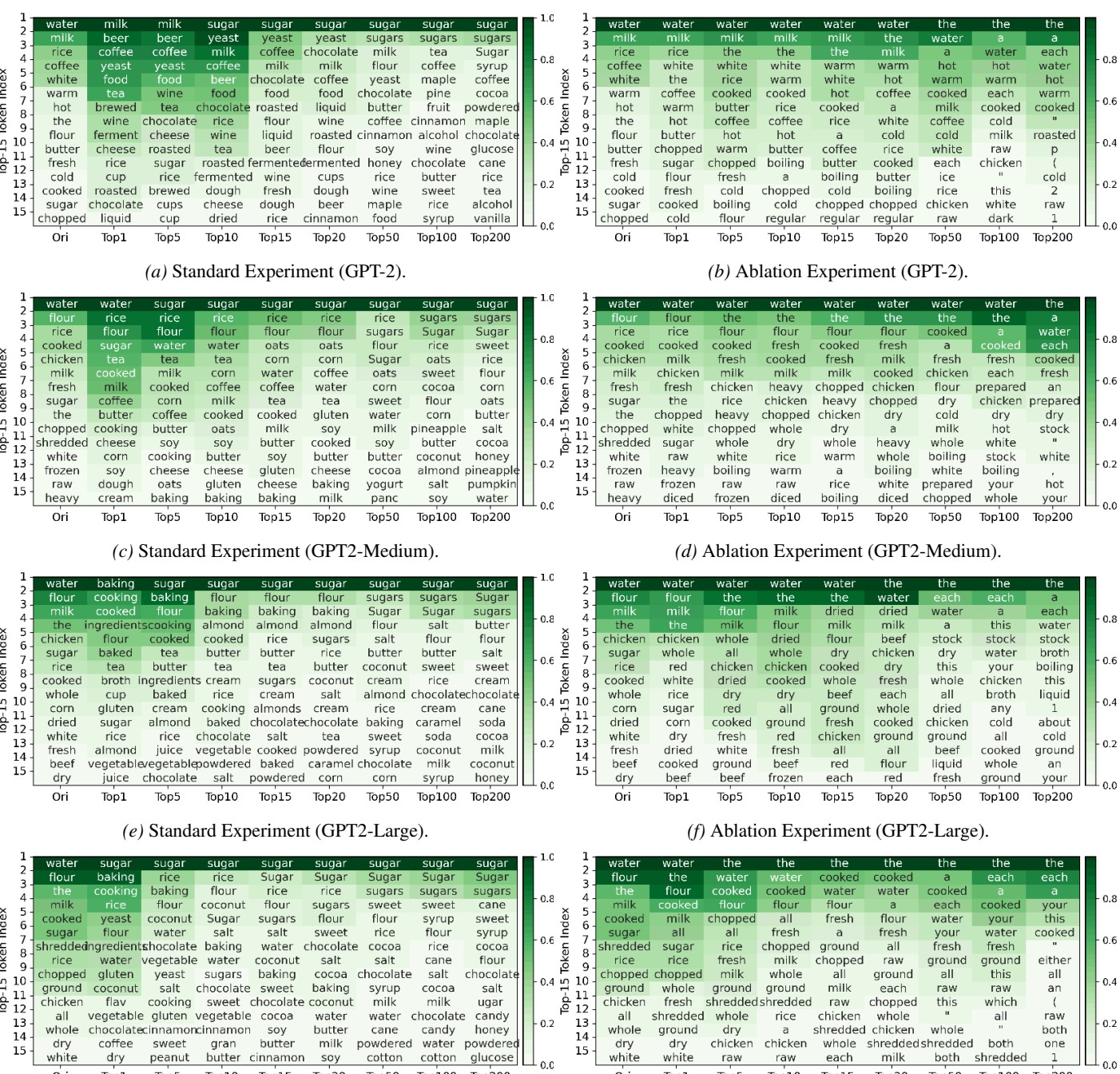

*(a)* Standard Experiment (GPT-2).

*(b)* Ablation Experiment (GPT-2).

*(c)* Standard Experiment (GPT2-Medium).

*(d)* Ablation Experiment (GPT2-Medium).

*(e)* Standard Experiment (GPT2-Large).

*(f)* Ablation Experiment (GPT2-Large).

*(g)* Standard Experiment (GPT2-XL).

*(h)* Ablation Experiment (GPT2-XL).

*Figure 34.* **Reconstruction of Middle-to-Late MLP Layers via Pathways within Top-$k$ High-Contribution Subspaces: Standard vs. Ablation Studies.** This figure illustrates a case study using the prompt "*The recipe calls for two cups of*" with the target token " *sugar*". We visualize the top-15 predicted tokens across GPT-2 variants under standard and ablation settings.

**Case 18: Prompt "To solve the equation, one must first" with Target Token " simplify".**

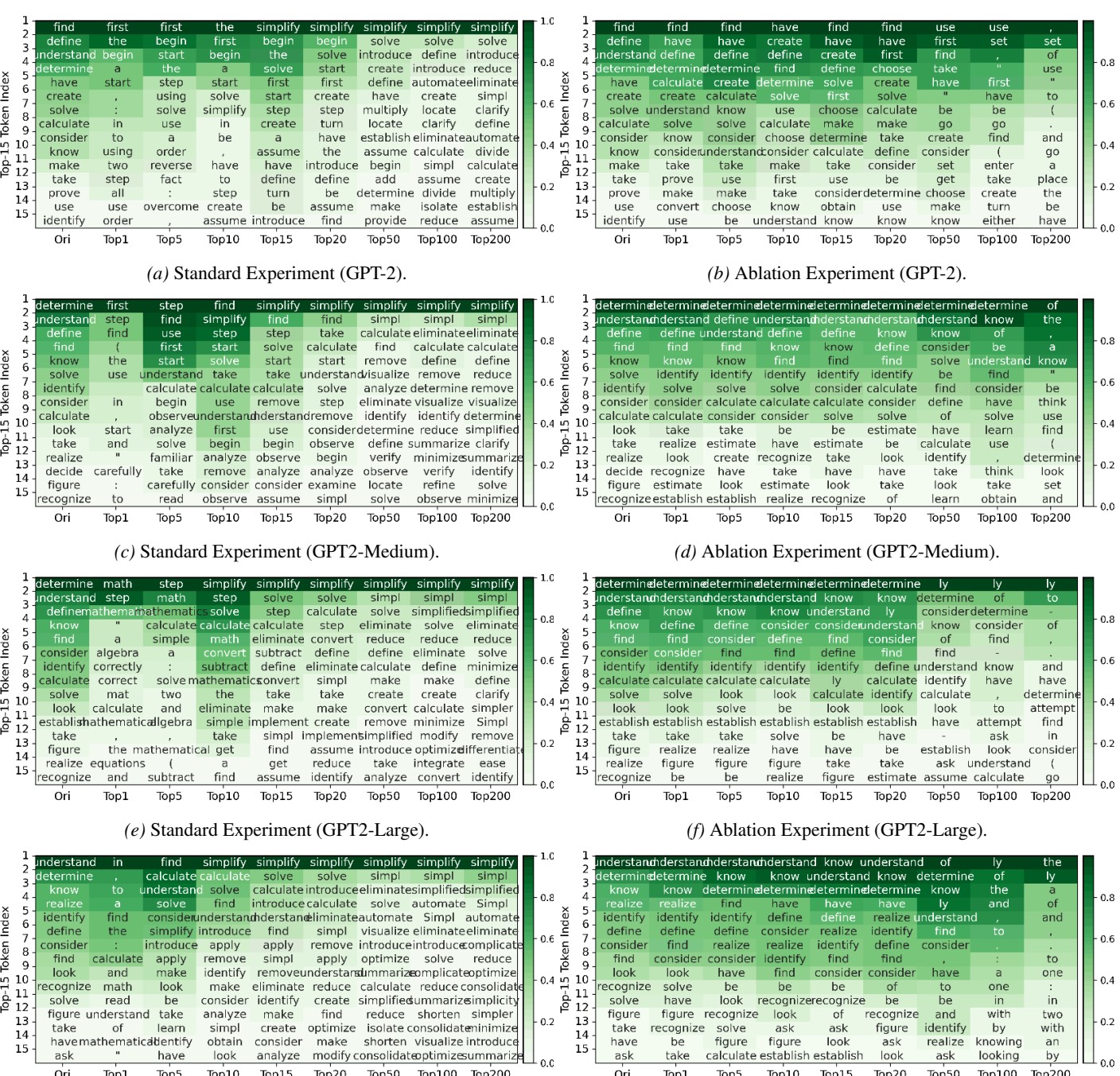

*(a)* Standard Experiment (GPT-2).

*(b)* Ablation Experiment (GPT-2).

*(c)* Standard Experiment (GPT2-Medium).

*(d)* Ablation Experiment (GPT2-Medium).

*(e)* Standard Experiment (GPT2-Large).

*(f)* Ablation Experiment (GPT2-Large).

*(g)* Standard Experiment (GPT2-XL).

*(h)* Ablation Experiment (GPT2-XL).

*Figure 35.* **Reconstruction of Middle-to-Late MLP Layers via Pathways within Top-$k$ High-Contribution Subspaces: Standard vs. Ablation Studies.** This figure illustrates a case study using the prompt "*To solve the equation, one must first*" with the target token " *simplify*". We visualize the top-15 predicted tokens across GPT-2 variants under standard and ablation settings.

**Case 19: Prompt "Alice gave Bob the book because she" with Target Token " wanted".**

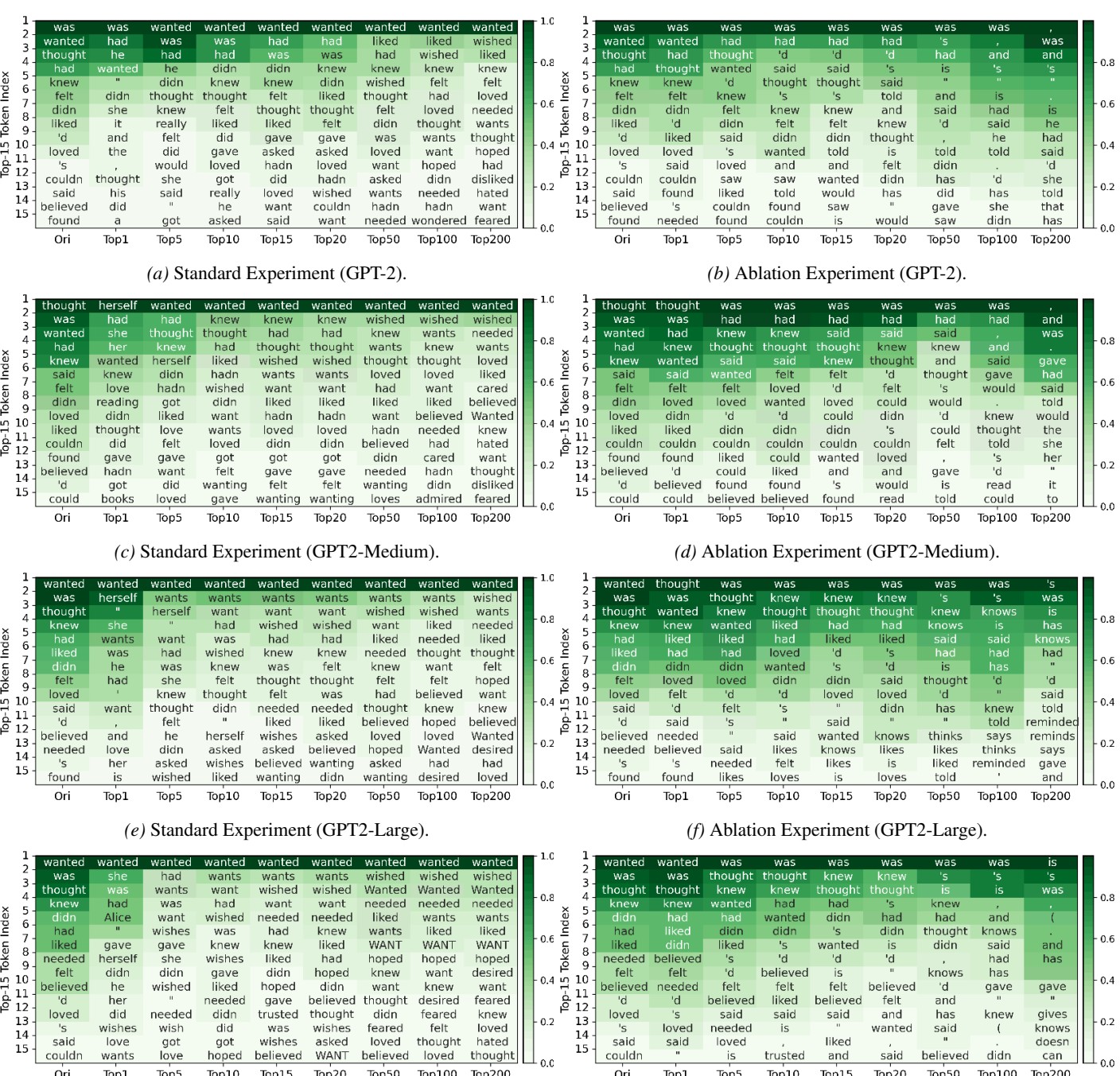

*(a)* Standard Experiment (GPT-2).

*(b)* Ablation Experiment (GPT-2).

*(c)* Standard Experiment (GPT2-Medium).

*(d)* Ablation Experiment (GPT2-Medium).

*(e)* Standard Experiment (GPT2-Large).

*(f)* Ablation Experiment (GPT2-Large).

*(g)* Standard Experiment (GPT2-XL).

*(h)* Ablation Experiment (GPT2-XL).

*Figure 36.* **Reconstruction of Middle-to-Late MLP Layers via Pathways within Top-$k$ High-Contribution Subspaces: Standard vs. Ablation Studies.** This figure illustrates a case study using the prompt "*Alice gave Bob the book because she*" with the target token " *wanted*". We visualize the top-15 predicted tokens across GPT-2 variants under standard and ablation settings.

**Case 20: Prompt "If I had studied harder, I" with Target Token " would".**

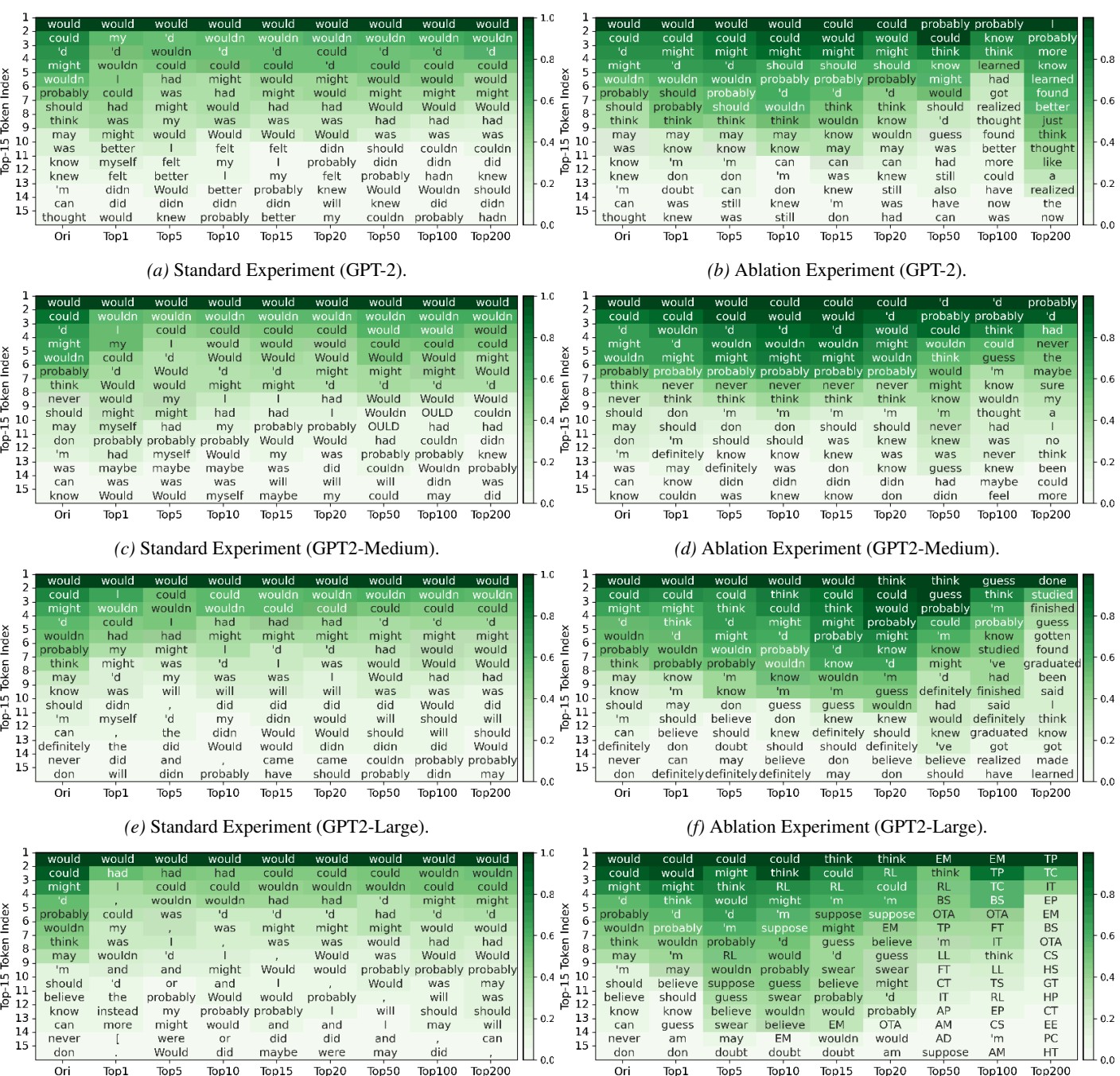

*(a)* Standard Experiment (GPT-2).

*(b)* Ablation Experiment (GPT-2).

*(c)* Standard Experiment (GPT2-Medium).

*(d)* Ablation Experiment (GPT2-Medium).

*(e)* Standard Experiment (GPT2-Large).

*(f)* Ablation Experiment (GPT2-Large).

*(g)* Standard Experiment (GPT2-XL).

*(h)* Ablation Experiment (GPT2-XL).

*Figure 37.* **Reconstruction of Middle-to-Late MLP Layers via Pathways within Top-$k$ High-Contribution Subspaces: Standard vs. Ablation Studies.** This figure illustrates a case study using the prompt "*If I had studied harder, I*" with the target token " *would*". We visualize the top-15 predicted tokens across GPT-2 variants under standard and ablation settings.

# I. Subspace-Level Analysis of Model Behavior and Lexical Preferences

## I.1. Case Study: Progressive Effect of Top-$k$ Subspaces

To elucidate the internal mechanisms of subspace pathways, we conduct a case study focused on positive semantics (e.g., "*the cat looks very*" → " *happy*"). We assess the fidelity and faithfulness of the identified pathways through reconstruction and ablation experiments. In Figure 38, we intervene on the MLP down-projection matrices across the middle-to-late layers of GPT-2 models of varying scales, and monitor the resulting changes in the target token's *rank*, *logit*, and *probability*. The result demonstrates that ablating the top-$k$ subspaces identified by SCA substantially degrades target prediction performance. Conversely, exclusively retaining these subspaces leads to a swift recovery of the target token, thereby verifying their pivotal role in the generation process.

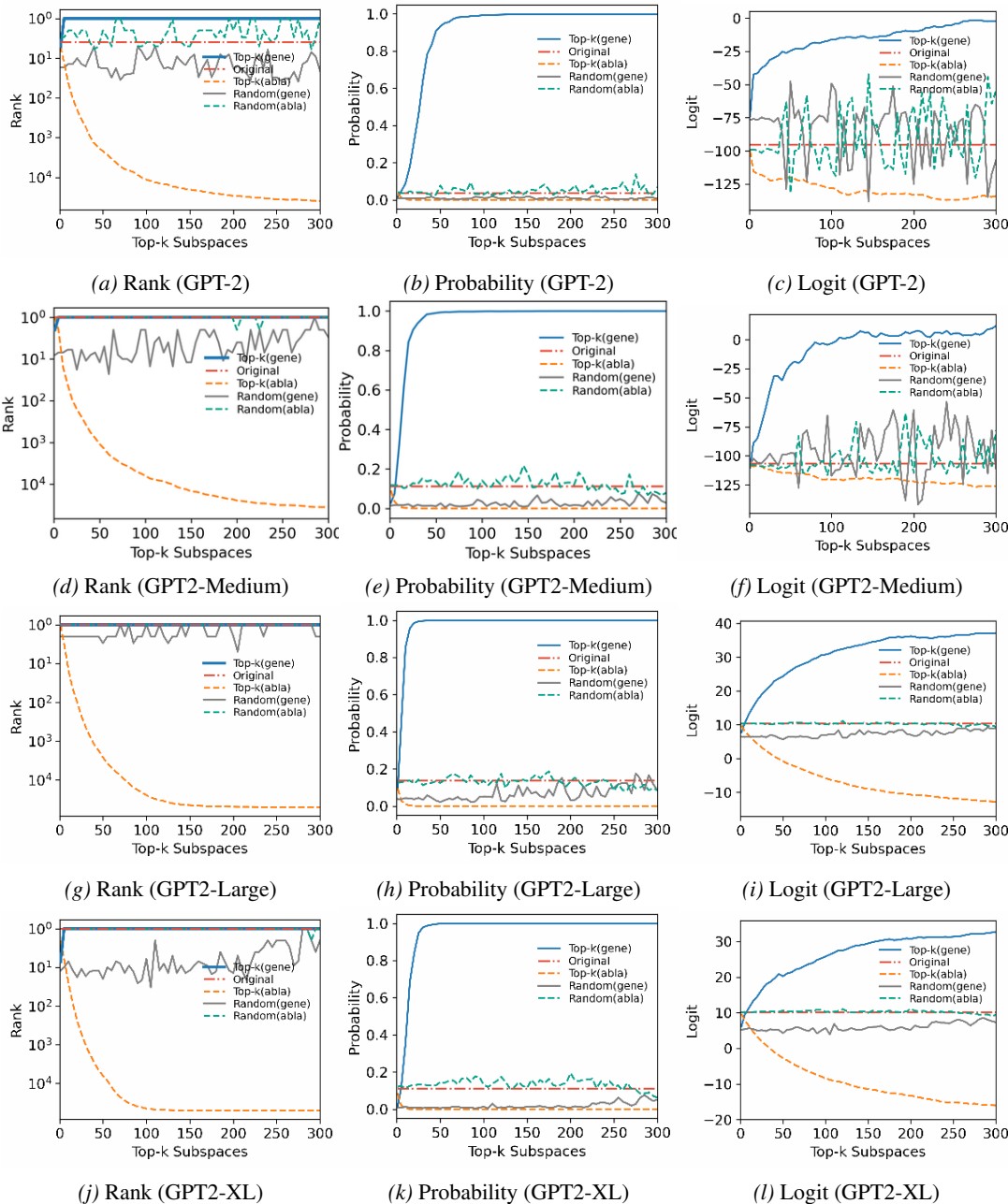

*(a)* Rank (GPT-2)  *(b)* Probability (GPT-2)  *(c)* Logit (GPT-2)

*(d)* Rank (GPT2-Medium)  *(e)* Probability (GPT2-Medium)  *(f)* Logit (GPT2-Medium)

*(g)* Rank (GPT2-Large)  *(h)* Probability (GPT2-Large)  *(i)* Logit (GPT2-Large)

*(j)* Rank (GPT2-XL)  *(k)* Probability (GPT2-XL)  *(l)* Logit (GPT2-XL)

*Figure 38.* **Pathway-Level Analysis via Top-$k$ Subspace Interventions.** We examine how the incremental inclusion of high-contribution subspaces influences the target token's *probability*, *rank*, and *logit* for the case ( "*the cat looks very*" → " *happy*") across GPT-2 models of varying scales. **Original** denotes the unmodified model. **Top-$k$ (gene/abla)**: Reconstructs or ablating the MLP down-projection matrix ($W_{\text{down}}$) using the top-$k$ high-contribution subspaces. **Random (gene/abla)**: Reconstructs or ablates $W_{\text{down}}$ using $k$ randomly selected subspaces. Here, interventions are applied to layers 4–11 (GPT-2), 7–23 (GPT2-Medium), 10–35 (GPT2-Large), and 12–47 (GPT2-XL).

## I.2. Layer-wise Distribution of Interpretable Subspaces

In this section, we analyze the layer-wise distribution of the top-20 high-contribution subspaces in GPT2-Medium, Large, and XL models for the case "*The cat looks very happy*" → "*happy*". As shown in Figure 45, dominant subspaces are concentrated at lower indices, reflecting the strong influence of principal components on model predictions. Meanwhile, higher-index subspaces provide complementary contributions that refine the leading components.

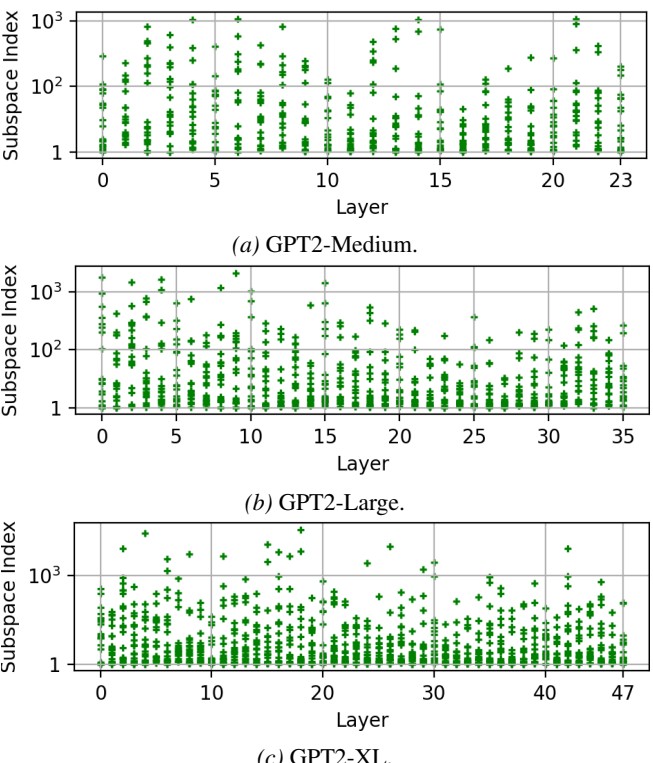

*(a)* GPT2-Medium.

*(b)* GPT2-Large.

*(c)* GPT2-XL.

*Figure 39.* **Layer-wise Distribution of Top-20 High-Contribution Subspaces in GPT-2 Models (Medium, Large, XL).** Analysis focuses on the case "*The cat looks very happy*" → "*happy*".

## I.3. Investigating Subspace Contributions to Non-Rank-1 Token Generation

Transformer MLP layers exhibit structured subspaces that play a key role in token generation. In this work, we focus on a case study to examine how these subspaces shape a specific prediction. Using the prompt "*The capital of Germany is*", where the model predicts "*Frankfurt*" instead of the correct answer "*Berlin*", we analyze how subspace-based pathways contribute to both the emergence and ranking of the predicted token.

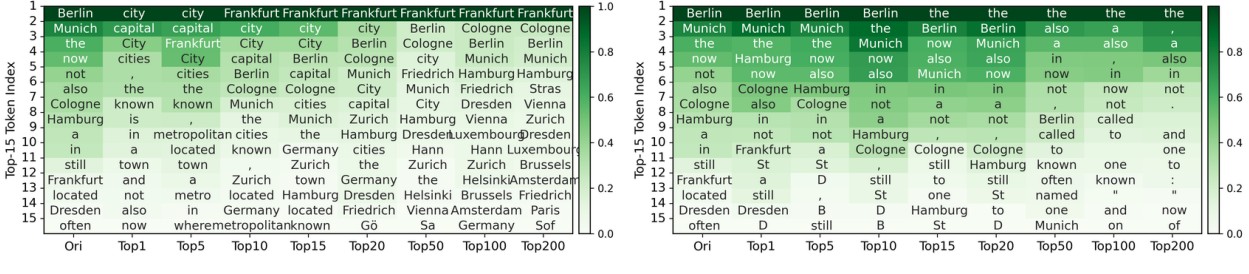

*(a)* Standard Experiment.      *(b)* Ablation Experiment.

*Figure 40.* **Reconstruction of Middle-to-Late MLP Layers via Pathways within Top-k High-Contribution Subspaces: Standard vs. Ablation.** Here, we consider the case "*The capital of Germany is*" → "*Frankfurt*".

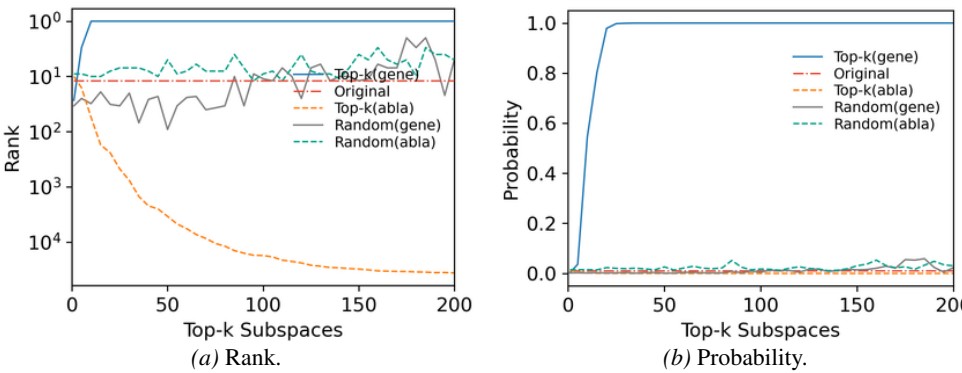

*(a)* Rank.  *(b)* Probability.

*Figure 41.* **Pathway Analysis via Top-k Subspace Interventions in GPT2-Medium.** Here, we consider the case "*The capital of Germany is*" → " *Frankfurt*".

## I.4. Additional Experiments on Model Lexical Preferences under Subspace Intervention

Following the setup in Section 4.2 (Section 4.2 (Experiment 3), we investigate the effect of intervening on Subspaces 5 and 9 in Layer 16 of GPT-2 Medium using the prompt "The cat looks very". We visualize the model outputs under amplification and suppression of these subspaces. The results in Figure 42 show that modulating each subspace leads to semantically consistent changes aligned with its learned direction. Here, Subspace 5 has a negative direction associated with "*insufficiency concepts*" and a positive direction corresponding to "*inspection actions*"; Subspace 9 exhibits a negative direction aligned with "*criticism and feedback*", and a positive direction associated with "*historical decades*". Since Subspaces 5 and 9 are not high-contribution subspaces, their contribution scores are only 0.073 and 0.0623, respectively. Consequently, their relative perturbation coefficients are higher than those used for the high-contribution Subspace 7 in Figure 4.

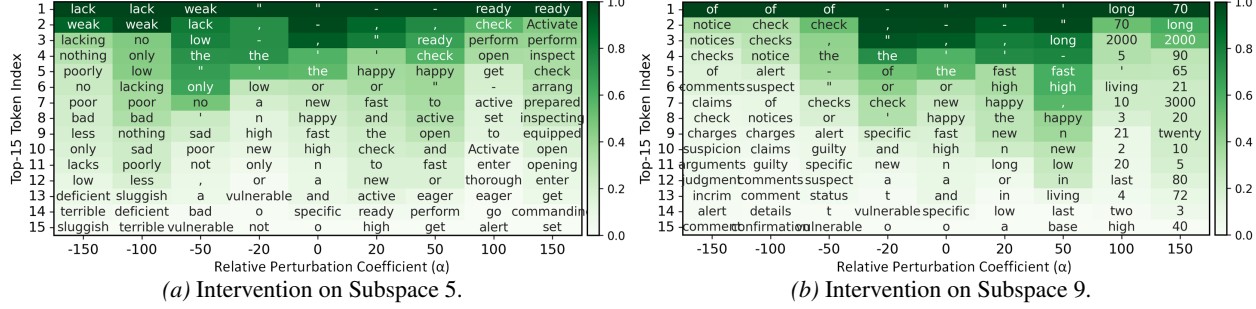

*(a)* Intervention on Subspace 5.  *(b)* Intervention on Subspace 9.

*Figure 42.* **Model Lexical Preferences under Intervention on Subspaces 5 and 9** Here, we consider the prompt "*The cat looks very*".

# J. Additional Evaluation on Benchmark Datasets and Advanced SAE Variants

## J.1. Evaluation on the Gender Pronoun Task

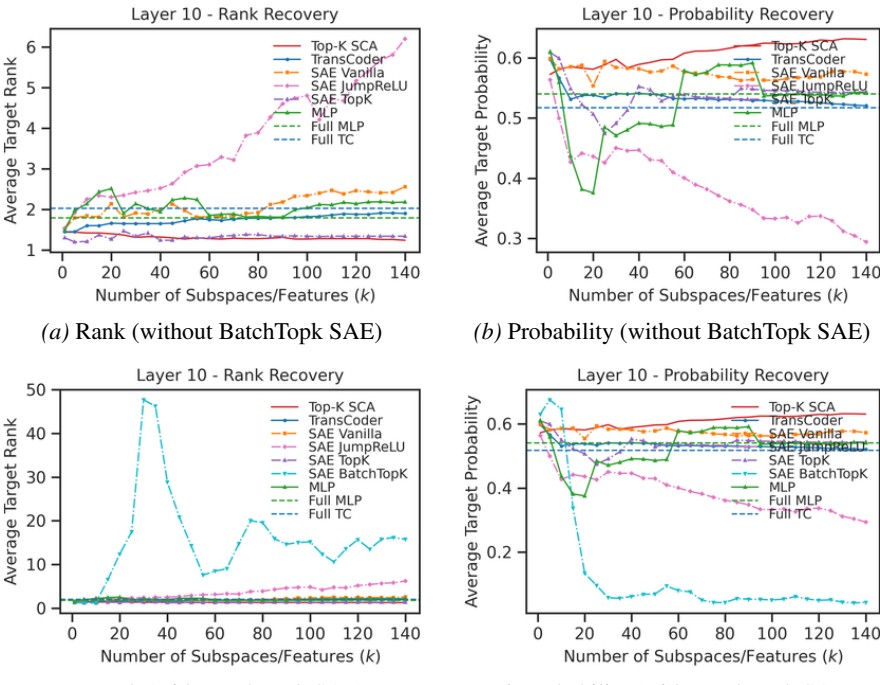

*(a)* Rank (without BatchTopk SAE)  *(b)* Probability (without BatchTopk SAE)

*(c)* Rank (with BatchTopk SAE)  *(d)* Probability (with BatchTopk SAE)

*Figure 43.* **Evaluation on the Gender Pronoun Task.** Due to the large fluctuations of the BatchTopk SAE curve, Figures 43a and 43b do not include BatchTopk SAE. Figures 43c and 43d include BatchTopk SAE.

### J.2. Evaluation on the Subject–Verb Agreement Task (Relative Clause)

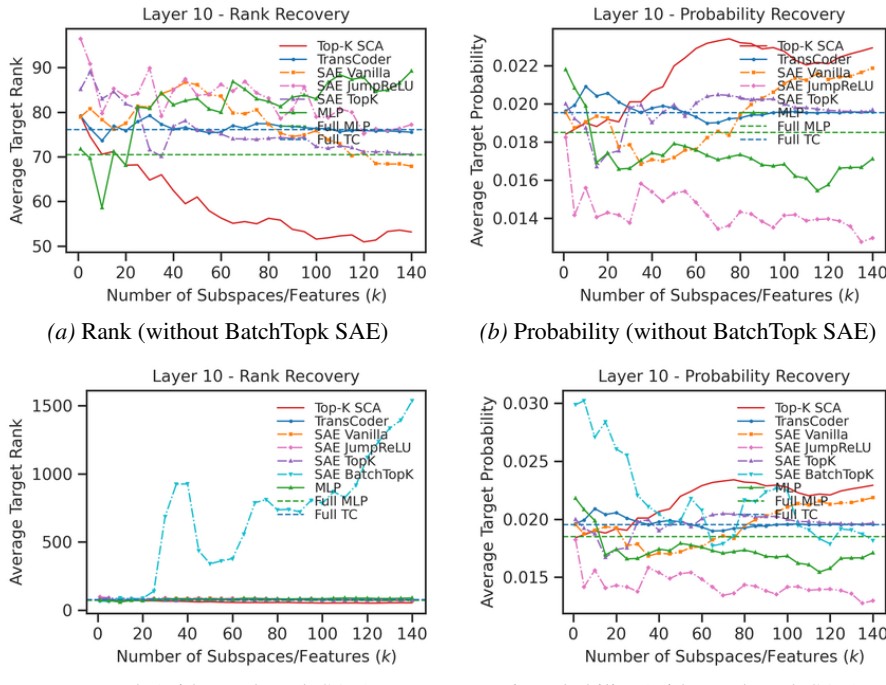

*(a)* Rank (without BatchTopk SAE)  *(b)* Probability (without BatchTopk SAE)

*(c)* Rank (with BatchTopk SAE)  *(d)* Probability (with BatchTopk SAE)

*Figure 44.* **Evaluation on the Subject–Verb Agreement Task (Relative Clause).** Due to the large fluctuations of the BatchTopk SAE curve, Figures 44a and 44b do not include BatchTopk SAE. Figures 44c and 44d include BatchTopk SAE.

### J.3. Evaluation on the Subject–Verb Agreement Task (Simple Structure)

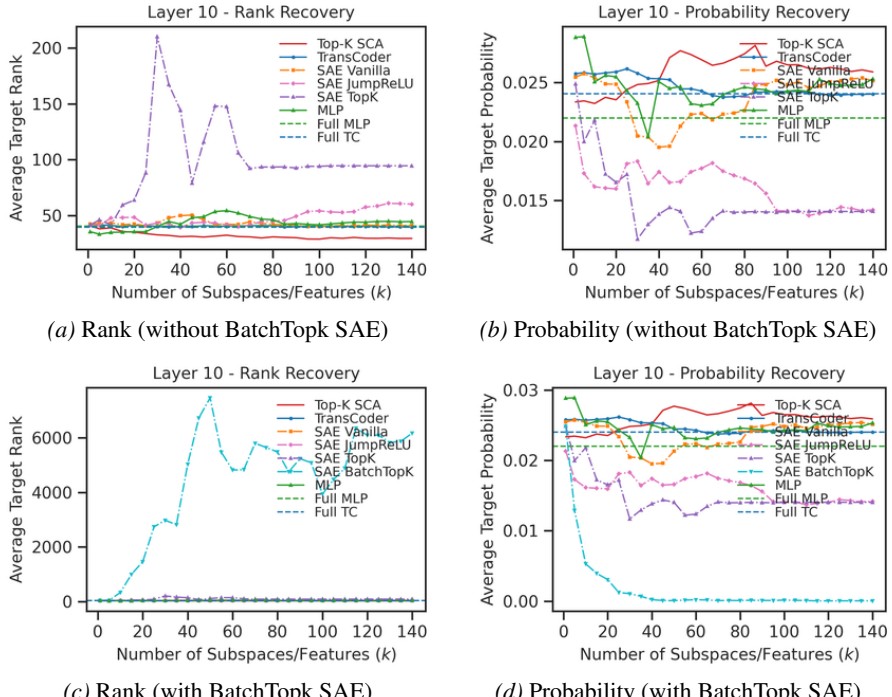

*(a)* Rank (without BatchTopk SAE)  *(b)* Probability (without BatchTopk SAE)

*(c)* Rank (with BatchTopk SAE)  *(d)* Probability (with BatchTopk SAE)

*Figure 45.* **Evaluation on the Subject–Verb Agreement Task (Simple Structure).** Due to the large fluctuations of the BatchTopk SAE curve, Figures 45a and 45b do not include BatchTopk SAE. Figures 45c and 45d include BatchTopk SAE.

## K. Properties of Highly Interpretable MLP Subspaces

### K.1. Layer-wise Count of Highly Interpretable Subspaces

In this section, we perform a layer-wise statistical analysis of interpretable subspaces derived from the Singular Value Decomposition (SVD) of MLP weight matrices. We analyze all DEU subspaces within the projection matrices across models of varying scales. Formally, a subspace is considered interpretable if it satisfies at least one of the following criteria: (i) the detector component in the up-projection matrix $W_{\text{up}}$ exhibits high interpretability; or (ii) the effector component in the down-projection matrix $W_{\text{down}}$ demonstrates high interpretability. Here, we focus exclusively on highly interpretable subspaces.

**Relationship Between Model Size and Interpretability.** In Figure 46, our empirical analysis reveals a robust positive correlation between model size and the absolute quantity of interpretable subspaces. This observation supports the hypothesis that increased parameter capacity alleviates feature superposition (Elhage et al., 2022; Arora et al., 2018).

**Asymmetric Layer-wise Specialization.** Figure 47 presents a detailed layer-wise analysis of the proportion of interpretable subspaces in both the up-projection ($W_{\text{up}}$) and down-projection ($W_{\text{down}}$) matrices.

*Layer-wise Distribution of Interpretable Subspaces in the Up-Projection Matrix:* In the up-projection matrices, interpretable subspaces are predominantly concentrated in the network's middle layers. This pattern aligns with the Transformer's hierarchical feature extraction paradigm: the initial layers primarily encode low-level lexical information, whereas the middle layers serve as the core reasoning engine, with $W_{\text{up}}$ detectors responsible for extracting abstract, high-level contextual features from the aggregated residual stream.

*Layer-wise Distribution of Interpretable Subspaces in the Down-Projection Matrix:* In contrast, interpretable subspaces within the down-projection matrices exhibit a pronounced shift toward the middle-to-late layers. This pattern indicates a mechanistic progression from feature detection to output generation. In deeper layers, the model moves beyond merely detecting context (feature construction) to actively steering the residual state toward the target output. These subspaces project the extracted features onto the output manifold, thereby enforcing specific token predictions.

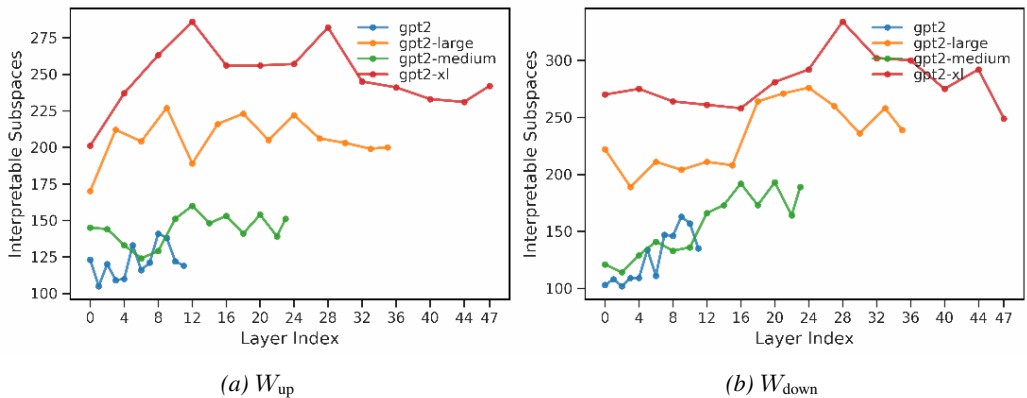

*(a)* $W_{\text{up}}$          *(b)* $W_{\text{down}}$

*Figure 46.* **Number of Interpretable Subspaces per Layer.** Here, a subspace is counted as interpretable if at least one direction (positive or negative) is interpretable.

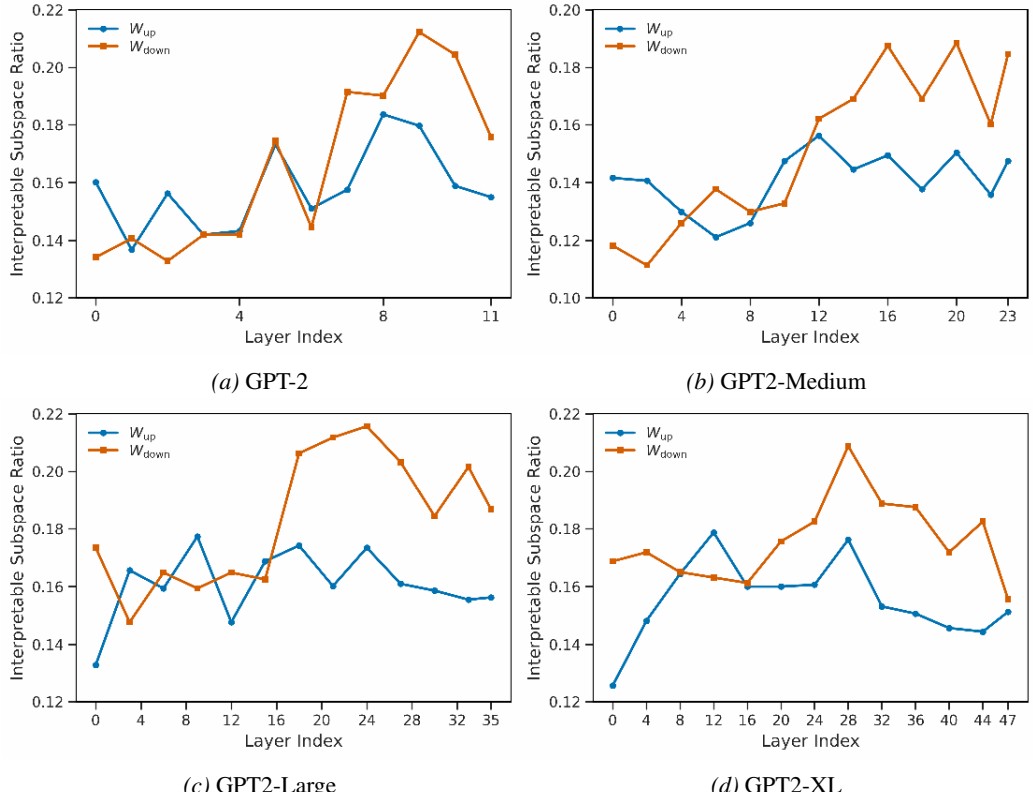

*(a)* GPT-2          *(b)* GPT2-Medium

*(c)* GPT2-Large          *(d)* GPT2-XL

*Figure 47.* **Layer-wise Proportion of Interpretable Subspaces Across GPT-2 Family.** Each figure shows the proportion of interpretable subspaces within the $W_{\text{up}}$ and $W_{\text{down}}$ projection matrices.

## K.2. Layer-wise Distribution of Interpretable Subspaces

As shown in Figure 48, we examine the spectral localization of highly interpretable subspaces in both the up-projection and down-projection matrices, analyzing how their alignment varies with network depth and model size.

**Layer-wise Spectral Concentration of Interpretable Subspaces.** Across both down-projection and up-projection matrices, we observe a consistent layer-wise trend: as network depth increases, highly interpretable subspaces progressively concentrate within the principal component regions, i.e., the low-index subspaces. This concentration effect is more pronounced in larger-scale models. It suggests that deeper and larger Transformer architectures progressively compress and align semantically meaningful representation into a low-dimensional but highly expressive spectral core. Notably, in up-projection matrices, interpretable subspaces exhibit strong concentration in small-index regions even in the earliest layers, indicating that these layers perform early semantic structuring, prioritizing core features to facilitate efficient downstream processing.

**Non-trivial Contribution of Tail Subspaces.** The dominant trend indicates that semantic information is primarily concentrated in low-index subspaces, which encode the core representations driving model predictions. At the same time, high-index (tail) subspaces retain non-trivial semantic content, providing complementary, fine-grained information rather than mere stochastic noise. This suggests that while the principal components capture the bulk of the model's semantics, the tail subspaces contribute to its expressive capacity.

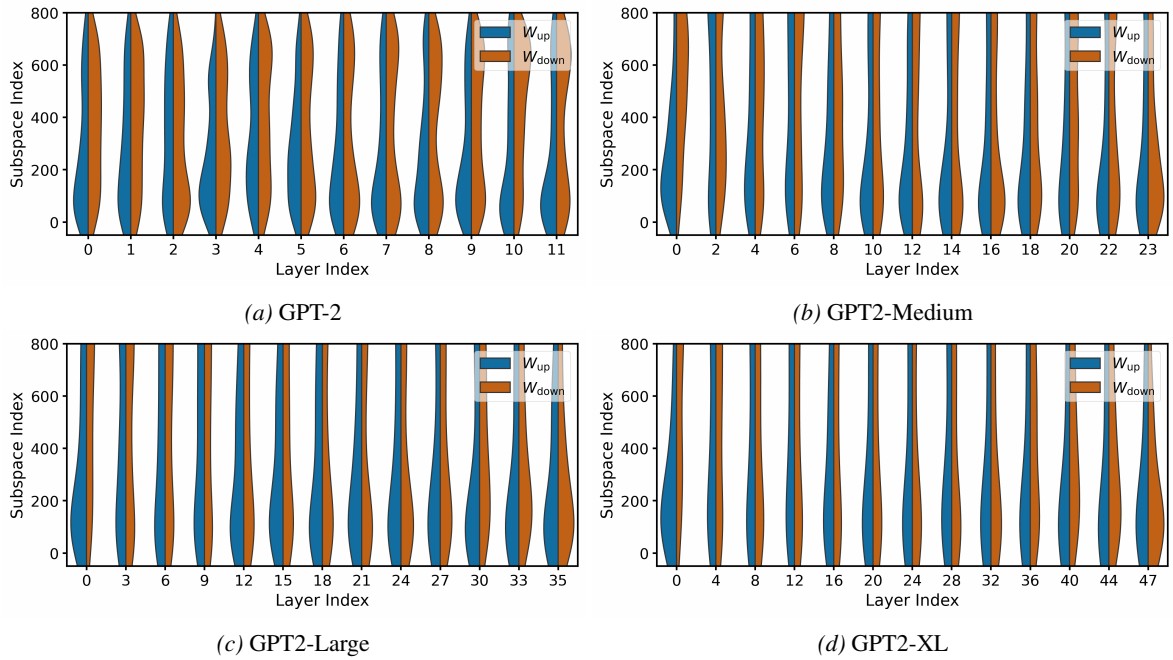

*(a)* GPT-2                     *(b)* GPT2-Medium

*(c)* GPT2-Large                *(d)* GPT2-XL

*Figure 48.* **Distribution of Interpretable Subspaces in MLP Modules Across GPT-2 Family.** Blue bars correspond to the up-projection matrix $W_{\mathrm{up}}$, and red bars correspond to the down-projection matrix $W_{\mathrm{down}}$.

## L. Effect of Starting Layer on Pathway Intervention Efficacy

In this section, we investigate how the starting depth of pathway interventions affects performance recovery. Experiments are conducted on GPT2-Medium using the $\mathcal{D}_{\mathrm{gene}}$ dataset, with interventions initiated at varying depths (Layers 5–8) and extending through Layer 23.

As illustrated in Figure 49, we evaluate performance using three metrics: mean probability, mean rank, and top-1 accuracy. The results indicate that the performance curves across different starting layers largely overlap, demonstrating that pathway interventions are robust to the choice of initial layer.

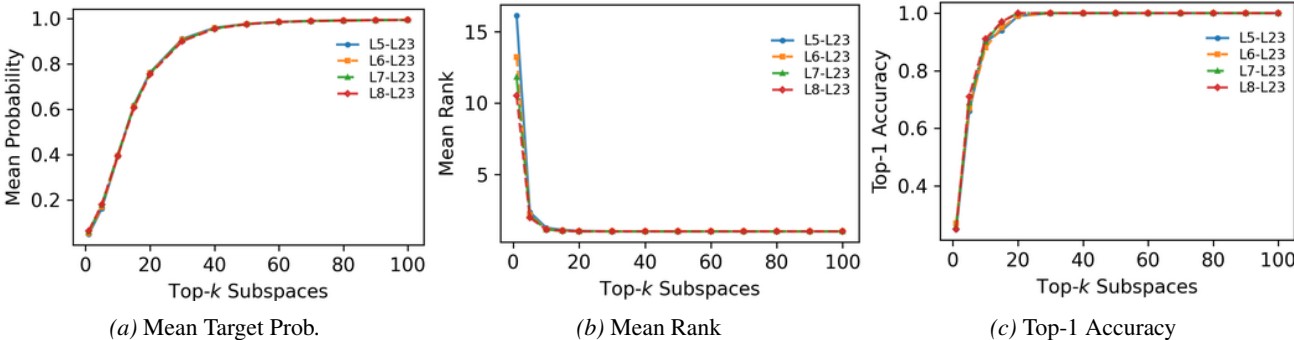

*(a)* Mean Target Prob.          *(b)* Mean Rank          *(c)* Top-1 Accuracy

*Figure 49.* **Impact of Starting Layer Initialization on Pathway Efficacy (GPT2-Medium).** The plot illustrates performance recovery trends for interventions initiating at Layers 5-8, evaluated against increasing pathway widths (subspaces per layer).

# M. Illustrative Examples of Interpretable Subspaces

In this section, we present illustrative examples of interpretable subspaces. Specifically, the interpretability of the up-projection matrix $W_{\text{up}}$ is attributable to its detector component, whereas the interpretability of the down-projection matrix $W_{\text{down}}$ derives from its effector component. Furthermore, we consider both the "Positive" and "Negative" directions, corresponding to the semantic concepts aligned with the opposing directions ($\pm\mathbf{u}_i$ or $\pm\mathbf{v}_i$) of the singular vector, thereby reflecting the inherent sign ambiguity in SVD (Kolda & Bader, 2009).

## M.1. Interpretation of DEU Subspaces Across Models

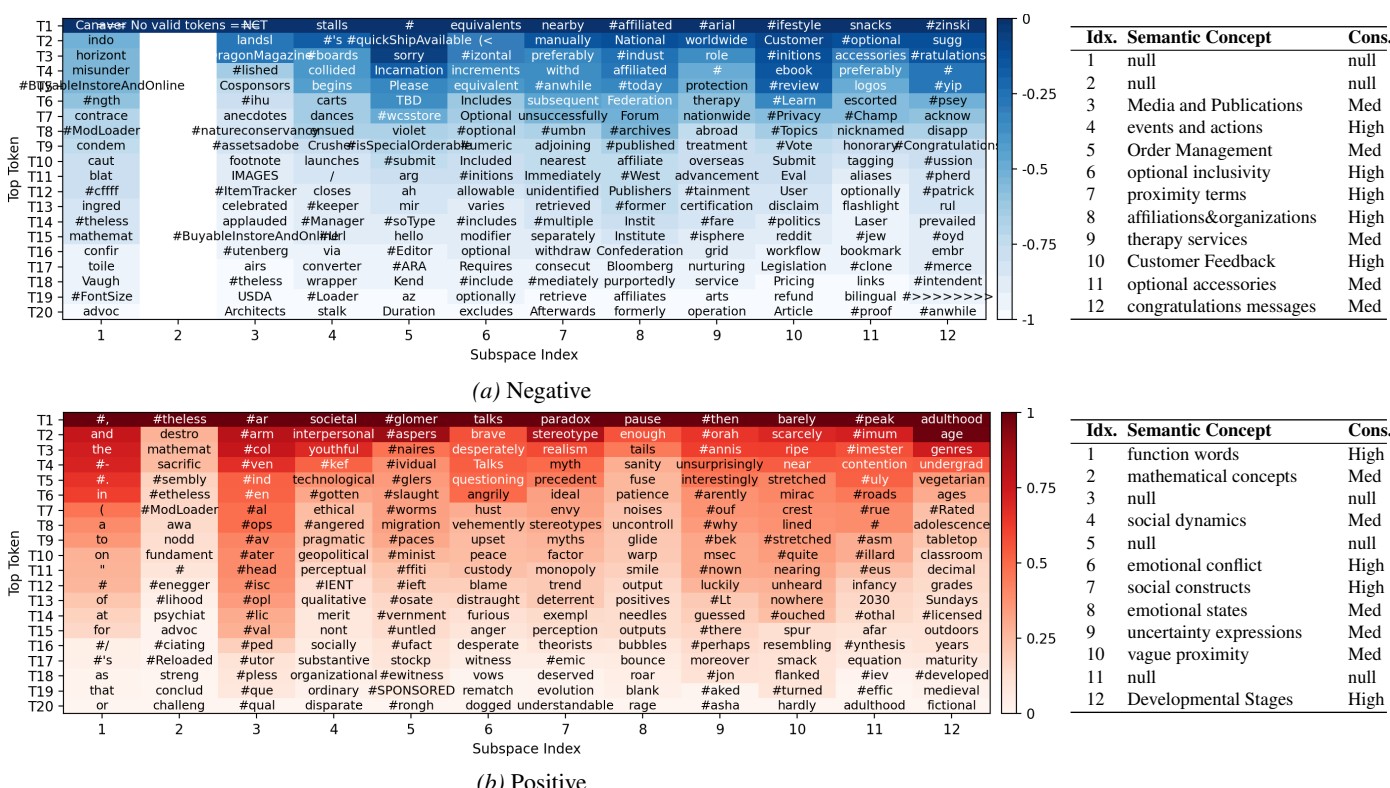

*Figure 50.* **Example of DEU Subspace Interpretability via Semantic Consistency in the Up-Projection Matrix (GPT-2, Layer 7).** Here, the color intensity represents the consistency score, while columns with blank entries correspond to uncaptured or unreadable tokens and have been omitted. "Idx." refers to the subspace index, and "Cons." indicates the subspace interpretability degree.

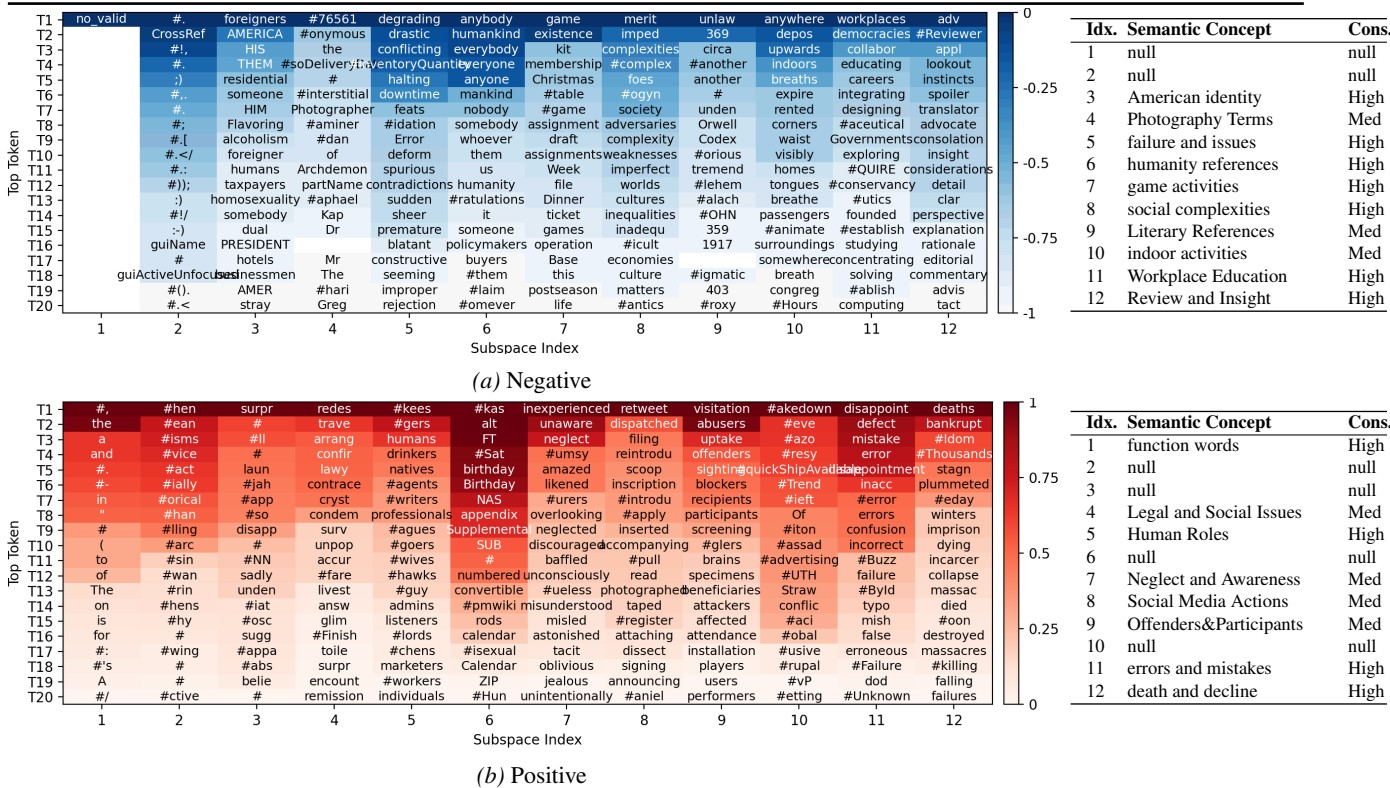

*(a)* Negative

*(b)* Positive

*Figure 51.* **Example of DEU Subspace Interpretability via Semantic Consistency in the Down-Projection Matrix (GPT-2, Layer 7).** Here, the color intensity represents the consistency score, while columns with blank entries correspond to uncaptured or unreadable tokens and have been omitted. "Idx." refers to the subspace index, and "Cons." indicates the subspace interpretability degree.

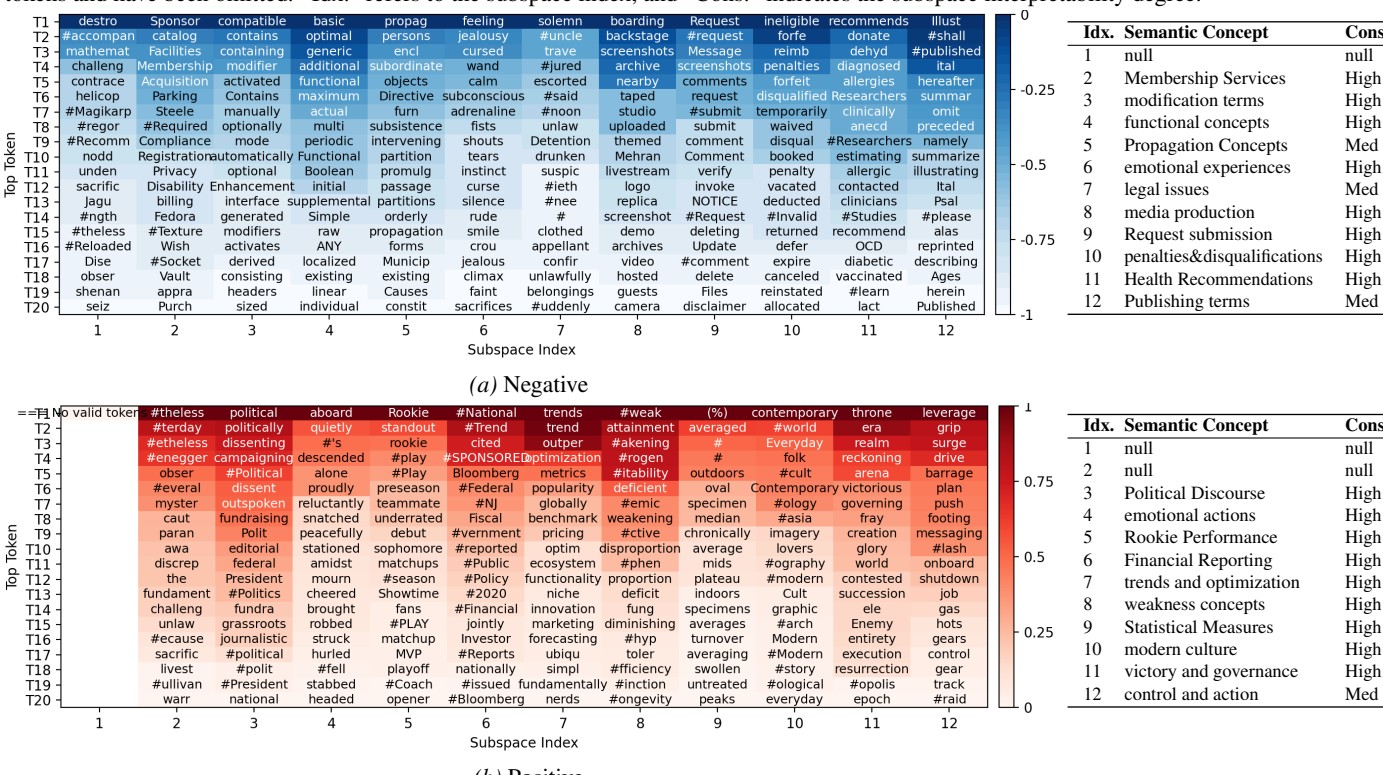

*(a)* Negative

*(b)* Positive

*Figure 52.* **Example of DEU Subspace Interpretability via Semantic Consistency in the Up-Projection Matrix (GPT2-Medium, Layer 16).** Here, the color intensity represents the consistency score, while columns with blank entries correspond to uncaptured or unreadable tokens and have been omitted. "Idx." refers to the subspace index, and "Cons." indicates the subspace interpretability degree.

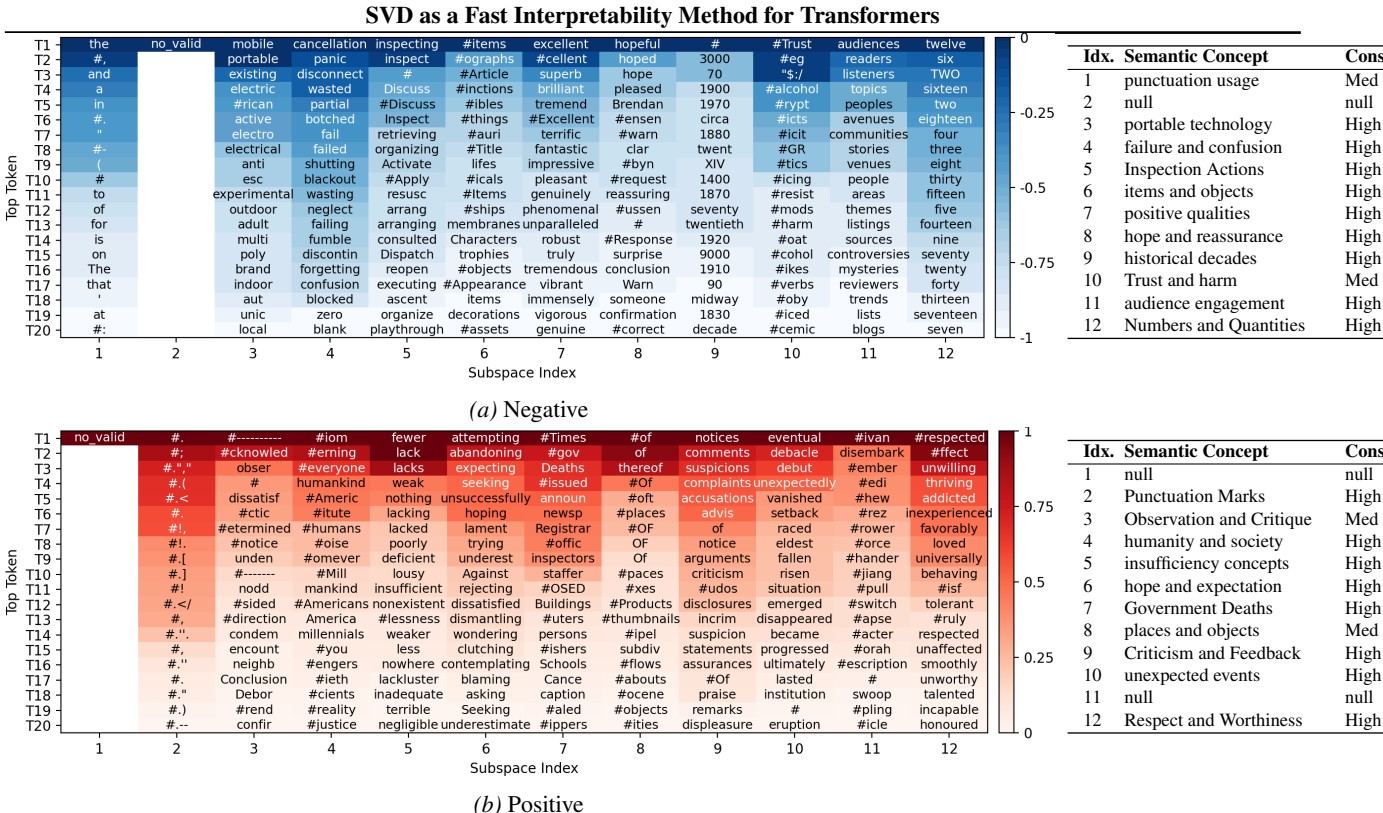

*(a)* Negative

*(b)* Positive

*Figure 53.* **Example of DEU Subspace Interpretability via Semantic Consistency in the Down-Projection Matrix (GPT2-Medium, Layer 16).** Here, the color intensity represents the consistency score, while columns with blank entries correspond to uncaptured or unreadable tokens and have been omitted. "Idx." refers to the subspace index, and "Cons." indicates the subspace interpretability degree.

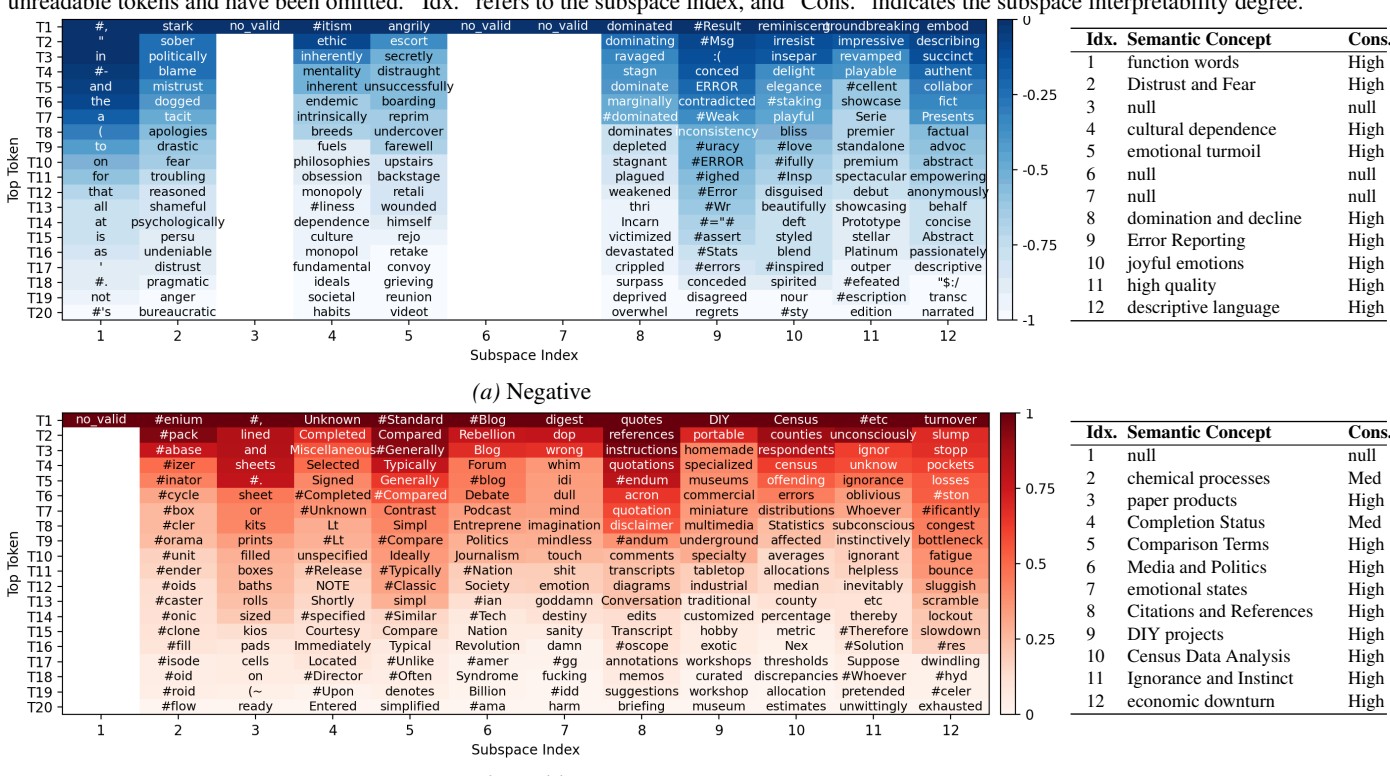

*(a)* Negative

*(b)* Positive

*Figure 54.* **Example of DEU Subspace Interpretability via Semantic Consistency in the Up-Projection Matrix (GPT2-Large, Layer 18).** Here, the color intensity represents the consistency score, while columns with blank entries correspond to uncaptured or unreadable tokens and have been omitted. "Idx." refers to the subspace index, and "Cons." indicates the subspace interpretability degree.

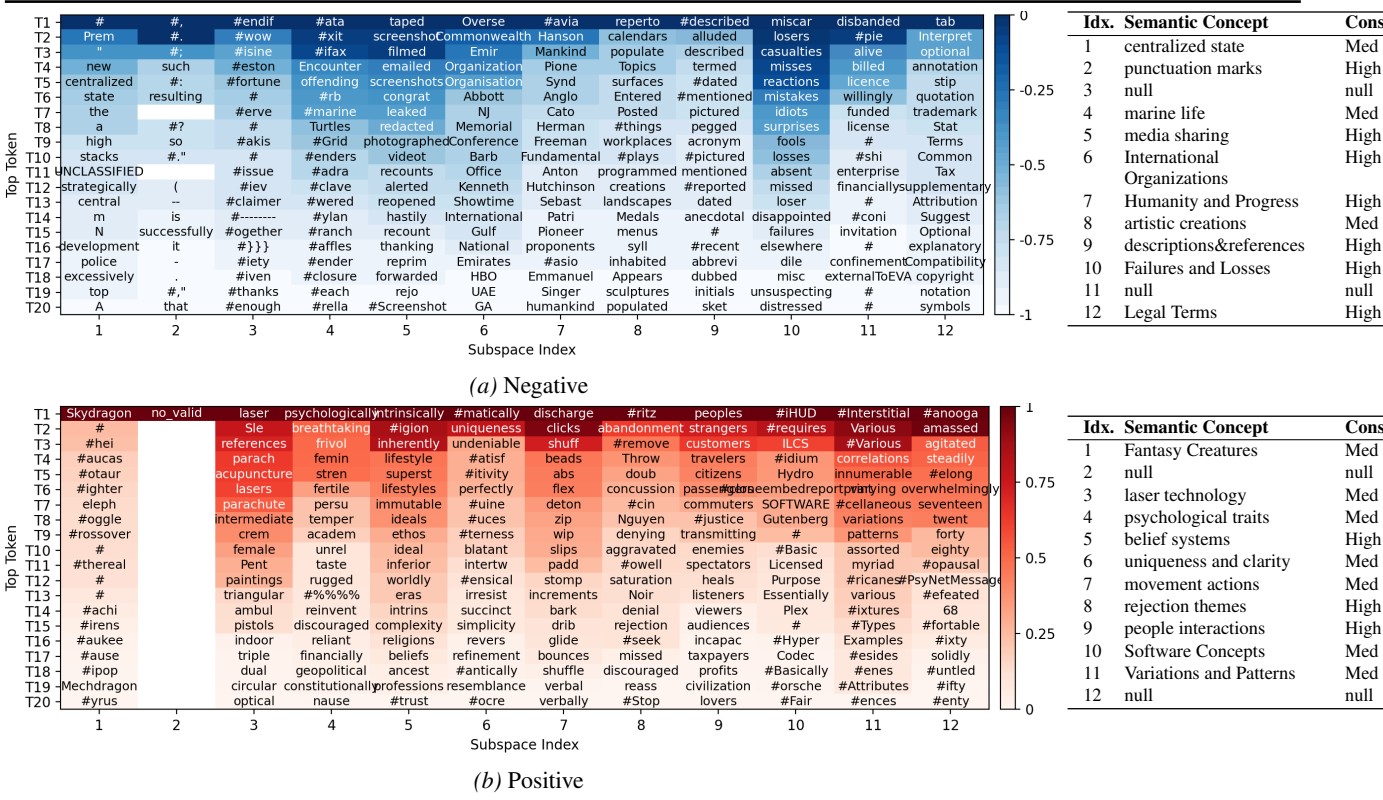

*(a)* Negative

*(b)* Positive

**Figure 55. Example of DEU Subspace Interpretability via Semantic Consistency in the Down-Projection Matrix (GPT2-Large, Layer 18).** Here, the color intensity represents the consistency score, while columns with blank entries correspond to uncaptured or unreadable tokens and have been omitted. "Idx." refers to the subspace index, and "Cons." indicates the subspace interpretability degree.

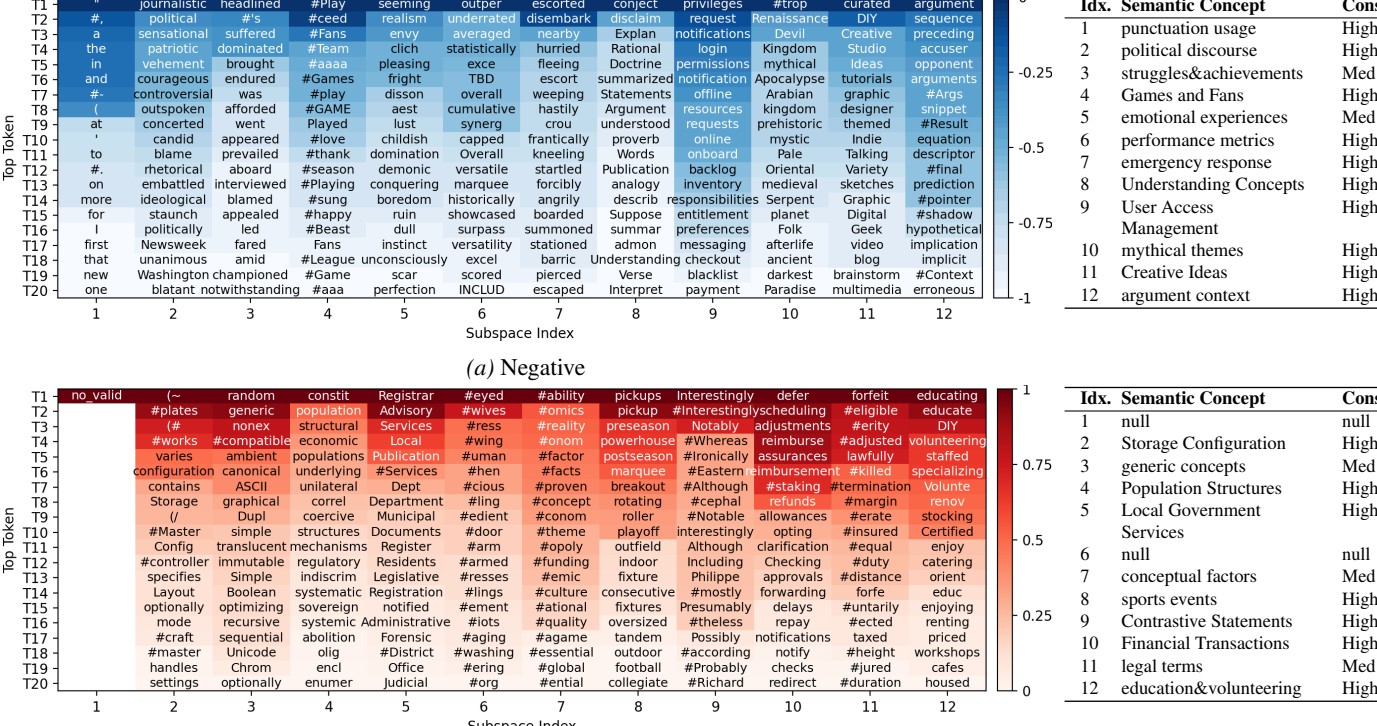

*(a)* Negative

*(b)* Positive

**Figure 56. Example of DEU Subspace Interpretability via Semantic Consistency in the Up-Projection Matrix (GPT2-XL, Layer 26).** Here, the color intensity represents the consistency score, while columns with blank entries correspond to uncaptured or unreadable tokens and have been omitted. "Idx." refers to the subspace index, and "Cons." indicates the subspace interpretability degree.

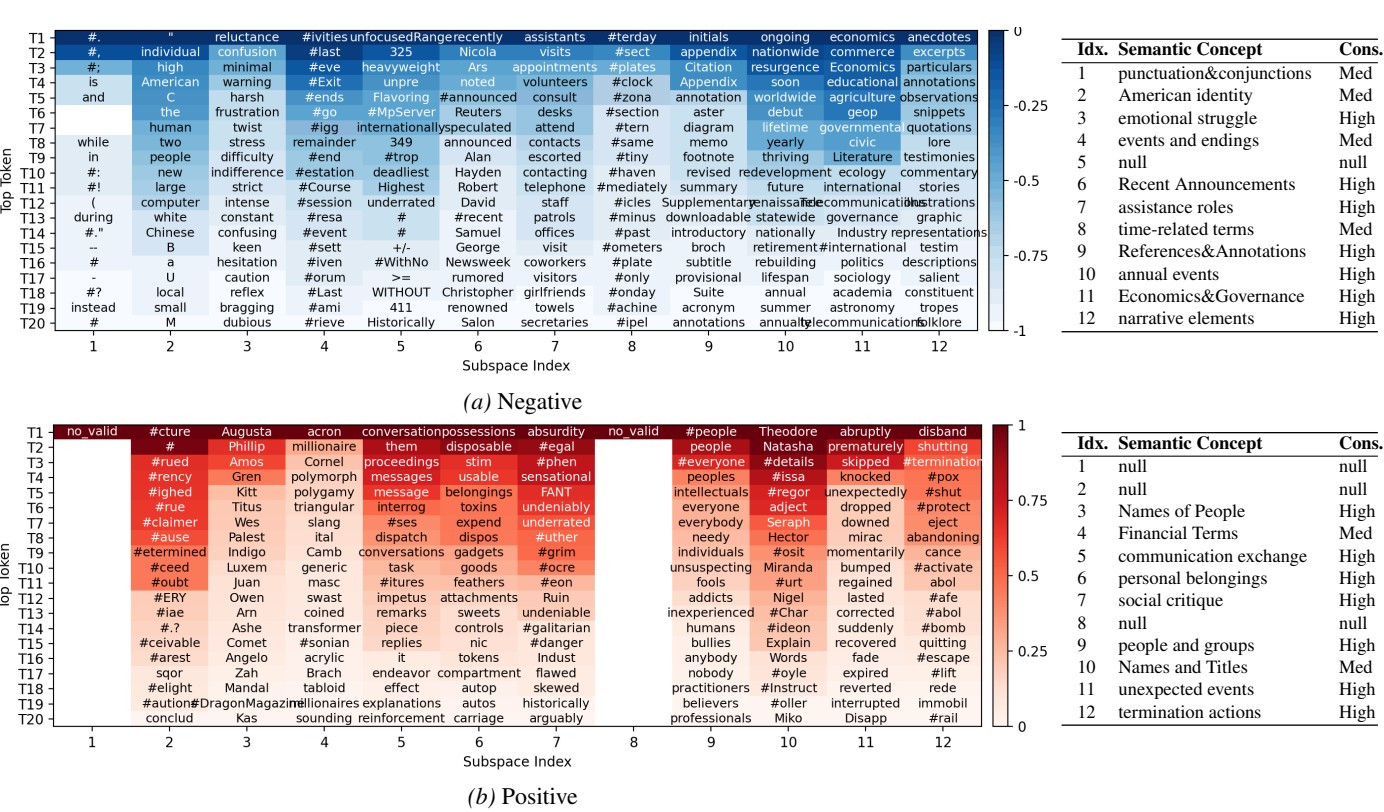

*Figure 57.* **Example of DEU Subspace Interpretability via Semantic Consistency in the Down-Projection Matrix (GPT2-XL, Layer 26).** Here, the color intensity represents the consistency score, while columns with blank entries correspond to uncaptured or unreadable tokens and have been omitted. "Idx." refers to the subspace index, and "Cons." indicates the subspace interpretability degree.

## M.2. Semantic Consistency of Top-100 Tokens Captured by DEU Subspaces

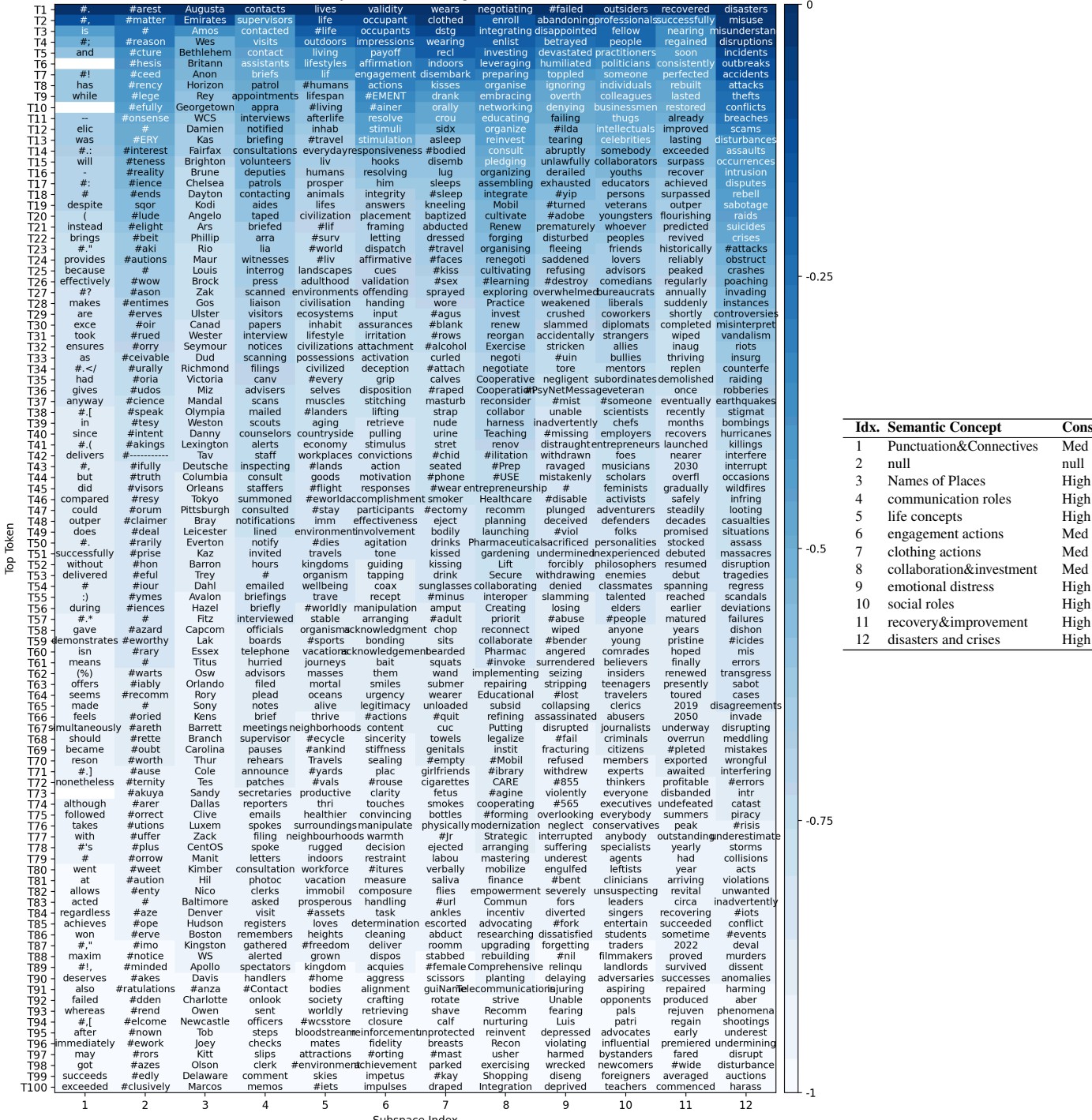

| Idx. | Semantic Concept | Cons. |
|------|------------------|-------|
| 1 | Punctuation&Connectives | Med |
| 2 | null | null |
| 3 | Names of Places | High |
| 4 | communication roles | High |
| 5 | life concepts | High |
| 6 | engagement actions | Med |
| 7 | clothing actions | Med |
| 8 | collaboration&investment | Med |
| 9 | emotional distress | High |
| 10 | social roles | High |
| 11 | recovery&improvement | High |
| 12 | disasters and crises | High |

*Figure 58.* **Semantic Consistency of Subspaces in the Down-Projection Matrix via Top-100 Tokens (GPT2-XL, Layer 25).** The table lists the top-100 tokens associated with the top-12 subspaces (negative), demonstrating high semantic coherence. Here, the color intensity represents the consistency score, while columns with blank entries correspond to uncaptured or unreadable tokens and have been omitted. "Idx." refers to the subspace index, and "Cons." indicates the subspace interpretability degree.

