# OpenReview forum: "SVD as a Fast Interpretability Method for Transformers"
_ICML.cc/2026/Conference — ICML 2026 spotlight_

### Official Review · Reviewer_H7DY · 2026-03-11

**Soundness:** 3
**Presentation:** 2
**Significance:** 2
**Originality:** 3
**Overall Recommendation:** 4
**Confidence:** 5

**Summary:**

The paper proposes Native Network Anatomy (NaNA), an alternative to post-hoc and proxy methods using Sparse Autoencoders and Transcoders. The proposed method analyzes the MLP of the transformer natively rather than building features in the reconstructed activation space. The paper builds on Beren et al.'s (2022) observation that the singular vectors of MLPs encode semantic structure.

**Compliance With Llm Reviewing Policy:**

Affirmed.

**Final Justification:**

As the authors have resolved all my concerns in the rebuttal, especially the discussion about how each rank-1 term in SVD corresponds to a functional subspace. The addition of experiments on the gender pronoun, subject–verb agreement (simple structure), and subject–verb agreement (relative clause) datasets strengthens the empirical validation of the work.  Thus, I am increasing the overall recommendation to 4.

**Key Questions For Authors:**

Please refer to the weaknesses section above.

**Limitations:**

1. Authors should clearly discuss how the proposed method differs from existing methods, such as those proposed by Beren & Black (2022).

2. The paper needs clear notations to improve readability.

**Strengths And Weaknesses:**

Strengths:
1. The proposed Subspace Contribution Analysis is training-free. It identifies sparse, dominant pathways while quantifying their impact on target token predictions.

2.  The use of singular vectors of projection matrices to analyze how token embeddings align with native model parameters is promising. This is a meaningful alternative to approaches that rely on learning auxiliary functions such as sparse autoencoders.

3. The intervention experiments are interesting and provide a way to understand how the proposed method works.

Weaknesses:
1. Section 3 is difficult to follow and would benefit from substantial clarification. It is not clear why each rank-1 term $\sigma_i u_i v_i^{\top}$ should be interpreted as defining a functional subspace. This point seems central to the method and should be carefully justified.

2. The paper will benefit from discussion on Beren & Black (2022), how the proposed method differentiates from the existing work.

3. The notations in Subsection 3.2 are hard to follow. The authors use many variables without clarifying their meanings. For example, $H^{(l-1)}$, $\bar{h}^{(l)}_{m}$ etc.

4. The main experimental results are qualitative and use a single example to discuss the advantages of the proposed method.

---

> ### Author Rebuttal · Authors · 2026-03-30
>
> Thank you for your valuable suggestion. Following your guidance, we have further revised the paper to improve its readability and have added additional generalization experiments on top of the existing ones.
>
> ### **Weaknesses 1:** *Section 3 is difficult to follow. Why is $\sigma_i u_i v_i^\top$ defined as a functional subspace?*
>
> 1. In Section 3, **we first outline the NaNA framework (Lines 105–113), then present the technical details of each component method in Sections 3.1–3.4**. Following your suggestion, we have further enhanced the clarity and readability of the manuscript.
>    - In Section 3.1, we first formalize *Detector–Effector Units* (DEUs) as the elementary functional units of linear projections.
>    - In Section 3.2, we then propose *Subspace Contribution Analysis* (SCA) to quantify the predictive role of individual rank-1 components.
>    - Finally, in Sections 3.3 and 3.4, we construct *directed DEU-based pathways* and define *spectral interventions* for causal analysis.
>
> 2. Regarding the description of $\sigma_iu_iv_i^\top$ as a functional subspace, we provide detailed explanations in **Lines 126–158**, which include:
>
>      -  **Each rank-1 term $\sigma_iu_iv_i^\top$ functions as a distinct information channel formed by an outer product.** We interpret each rank-1 operator as a *Detector–Effector Unit*: $v_i$ acts as a *detector* measuring input alignment, $u_i$ as an *effector* determining output orientation, and $\sigma_i$ as a *signal gain* (**technical details in Lines 126–137**).
>      -  **Furthermore, the MLP sublayer can be expressed as a superposition of detector-effector units (i.e., $\sigma_i u_i v_i^\top$)**, with activations $h_i$ modulating each effector direction (**technical details in Lines 150–158**).
>
> ### **Weaknesses 2 and Limitations 1:** *The paper will benefit from discussion on Beren & Black (2022).*
> Thanks for your suggestion. Below, we provide the current discussion of the distinctions and connections between Beren & Black (2022) and our work, as presented in the paper. Following your suggestion, we have further improved the clarity in the revised version.
> 1. **Distinction from Beren & Black (2022):** Beren & Black (2022) proposed methods for interpreting the singular vectors of MLP weights. In contrast, our work extends this static interpretability tool into a dynamic functional analysis framework, formalizing these spectral components as detector–effector units that directly mediate token prediction, thereby identifying sparse, dominant pathways. The relevant discussions can be found in:
>    - In the *Introduction* section (Lines 067–071), we highlight the differences between our work and Beren & Black (2022);
>    - In the *Related Work* section (Lines 665–677), we discuss the limitations of Beren & Black (2022)'s work.
> 2. **Connection to Beren & Black (2022):** In our study, Beren & Black (2022)'s work is used primarily as an interpretability tool for DEU subspaces. The relevant discussions can be found in:
>     - In Section 2.2 (Lines 076–102), we provide the technical details of the interpretation method of Beren & Black (2022).
>    - In Section 3.1 (Lines 145–149), we discuss the connection between Beren & Black (2022)'s methods and our approach.
>
> ### **Weaknesses 3 and Limitations 2:** *The notations are hard to follow. For example, $H^{(l-1)}$, $\bar{h}^{(l)}_{m}$ etc.*
> 1. Many notations are first introduced in the *Preliminaries* section (Sections 2.1 and 2.2), where we also provide detailed explanations. For example, $H^{(l-1)}$ and $\bar{h}^{(l)}_{m}$ can be found in:
> > **Lines 060-066:**\
> The attention sublayer produces an intermediate residual state
> \begin{equation}
> \mathbf{\bar{h}}_m^{(l)}=\mathbf{h}_m^{(l-1)}+\text{Attn}^{(l)}(\gamma(\mathbf{H}^{(l-1)}))_m,
> \end{equation}where $\gamma(\cdot)$ denotes layer normalization and $\mathbf{H}^{(l-1)}=(\mathbf{h}_1^{(l-1)},\dots,\mathbf{h}_M^{(l-1)})$ is the sequence of hidden states.
> 2. Based on your suggestion, in the revised version we have further checked and refined all formulas and notations to enhance clarity.
>
> ### **Weaknesses 4:** *The main experimental results are qualitative and use a single example to discuss the advantages of the proposed method.*
> **Our evaluation includes both "Individual Case Studies" and "Generalization Experiments".**
> 1. In our experiments, we extend the analysis from case studies (section 4.2) to broader **generalization evaluations (section 4.3 and section 4.4 part 1)**, using the *generalization*, *target variance*, and *prefix variance datasets*.
> 2. In addition, we conduct **new experiments** on the *gender pronoun, subject–verb agreement (simple structure)*, and *subject–verb agreement (relative clause) datasets*. **All results are provided in Figures 1, 2, and 3 via the following link**: https://github.com/Anonymous2026SSS/results/blob/main/to_reviewer_onrA.md .
> 3. Finally, in Appendix G, we present more representative cases covering a broad spectrum of linguistic phenomena.

---

> > ### Author Rebuttal · Reviewer_H7DY · 2026-04-02
> >
> > I thank the authors for their thorough response. The rebuttal has addressed several of my concerns, and I have one follow-up point regarding Weakness 1.
> >
> > The lines 126–158 clarify the interpretation of the Detector-Effector unit. But an SVD decomposition by itself does not imply that each rank-1 term corresponds to a functional subspace. It remains unclear why spectral components should correspond to meaningful, separable functional units rather than just a convenient basis decomposition. Reviewer wjvF raised a similar point, that SVD identifies directions of maximum variance in the weight matrix, but these directions do not necessarily correspond to meaningful or disentangled features. I would appreciate clarification on this.
> >
> > Suggestion: To justify each rank-1 term as a functional subspace, either a theoretical justification for why the SVD basis should be privileged over other orthogonal bases, or empirical evidence, would strengthen the paper.
> >
> > For now, I will maintain my overall recommendation.

---

> > > ### Author Response · Authors · 2026-04-02
> > >
> > > Thank you for the thoughtful follow-up. We agree that this is a crucial point and appreciate the opportunity to clarify it here. We will improve the manuscript by including an explanation of this point in the manuscript. Below, we provide both **theoretical and empirical reasons** why the SVD-induced basis is privileged and aligns with functional "Detector-Effector" subspaces.\
> > > **Theoretical properties distinguishing an SVD-induced basis from an arbitrary orthogonal basis:**
> > > 1. Following the optimal low-rank approximation (Eckart–Young theorem) [1], among all rank-k decompositions, the top-k SVD terms minimise the approximation error to the full weight matrix under the Frobenius norm (or any unitarily invariant norm). This means that each additional rank-1 SVD term captures the maximum possible remaining influence on the matrix's action. For example, a rank-1 component $u_1 \sigma_1 v_1^\top$ represents the direction in the input space ($v_1$) that undergoes the maximum possible amplification ($\sigma_1$) into the output space ($u_1$). A random orthogonal basis enjoys no such ordering: its components do not correspond to any notion of functional contribution, and there is no principled way to rank or select among them. In more informal terms, because the SVD isolates the directions of maximum variance, these components represent the most "dominant" pathways through the layer. This is one of the aspects making this basis special.
> > > 2. Further justification comes from the diagonalization property of the SVD. By construction, $W = U\Sigma V^\top$ maps each input direction $v_i$ exclusively onto the output direction $u_i$, scaled by $\sigma_i$, with zero coupling to any other output direction $u_j,; j\neq i$. This is not true of an arbitrary orthogonal decomposition, where cross-terms between different basis directions generally remain. As a consequence, each rank-1 SVD term represents a linearly decoupled input–output channel: activating detector $v_i$ produces a response only in effector $u_i$, not in any other effector. It is in this precise sense, and not merely because the basis vectors happen to be orthogonal as is true of any orthonormal basis, that SVD-induced Detector-Effector units operate with minimal interference between one another, representing the linearly decoupled functional streams the model has learned [2,3].
> > > 3. Another indication (yet not a rigid proof) that the SVD basis is privileged comes from studies on the learning dynamics of gradient descent (e.g., Saxe et al. [4]). This work demonstrates that for deep linear neural networks, gradient descent tends to learn independent input-output mappings sequentially, exactly along the singular vectors of the task's structural covariance. The network does not learn in an arbitrary orthogonal basis; it explicitly builds its weights by incrementally adding capacity along the principal directions of the data manifold. This provides an existence proof that SVD-aligned structure can emerge from gradient-based learning, rather than being an arbitrary post-hoc decomposition.\
> > >   [1] The approximation of one matrix by another of lower rank.\
> > >   [2] Sculpting Subspaces: Constrained Full Fine-Tuning in LLMs for Continual Learning.\
> > >   [3] Task Singular Vectors: Reducing Task Interference in Model Merging.\
> > >   [4] Exact solutions to the nonlinear dynamics of learning in deep linear neural networks.
> > >
> > > **Empirical validation**\
> > > We provide a series of evidence showing that high-contribution SVD components behave as functional subspaces, rather than merely convenient basis vectors:
> > > 1. **Dominance in token prediction.** If SVD components were merely a basis artefact, one would expect that predicting a specific token would require a diffuse mixture of many components, and that the identity of the dominant components would vary arbitrarily across prompts. Instead, Sections 4.2 and 4.3 show that a small, consistent set of high-contribution rank-1 terms suffices to dominate target-token prediction, while ablating them causes a marked drop. This selectivity, with specific components for specific predictions, is the empirical signature of functional separability.
> > > 2. **Cross-prompt robustness.** Section 4.3 (Part 2) and Appendix E use Controlled Variation Sets to show that the same subspaces recur as dominant contributors across prompts sharing a common semantic target (e.g., positive-adjective semantics), even when surface form and prefix vary. An arbitrary orthogonal basis would not exhibit this consistency; the fact that SVD components do suggests they track something functionally real in the network's computation.
> > >
> > > **Changes to the Manuscript:**\
> > > We deeply appreciate you pushing us on this point. In the final revision, we will add a dedicated paragraph titled "Why SVD Rank-1 Components Represent Functional Subspaces", incorporating both the theoretical implicit bias of SGD and the Eckart–Young theorem, along with empirical ablation results to justify this framing.

---

### Official Review · Reviewer_wjvF · 2026-03-12

**Soundness:** 3
**Presentation:** 4
**Significance:** 3
**Originality:** 4
**Overall Recommendation:** 5
**Confidence:** 3

**Summary:**

This paper proposes Native Network Anatomy (NaNA), a training-free interpretability method
for transformer models based on Singular Value Decomposition (SVD) of MLP weight matrices.
The authors interpret each rank-1 component of the decomposition as a Detector-Effector Unit
(DEU) consisting of an input detector, an output effector, and a gain. They introduce Subspace
Contribution Analysis (SCA) to estimate how individual subspaces contribute to model
predictions and extend the method to identify pathways across layers. Experiments on GPT-2
models show that a small number of subspaces dominate predictions and that manipulating
them can significantly change token probabilities. The method is also computationally efficient
compared to interpretability approaches that require training auxiliary models.

**Compliance With Llm Reviewing Policy:**

Affirmed.

**Final Justification:**

I stand by my previous recommendation after having engaged with the authors of this paper, they have addressed my most important concerns and provided enough arguments.

**Key Questions For Authors:**

1. The paper focuses primarily on MLP layers. Could the proposed framework, or a simple
extension of it, be used to analyze matrices involved in the attention mechanism (Q, K, V projections), and if so,
would the same detector-effector interpretation still hold?
2. How robust are the identified subspaces across different prompts or datasets? Do the
same subspaces consistently appear as dominant contributors?
3. The experiments are conducted only on models from the GPT-2 family, which are
relatively small compared with modern large language models. It remains unclear
whether the proposed approach scales effectively to larger architectures with billions of
parameters. Would the method remain computationally efficient and informative when
applied to larger modern language models?

**Limitations:**

yes

**Strengths And Weaknesses:**

Strengths
The main strength of the paper is the simplicity of the proposed approach. The framework
relies on a well-known tool, singular value decomposition, and applies it directly to model
weights. This makes the method conceptually straightforward and easy to implement. Unlike
many recent interpretability techniques, the approach does not require additional training
procedures, datasets, or hyperparameter tuning.
Another positive aspect is the computational efficiency of the method. The analysis procedure
can be performed quickly and with relatively low memory requirements. This makes the
method potentially useful for interpretability studies or exploratory analyses across multiple
models.
The paper also includes causal intervention experiments, where the activations of specific
subspaces are amplified or suppressed to observe changes in token probabilities. These
experiments provide some evidence of the correlation of the identified subspaces with
predictions and the influence they have on the model behavior.

Weaknesses
As the authors mention in the Discussion section, the approach focuses only on MLP layers and
does not analyze attention mechanisms. In transformer architectures, attention heads often
play an important role in information routing and reasoning. Ignoring attention components
may therefore overlook important parts of the model’s internal computation. Extending the
framework to attention matrices would significantly strengthen the contribution. Yet, I still
believe this work is a good first step for an extension on attention mechanisms to be done
separately.
Another limitation concerns the interpretation of singular vectors as semantic features. SVD
identifies directions of maximum variance in the weight matrix, but these directions do not
necessarily correspond to meaningful or disentangled features. Some of the semantic
interpretations presented in the paper appear a bit subjective, and the process used to assign
these labels is not clearly explained. More systematic analysis or automated labeling methods
would help support the interpretability claims.

---

> ### Author Rebuttal · Authors · 2026-03-30
>
> We sincerely thank you for your valuable feedback. Following your guidance, we have added new experiments to strengthen our work.
>
> ### **Weaknesses:** *Lack of systematic analysis or automated labeling for singular vector interpretation.*
>
> In this work, we employ an LLM as an automated labeling tool to interpret singular vectors and conduct a series of analyses.
> - In Appendix D.2 (Figure 9), we provide the prompt design, example outputs, and the LLM used for automatic interpretation.
> - In Appendix H.1, we perform a layer-wise statistical analysis of interpretable subspaces across models of varying scales.
> - In Appendix H.2, we examine the spectral localization of highly interpretable subspaces in MLP matrices, analyzing how their alignment varies with network depth and model size.
> - In Appendix J, we present illustrative examples of interpretable subspaces.
> - In Appendix C, we discuss potential considerations when relying on external LLMs for automated subspace interpretation.
>
> ### **Questions 1:** *Could the proposed framework, or a simple extension of it, be used to analyze matrices involved in the attention mechanism (Q, K, V projections), and if so, would the same detector-effector interpretation still hold?*
>
> 1. Our work, based on the observation by Beren & Black [1] that singular vectors of MLP weights encode semantic structure, moves beyond a purely representational view to formalize these spectral components as detector-effector units that directly mediate token prediction, thereby revealing sparse, dominant pathways.
> 2. Beren and Black [1] also explored interpretability methods for attention mechanisms based on SVD. Intuitively, our detector-effector explanation remains applicable. However, unlike the relatively independent semantic transformations in MLPs, attention mechanisms involve dynamic weighting and cross-position information aggregation. We need to consider these factors when extending our approach to attention mechanisms.
>
> ### **Questions 2:** *How robust are the identified subspaces across different prompts or datasets? Do the same subspaces consistently appear as dominant contributors?*
> In **Section 4.3 (Part 2) and Appendix E, we provide a comprehensive experimental analysis regarding this question.** Our results suggest that the identified subspaces are robust across different prompts, with the same subspaces consistently emerging as primary contributors.
> - In our experiments, we investigate whether certain subspaces are consistently highly active and serve as interpretable mechanisms for specific contextual semantics using *Controlled Variation Sets*.  These sets consist of a target-variance dataset $D_{target}$ and a prefix-variance dataset $D_{prefix}$, both focused on positive-adjective semantics.
> - Figure 6 (Section 4.3, Part 2)  illustrates the top-10 active subspaces in Layers 14-17 of GPT2-Medium, identified by their recurrence frequency among the top-10 contributors across all positive semantic samples. Appendix E.1 further provides layer-wise statistics of active subspaces across GPT-2, GPT2-Medium, GPT2-Large, and GPT2-XL. The results show that only a small subset of subspaces are consistently highly active within each layer, suggesting that a limited number of subspaces serve as principal carriers of semantic information.
> - We further examine whether these highly active subspaces exhibit direct and interpretable associations with positive semantic attributes. Table 1 (Section 4.3, Part 2)  details the semantic interpretations of these subspaces under $\mathcal{D}_{\text{target}}$, where L$m$\_SP$i$ denotes subspace $i$ in layer $m$. Appendix E.2 provides further layer-wise interpretations of active subspaces across various  models.  Our results show that these high-frequency subspaces are typically related to positive semantics.
>
> ### **Questions 3:** It remains unclear whether the proposed approach scales effectively to larger architectures with billions of parameters while remaining computationally efficient and informative.
> - Our evaluation spans GPT-2, GPT2-Medium, GPT2-Large, and GPT2-XL, with model sizes **117 million~1.5 billion** parameters. Following your suggestion, we have included GPT-Neo-2.7B and GPT-J-6B, with **2.7 billion** and **6 billion** parameters respectively, and evaluated the computational cost of constructing an single-layer interpretability datastore on a NVIDIA A100 GPU (40 GB).
>
> - **All new results are provided in Figures 1 and 2 via the following link**: https://github.com/Anonymous2026SSS/results/blob/main/to_reviewer_wjvF.md  . Both total runtime and peak GPU memory consumption exhibit a linear relationship with matrix size, indicating that our method can maintain computational efficiency when scaled to larger models.
>
> [1] The Singular Value Decompositions of Transformer Weight Matrices are Highly Interpretable.

---

> > ### Author Rebuttal · Reviewer_wjvF · 2026-04-02
> >
> > Thank you for your efforts in clarifying and completing your results based on my comments, in particular extending to much larger models. At this point I have no other changes or clarifications to ask and am satisfied with the proposed modifications.

---

> > > ### Author Response · Authors · 2026-04-02
> > >
> > > We sincerely appreciate your time and thoughtful feedback, which have greatly improved this manuscript. We are pleased to know that our previous revisions and clarifications have successfully addressed your concerns.

---

### Official Review · Reviewer_rK9R · 2026-03-13

**Soundness:** 3
**Presentation:** 3
**Significance:** 2
**Originality:** 3
**Overall Recommendation:** 4
**Confidence:** 4

**Summary:**

This paper propose a novel, SVD-based mechanistic interpretability method Detector-Effector Units(DEUs), which decompose the up-projection and down-projection matrix of MLP into interpretable rank-1 units. Then, this paper introduces Subspace Contribution Analysis(SCA) which quantifies the causal contribution of any unit to any token. The authors prove these two can help find the crucial pathway in next token prediction from the perspective of fidelity and faithfulness across a wide task scope, and are interpretable. The authors also compare SCA with features of SAEs and Transcoders, and show that their units have better fidelity on the prediction of target token. Another advantage of their framework compared to sparse dictionary learning methods is its extremely high efficiency and no error is introduced.

**Compliance With Llm Reviewing Policy:**

Affirmed.

**Key Questions For Authors:**

1. How do we compare the degree of interpretability with SAE and transcoder features? Although I understand it may not be the focus of this paper. However, it's an issue that cannot be ignored when comparing new framework with sparse dictionary learning methods like SAE and transcoders.
2. How will the results of the experiment in Figure 2,3,4,5 change when using Controlled Variation Sets (the target token is not the rank 1 token in probability)? I am particularly curious about the changes in the results in Figure 4. Will the target token's rank rise to first place as the perturbation coefficient increases?

**Limitations:**

Yes, authors have discussed the limitations in Appendix B.

**Strengths And Weaknesses:**

strengths:
1. Despite building on some prior research, the authors provide substantial empirical evidence to prove that, when mapping DEU(the SVD result of MLP projection matrix) to the word (un)embeding matrix, the top activating tokens exhibit considerable interpretability.
2. Experiments were conducted on various tasks and model sizes when demonstrating that DEU found using SCA has a causal effect and good fidelity on the target token.
3. The entire framework requires very few resources compared to sparse dictionary learning methods.
4. This paper is well written.
weakness:
1. This framework provides limited depth in interprete the model compared with SAE and transcoder: the interpretable units(DEUs) include only features related to the current token or to the target token predicted at current position. In contrast, SAE and transcoder can capture high-level features (e.g. features required summarizing context).
2. The evaluation of the degree of interpretability of singular directions lacks a baseline: simply checking whether the top activating tokens of a direction are monosemanticity is insufficient. I suggest using the same criterion to evaluate the randomly sampled directions.

---

> ### Author Rebuttal · Authors · 2026-03-30
>
> We sincerely thank you for your valuable feedback. Following your suggestion, we have added extensive **new experiments**, which have further strengthened our work.
> ### **Strengths and Weaknesses 5:**
> Thank you for your suggestion. We have incorporated it into the Discussion section. Below are some insights regarding our method:
>  1. Unlike TransCoder and SAE, within the NaNA framework, SCA serves as a directed pathway extraction method, distinguished by its explicit filtering of *causal bridges*. We define these bridges as subspaces that simultaneously exhibit input activation (detector alignment) and target alignment (effector alignment).
> 2. In addition, the input to the MLP layers is not the original token embedding, but the residual stream after processing through multiple layers of attention. In this context, the DEU's "detector" ($v_i$) identifies a dominant direction in the input vectors that inherently encodes contextual information.
>
> ### **Strengths and Weaknesses 6:**
> - In **Appendix G.3 of the paper, we have provided a comparison between ''Randomly Sampled Subspaces'' and the ''Top‑k High-Contribution Subspaces''** obtained via the SCA method, in both general and ablation experiments.
>
> ### **Questions 1:**
> 1. Thanks for your suggestion. We have incorporated this into the Discussion section for further exploration. We consider the primary evaluation criterion to be the degree to which high-contribution components faithfully reflect the behavior of the original model. Rather than merely assessing whether the features appear intuitively meaningful, it is crucial to examine whether they truly govern the model’s behavior.
> 2. In Section 4.4 (part 1), we evaluate the reconstruction fidelity of the top-$k$ active components in recovering the target token's prediction on the positive semantics dataset. Furthermore, we extend the experiments to the gender pronoun, subject–verb agreement (simple structure), and subject–verb agreement (relative clause) datasets. ***Full new results are provided in Figures 1-3 via the following link:*** https://github.com/Anonymous2026SSS/results/blob/main/to_reviewer_onrA.md .
>
> ### **Questions 2:**
> 1. The NaNA framework is fully applicable even when the target tokens do not correspond to the model’s original top-1 predictions.  **In Appendix G, we provide 20 representative cases**. Notably, for these 20 anchor cases, the target tokens need not match the top-1 predictions. For example, see case 6.
> 2. Following your suggestion, we conducted **new experiments** on "*The capital of Germany is*" → "*Frankfurt*" and on interventions in subspaces unrelated to the target token.
>    - **Experiment_1: Causal Evidence of Pathway-Driven Predictions (When the Target Token Is Not Rank-1).** Following the experiment in Section 4.2 (part 2) of the paper, we use the NaNA framework to extract the pathway for "*The capital of Germany is*" → "*Frankfurt*".  We perform both *standard and ablation experiments* by reconstructing the middle-to-late MLP layers of GPT-2 Medium using an increasing number of high-contribution subspaces.  The results demonstrate that using only the top-15 subspaces is sufficient to successfully predict "*Frankfurt*" for the prompt "*The capital of Germany is*". ***Full new results are provided in Figure 1 via the following link***: https://github.com/Anonymous2026SSS/results/blob/main/to_reviewer_rK9R.md .
>    - **Experiment_2: Impact of Pathways on Predictive Performance (When the Target Token Is Not Rank-1).** Following the experiment in Section 4.2 (part 1) of the paper, we conduct *standard and ablation experiments* to evaluate the impact of high-contribution subspaces on the target token’s probability (↑) and rank (↓) in the case "*The capital of Germany is*" → "*Frankfurt*". The results indicate that the dominant subspaces identified by SCA are necessary for predicting the specific target token. ***Detailed results are shown in Figure 2 and are accessible via the following link***: https://github.com/Anonymous2026SSS/results/blob/main/to_reviewer_rK9R.md .
>    - **Experiment_3: Subspace Interventions.** Following the experiment in Section 4.2 (part 3) of the paper, we visualize the effects of manipulating Subspaces 5 and 9 of Layer 16 in GPT2-Medium for the prompt "*The cat looks very*".  The table below presents the semantics of these subspaces and their contributions.  We observe that amplifying or suppressing subspaces 5 or 9 correspondingly enhances their positive or negative semantics. ***Results can be found in Figure 3 via the link***: https://github.com/Anonymous2026SSS/results/blob/main/to_reviewer_rK9R.md .
>
>      **Table 1. Semantic Interpretation and Contributions of Subspaces 5 and 9.** Here, we consider the prompt "*The cat looks very*".
>      |Subspace|  Negative Direction | Positive Direction | Contribution|
>      |-|-|-|-|
>      | 5 | insufficiency concepts | Inspection Actions | 0.073 |
>      | 9 |Criticismand Feedback | historical decades| 0.0623 |

---

> > ### Author Rebuttal · Reviewer_rK9R · 2026-04-07
> >
> > We thank the authors for providing additional details and replies. I still lean towards acceptance.

---

> > > ### Author Response · Authors · 2026-04-07
> > >
> > > We sincerely thank the reviewer for their positive assessment and encouraging feedback. We are glad that the additional details and clarifications have addressed the concerns. We appreciate your support.

---

### Official Review · Reviewer_onrA · 2026-03-13

**Soundness:** 3
**Presentation:** 3
**Significance:** 3
**Originality:** 3
**Overall Recommendation:** 5
**Confidence:** 4

**Summary:**

The authors propose a framework called Native Network Anatomy (NaNA), which is designed for mechanistic interpretability. It relies on applying the Singular Value Decomposition (SVD) technique to the weight matrices in the MLP sublayers of the Transformer model. To evaluate the performance of the proposed framework, they used several variants of GPT2 Transformer and 3 constructed datasets.

**Compliance With Llm Reviewing Policy:**

Affirmed.

**Final Justification:**

The authors conducted additional experiments using SOTA baselines and datasets. They addressed my main concerns. Based on this, I have revised my score.

**Key Questions For Authors:**

- Please see the comments

**Limitations:**

yes

**Strengths And Weaknesses:**

The authors propose a framework called Native Network Anatomy (NaNA), which is designed for mechanistic interpretability. It relies on applying the Singular Value Decomposition (SVD) technique to the weight matrices in the MLP sublayers of the Transformer model. To evaluate the performance of the proposed framework, they used several variants of GPT2 Transformer and 3 constructed datasets. The method idea seems interesting. The main concerns are as follows:

1- In  (page 14, section D.1. Datasets), you claimed that you constructed 3 datasets. However, in the literature, we adopt popular real-world datasets used in previous works. Why was this not the case in this work? Could you provide concrete proof that the method should work and maintain the same performance if tested on benchmark datasets (not your constructed ones)?

2- I suggest using at least a benchmark dataset already adopted in a previously published interpretability paper, and report the performance of your approach compared to baselines adopted.

3- Baselines. Section, p.3 , you mentioned that you used SAEs and TransCoders. Please include citations of the papers presenting these baselines.

4- As you used SAEs, I guess you pointed out vanilla SAEs, but in the literature, there are plenty of novel SAEs architectures that demonstrated good performance and outperformed Vanilla SAEs. Why did you not use some novel SAEs variants? I suggest to consider this in the experiments.

---

> ### Author Rebuttal · Authors · 2026-03-30
>
> We sincerely appreciate your feedback on our work.  Following your suggestion, we have added **three new baselines** and **three additional datasets**, which significantly enhance the comprehensiveness of our evaluation.
>
> ### **Weaknesses 1 and 2:** *Lack of evaluation on benchmark datasets*
>
> - We extend our study by including **three additional datasets**, with experimental details following Section 4.4 (Part 1). **Three additional datasets include:**
>   - Gender Pronoun Task [1]
>   - Subject–Verb Agreement Task (Simple Structure) [2]
>   - Subject–Verb Agreement Task (Relative Clause) [2]
> - Across these datasets, our method (SCA) consistently outperforms all baseline models. **All new results are shown in Figures 1–3 and are available at the following link:** https://github.com/Anonymous2026SSS/results/blob/main/to_reviewer_onrA.md  .
>
>
> ### **Weaknesses 3:** *Citations for baseline models*
>
> - In **Appendix D.2**, we provide the access links, references, and parameters for TransCoder, SAE, and GPT-2 series models.
> - We appreciate your helpful suggestion and have added citations for TransCoder and SAE in Section 4.1 of the revised version.
>
>
> ### **Weaknesses 4:** *Lack of evaluation on advanced SAE variants*
>
> - We extend our study by including **three additional advanced SAE variants** across the Gender Pronoun, Subject–Verb Agreement (Simple Structure), and Subject–Verb Agreement (Relative Clause) datasets, with experimental details following Section 4.4 (Part 1) of the paper. **Three advanced SAE variants include:**
>   - JumpReLU SAE [3]
>   - Top-K SAE [4]
>   - BatchTopK SAE [5]
>
> - **All new results are provided in Figures 1, 2 and 3 via the following link**: https://github.com/Anonymous2026SSS/results/blob/main/to_reviewer_onrA.md  . Across these datasets, our method (SCA) consistently outperforms all baseline models on Average Target Rank ($\downarrow$) and Average Target Probability ($\uparrow$).
>
> [1] LLM Circuit Analyses Are Consistent Across Training and Scale\
> [2] Sparse Feature Circuits: Discovering and Editing Interpretable Causal Graphs in Language Models\
> [3] Improving Reconstruction Fidelity with JumpReLU Sparse Autoencoders  \
> [4] Scaling and evaluating sparse autoencoders \
> [5] BatchTopK Sparse Autoencoders

---

> > ### Author Rebuttal · Reviewer_onrA · 2026-04-03
> >
> > I am thankful to the authors for their response and the experiments conducted using additional SOTA baselines and datasets. Their response addressed my concerns. Based on this, I have revised my score.

---

> > > ### Author Response · Authors · 2026-04-03
> > >
> > > We are very grateful for your review suggestions, which have significantly improved our manuscript. Thank you again for the time and dedication you have invested in our work.

---

### Decision · Program_Chairs · 2026-04-30

**Decision:**

Accept (spotlight)

**Comment:**

This paper proposes to use SVD as a fast interpretability method, which is a simple idea, but very important to pursue. Too much interpretability fails to consider simple baselines and take a hard look at whether a new more complicated and expensive method is actually providing differential value over simpler alternatives. I hope that this work helps fight against that trend.

I broadly agree with the reviewer's positive sentiment, the work seems sound even if the writing needs improvements. The authors had an engaging discussion that has improved the paper quality.